# The influence of the ocean circulation state on ocean carbon storage and $CO_2$ drawdown potential in an Earth system model

Malin Ödalen[1], Jonas Nycander[1], Kevin I. C. Oliver[2], Laurent Brodeau[1,3], and Andy Ridgwell[4,5]

[1]Department of Meteorology, Stockholm University, 106 91 Stockholm, Sweden
[2]National Oceanography Centre, Southampton, University of Southampton, Southampton SO14 3ZH, United Kingdom
[3]Barcelona Supercomputer Center, Barcelona, Spain
[4]School of Geographical Sciences, Bristol University, Bristol, UK
[5]Department of Earth Sciences, University of California-Riverside, Riverside, CA, USA

*Correspondence to:* Malin Ödalen (malin.odalen@misu.su.se)

**Abstract.** During the four most recent glacial cycles, atmospheric $CO_2$ during glacial maxima has been lowered by about 90-100 ppm with respect to interglacials. There is widespread consensus that most of this carbon was partitioned in the ocean. It is however still debated which processes were dominant in achieving this increased carbon storage. In this paper, we use an Earth system model of intermediate complexity to explore the sensitivity of ocean carbon storage to ocean circulation state. We

carry out a set of simulations where we run the model to pre-industrial equilibrium, but where we achieve different states of ocean circulation by changing forcing parameters such as wind stress, ocean diffusivity and atmospheric heat diffusivity. As a consequence, the ensemble members also have different ocean carbon reservoirs, global ocean average temperatures, biological pump efficiencies and conditions for air-sea $CO_2$ disequilibrium. We analyse changes in total ocean carbon storage and separate it into contributions by the solubility pump, the biological pump and the $CO_2$ disequilibrium component. We also relate these

contributions to differences in strength of ocean overturning circulation. Depending on which ocean forcing parameter that is tuned, the origin of the change in carbon storage is different. When wind stress or ocean diapycnal diffusivity is changed, the response of the biological pump gives the most important effect on ocean carbon storage, whereas when atmospheric heat diffusivity or ocean isopycnal diffusivity is changed, the solubility pump and the disequilibrium component are also important and sometimes dominant. Despite this complexity, we obtain a negative linear relationship between total ocean carbon and the

combined strength of the northern and southern overturning cells. This relationship is robust to different reservoirs dominating the response to different forcing mechanisms. Finally, we do a drawdown experiment, where we investigate the capacity for increased carbon storage by artificially maximising the efficiency of the biological pump in our ensemble members. We conclude that different initial states for an ocean model result in different capacities for ocean carbon storage, due to differences in the ocean circulation state and the origin of the carbon in the initial ocean carbon reservoir. This could explain why it is

difficult to achieve comparable responses of the ocean carbon pumps in model inter-comparison studies, where the initial states vary between models. We show that this effect of the initial state is quantifiable. The drawdown experiment highlights the importance of the strength of the biological pump in the control state for model studies of increased biological efficiency.

# 1 Introduction

The transition from interglacials to glacial maximums is associated by a substantial reduction in atmospheric $CO_2$ (henceforth, $pCO_2^{atm}$). During the last four glacial cycles (since $\sim 400,000$ years B.P.), the decrease has been about 1/3, and atmospheric $pCO_2$ at these glacial maxima (marine isotope stages 2, 6, 8 and 10) was approximately 180 ppm (see e.g., Petit et al., 1999; Lüthi et al., 2008, and references therein). When $CO_2$ dissolves in water, most of the molecules react with the water to form bicarbonate and carbonate ions. To achieve equilibrium between the atmospheric and surface ocean partial pressures of $CO_2$, further dissolution is then required (c.f. e.g. $O_2$, which does not react with water). As a consequence, the ocean holds 50 times more carbon than the atmosphere (Williams and Follows, 2011; Falkowski et al., 2000) and over 13 times that of the terrestrial biosphere (Falkowski et al., 2000; IPCC, 2007). Due to this size difference between the carbon reservoirs, a larger fraction of the $CO_2$ that was missing from the atmosphere during glacials is likely to have been stored in the deep ocean than in the terrestrial biosphere (Kohfeld and Ridgwell, 2009).

That the oceanic carbon storage increased during glacials is a well established idea, and there are numerous studies of how and why this may have happened (e.g., Broecker, 1982; Sarmiento and Toggweiler, 1984; Archer et al., 2000a; Sigman and Boyle, 2000; Brovkin et al., 2007; Hain et al., 2010; Sigman et al., 2010). However, the relative effects of different processes contributing to this oceanic uptake have not yet been well constrained and, so far, there is a lack of consensus on which processes that were dominant (reviewed in Kohfeld and Ridgwell, 2009).

To understand the controls on oceanic storage of $CO_2$ and the different processes involved, it is helpful to think about the different pathways that exist for carbon that is taken up in the surface layer to reach the deep ocean. These pathways are often referred to as the solubility pump and the biological pump (further described in Section 2). In this work, we focus on better constraining: 1) the effects of changes in global ocean mean temperature on the abiotic ocean-atmosphere $CO_2$ equilibrium and hence on the solubility pump; 2) the effect of changed $CO_2$ disequilibrium; and 3) the effects of increased efficiency of the biological pump. By performing ensemble runs using the Earth system model cGENIE (Ridgwell et al., 2007; Cao et al., 2009), we examine the changes in the carbon system that result from changes in ocean circulation.

Changes in ocean circulation, which can be due to climate change or other independent physical processes (e.g. tectonic and ocean gateway changes, such as the opening of the Drake Passage, (e.g., von der Heydt and Dijkstra, 2006; Sijp et al., 2014)), will lead to changes in global ocean mean temperature, through e.g. changes in the locus and strength of deep water formation. Firstly, if surface ocean temperature changes, this will have a direct effect on $CO_2$ solubility and hence on the atmosphere-ocean $CO_2$ equilibrium. Secondly, if the temperature, and thus the concentration of dissolved $CO_2$, in the deepwater formation areas changes, this will impact on the deep-ocean concentration of $CO_2$ (Goodwin et al., 2011). Ocean circulation changes will also affect the ocean carbon content by influencing nutrient distributions, biological efficiency and time scales for outgassing of $CO_2$ in upwelling areas.

When studying glacial ocean $CO_2$ uptake, the most common modelling approach is to aim at reproducing a glacial climate by adjusting physical parameters, such as orbital parameters, $pCO_2^{atm}$, bathymetry, sea level, topography and/or ice sheets (e.g., Ganopolski et al., 2010; PMIP3), in ways that they may have changed during glacials. These changes influence the ocean

circulation by affecting e.g. climate (temperature), tidal dissipation (Schmittner et al., 2015) and wind stress (Sime et al., 2013). Model studies of glacial climate are generally based on a pre-industrial control state, where $pCO_2^{atm}$ is prescribed, while the circulation model is tuned in order to achieve the target ocean fields of tracers such as salinity, temperature, and dissolved chemical compounds (e.g. Heinze et al., 1991; Brovkin et al., 2007; PMIP3). However, the desired tracer fields can be achieved through multiple different combinations of the tuning parameters, which means that similar tracer fields can be achieved in different model states despite differences in their circulation. In this study, we investigate the consequences for the carbon storage of the initial state circulation differences that result from this tuning: are there other aspects of the climate system, such as the strengths of the ocean carbon pumps, that become so significantly different that they can be crucial for the outcome of e.g. model inter-comparison studies of glacial $CO_2$ drawdown? As discussed in Zhang et al. (2013), overturning circulation differences in an initial glacial state could similarily cause differences in results in model inter-comparison studies of deglacial $CO_2$ rise.

Specifically, we aim to clarify how the initial equilibrium state, not only in terms of ocean circulation, but in terms of ocean carbon inventory and the origin of the already stored carbon (e.g. biological or solubility pump), is crucial for the outcome of a study investigating increased efficiency of the biological pump. This will provide insight about why it is difficult to compare results from different model studies that have attempted to simulate and explain the glacial lowering of $pCO_2^{atm}$. One example of a model study where this dependence on the initial state may have been key is Archer et al. (2000b). They investigated the abiotic chemical equilibrium in a few different models and showed that there was a consistent difference between box models and general circulation models (GCMs). They attributed this to differences in complexity, although Ridgwell (2001) found that the box vs. GCM differences could be explained by inappropriately low ocean surface temperature assumptions in the original box model descriptions. However,Archer et al. (2000b) also found a significant difference in behaviour between different GCMs, which they were unable to explain. We show that such differences could instead be due to differences in the initial state, where differences in circulation are causing the strengths of the carbon pumps, and thus model carbon inventories, to be different. More importantly, we highlight that this effect of the initial state is quantifiable.

In this study, we do not attempt to reproduce glacial climate. In the first step of the modelling, we instead carry out a process study where we change physical parameters in the model, one or two at a time, while restoring $pCO_2^{atm}$ to the pre-industrial value. The parameters we change are common tuning parameters in climate models, such as wind stress intensity and ocean diffusivity. This approach allows us to see how the ocean circulation changes that follow from tuning each of these parameters affect the ocean carbon storage. We are particularly interested in the relative importance of the changes in the solubility pump, the biological pump and the $CO_2$ disequilibrium. Through the first step, we will show that the effect of changes in ocean circulation on global ocean mean temperature, and thus on the solubility pump, is non-negligible and in some cases of similar importance as the effect on the biological pump. We will also show that the relative contributions by the biological pump, the solubility pump and by $CO_2$ disequilibrium to the ocean carbon inventory in the initial equilibrium state depends on the model tuning strategy, yet these combine in a manner to give a response in total carbon that is more straightforwardly related to the circulation than any of the individual components. This first step will also allow us to discuss how specific changes to

circulation parameters will influence the total carbon uptake in model simulations of e.g. glacial scenarios. In particular, we will focus on the influence of the strength of the global and basin scale overturning circulation.

In the second step, we enforce 100 % nutrient utilisation efficiency (see Section 2) in the different circulation patterns of the ensemble. This allows us to measure the difference in drawdown potential for $CO_2$ between different ocean circulation states. A new equilibrium in $CO_2$ between atmosphere and ocean will be established, and thus also $pCO_2^{atm}$, will be different

depending on the ocean circulation. This will illustrate how the initial state of a model can be important for the outcome of a glacial $CO_2$ drawdown experiment and highlight the importance of differences in the initial states of models in inter-comparison studies. The effect of the solubility pump on $pCO_2^{atm}$ has previously been studied by e.g. DeVries and Primeau (2009). The effect of air-sea disequilibrium of $CO_2$ and its connection with biological carbon sequestration was studied by Ito and Follows (2013) and in recent work by Eggleston and Galbraith (2017) (in review). Our theoretical approach is similar to those taken

by Marinov et al. (2008a), Marinov et al. (2008b) and Kwon et al. (2011). However, these studies all focus mainly on the contributions to ocean dissolved inorganic carbon (DIC) by the biological pump. In this study, we aim to determine a robust relationship between a model's initial-state and its drawdown potential that accounts for the biological pump, the solubility pump and disequilibrium contributions. Such a relationship, if it is sufficiently simple, will be useful in assessing the effect of model biases on the sensitivity of the carbon cycle to climate perturbations.

## 2 Framework and general concepts

$CO_2$ that is dissolved in the ocean surface layer is often described as being able to reach the deep ocean via two pathways – the solubility pump and the biological pump. These pathways are thouroughly described in Volk and Hoffert (1985) and later in Williams and Follows (2011). More details are given in Appendix A1. Our separation of DIC into its different sources of origin (see Section 3.2), which is related to the two carbon pumps, will largely follow the framework of Ito and Follows (2005). This

framework has been widely applied (e.g., Williams and Follows, 2011; Marinov et al., 2008b; Kwon et al., 2011; Lauderdale et al., 2013).

Of the total ocean carbon reservoir, about 90 % is expected to be preformed carbon, i.e. from the solubility pump, and the remaining 10 % organic or regenerated carbon, i.e. from the biological pump (Williams and Follows, 2011). If the net capacity of the carbon pumps increased, this would act to decrease $pCO_2^{atm}$. For instance, increased efficiency of the biological pump is

a frequently proposed explanation for the glacial $CO_2$ drawdown (e.g. Sarmiento and Toggweiler, 1984; Martin, 1990; Sigman and Boyle, 2000). By "increased efficiency" we mean that more of the available nutrients in the surface layer are used for biological production before the water is subducted into the deep ocean. When the nutrient utilisation efficiency ($NUE$) is increased, the biological pump gets stronger and transfers more carbon to the deep ocean. As a result, the air-sea equilibrium of $CO_2$ is shifted and more $CO_2$ is drawn from the atmosphere into the surface layer of the ocean, reducing $pCO_2^{atm}$.

As described in the framework of Ito and Follows (2005), $NUE$ can be described in terms of the parameter $\overline{P^*}$, which is the ratio between the global averages of regenerated and total nutrients in the ocean (see Appendix A2). In Earth system models, how much $CO_2$ that can be removed from the atmosphere by increasing the $NUE$ depends on $\overline{P^*}$, which can differ between

models and between different climate states achieved in the same model (Marinov et al., 2008b). The difference between $pCO_2^{atm}$ in the initial equilibirum state, $pCO_2^{eq}$, and the lowest $pCO_2^{atm}$ that can be achieved by increasing $NUE$, $pCO_2^{min}$ (achieved when $\overline{P^*} = 1$), will be referred to as the $CO_2$ *drawdown potential* of a model, $DP$:

$$DP = pCO_2^{eq} - pCO_2^{min} \tag{1}$$

When the biological pump is working at maximum efficiency (when $\overline{P^*} = 1$), a specific amount of carbon proportional to the total amount of nutrients in the ocean will be trapped in the deep ocean at all times, assuming fixed stoichiometric ratios (see Section 3.1.1). This carbon will not participate in the chemical equilibrium between atmosphere and ocean which decides the $pCO_2^{atm}$.

An important component of the oceanic carbon cycle is alkalinity. It is related to the buffer capacity of the ocean; hence, the ocean's capacity to resist a change in pH despite the addition of an acid, such as $CO_2$. Biological production affects the vertical distribution of ocean alkalinity through the hard-tissue pump. When the $CaCO_3$ shells of micro-organisms are exported to the deep ocean and dissolved, alkalinity is returned to solution. Hence, some of the alkalinity in the deep ocean is of biological, or regenerated, origin. This part of the alkalinity, we will denote $A_{reg}$. The rest of the alkalinity in the deep ocean was set at the surface and then brought into the deep ocean by the circulation; this is preformed alkalinity, $A_{pre}$. Total alkalinity, $A_T$, is

$$A_T = A_{pre} + A_{reg} \tag{2}$$

This partitioning of alkalinity will be useful for understanding the relative importance of the hard-tissue biological pump in oceanic carbon storage.

It is likely that ocean alkalinity increased during glacials due to more weathering of carbonates caused by lower sea levels (e.g., Munhoven, 2002; Brovkin et al., 2012). This may have contributed to drawdown of $CO_2$ into the ocean (see e.g. Sigman and Boyle, 2000, and references therein). In order to model the effect of alkalinity changes on $pCO_2^{atm}$, we would need an open ocean-atmosphere system with river supply and sedimentation of alkalinity. This would require a different type of modelling than we do here. Our analysis is restricted to the ocean-atmosphere system, excluding sediment feedbacks. $A_T$ thus has no external sources/sinks and is only affected by the biological pump.

## 3   Methods

### 3.1   Model

We use the model cGENIE, an Earth System Model of Intermediate Complexity (EMIC), which is a computationally efficient model developed for studying the ocean carbon cycle on timescales of 100–100,000 years. cGENIE is higher in complexity than box models, but is still efficient enough to allow running a large ensemble to equilibrium for the carbon system. In terms of ocean carbon system tracers, the minimum required for this type of study is $P$, $DIC$, $O_2$ and $A_T$. cGENIE includes many

additional tracers, out of which only some are used in this study. For example, particulate ($POC$) and dissolved organic carbon ($DOC$) are included in the calculations of model carbon inventory (see Appendix B). Of particular importance for this study is the possibility to run with preformed tracers ($P_{pre}, C_{pre}, O_{2_{pre}}, A_{pre}$, see Section 3.1.2). Model characteristics are described in Edwards and Marsh (2005); Ridgwell et al. (2007) and Cao et al. (2009).

The physical ocean is modeled using a frictional-geostrophic 3D model on a 36x36 equal area grid in the horizontal and 16 depth levels. The atmospheric model is an Energy Moisture Balance Model (EMBM) with prescribed, climatological wind-fields (Cao et al., 2009). Ocean biogeochemistry and atmospheric chemistry are treated by separate modules that are coupled to the physical models and to each other (Ridgwell et al., 2007). The biogeochemical module is based on a phosphate-only nutrient scheme. Hence, phosphate (P) is the limiting nutrient. The model description of export flux of organic matter is based on available surface nutrients (see Ridgwell et al., 2007, Eqs. 1-4), and instead of having a "standing plankton biomass" in the model, the export of particulate organic matter is derived directly from uptake of $P$. Remineralisation in cGENIE primarily depends on oxygen availability (Ridgwell et al., 2007). $POC$ is modelled as two fractions: one more easily degradable (labile) fraction, which undergoes an exponential decay with depth, and one fraction that is more resistant to degradation, which remineralises at the ocean floor. In the drawdown experiment (Section 3.2), the length scale of the exponential decay is adjusted. The remineralisation of CaCO$_3$ in the water column is treated in a similar manner to $POC$. The carbonate precipitation rate is thermodynamically-based and relates export flux of CaCO$_3$ to the flux of $POC$ (see Ridgwell et al., 2007, Eq. 8). As investigation of carbonate system feedbacks are not the purpose of this study, interactive sediments are not used, and in terms of carbon cycling, the atmosphere-ocean is studied as a closed system.

As our control state, we use the pre-industrial equilibrium state described in Cao et al. (2009). During the spin-up (10,000 years) to this equilibrium state, $pCO_2^{atm}$ is restored to 278 $\mu atm$ ($\approx$ 278 ppm), while the inventory of carbon in the model is allowed to change. Henceforth, this pre-industrial equilibrium state will be referred to as $PIES278$.

### 3.1.1 Stoichiometry

The stoichimoetric relationships in cGENIE is based on Redfield (1963). Thus, when organic material is formed, the elements are taken up in the proportions $N : P : C : O_2 = 16 : 1 : 106 : -138$. For example, 138 moles of $O_2$ are released for every mole of phosphorus that is taken up. The same relationship applies to the decomposition of organic material, which releases N, P and C, but consumes $O_2$. The nitrogen (N) cycle is not modelled, but the effect of N on alkalinity during production and remineralisation of organic matter is represented, based on the fixed stoichiometric relationship with P. Adjustments to the stoichiometric ratios given in Redfield (1963) have been proposed by e.g. Takahashi et al. (1985) and Anderson and Sarmiento (1994), but the classic Redfield ratios are still widely accepted and used. The choice of constants is not crucial for the outcome of the study, and we have hence stayed with the default model setup of the official release of cGENIE.

More recently, the stoichiometry of production of new organic material has been shown to be highly variable, between species but also within the same species while living under different conditions, such as nutrient availability (e.g., Quere et al., 2005; Galbraith and Martiny, 2015). While this does not contradict that the ratios are on average similar to the results by

Redfield (1963), this *in-situ* variability in stoichiometric ratios could potentially be important in a glacial scenario. However, evaluating the influence of such variability is beyond the scope of the present study.

### 3.1.2 Preformed tracers

In cGENIE, we also employ a set of preformed nutrients, carbon, oxygen and alkalinity ($P_{pre}$, $DIC_{pre}$, $O_{2pre}$ and $A_{pre}$). The initial concentrations of the preformed tracers are set in the surface ocean, where they are set equal to the concentration of the corresponding active tracers. After the surface water is submerged, they are passively advected and mixed in the interior ocean. We use these tracers to calculate e.g. $\overline{P^*}$ (see Appendix A2), apparent oxygen utilisation ($AOU$, see Appendix B2) and for separating ocean DIC into categories of different origin (see Appendices B1 and B2).

Preformed tracers are a recent addition to the model, which have only been used to a limited extent in previous studies (Goodwin et al., 2015). Studies using other models have shown the usefulness of e.g. $P_{pre}$, $DIC_{pre}$, $A_{pre}$ as explicit model tracers (e.g., Marinov et al., 2008b; Duteil et al., 2012; Bernardello et al., 2014; Eggleston and Galbraith, 2017), but in this case we expand this by using as many as four preformed tracers simultaneously. This eliminates errors that would result from using a linear regression for $A_{pre}$ (Lauderdale et al., 2013; Eggleston and Galbraith, 2017), or from the presence of oxygen disequilibrium in the calculation of $AOU$ (Eq. B3), as discussed by Ito et al. (2004). Consequently, there is no need to make simulations with infinitely fast gas exchange, which is a common approach to remove errors due to $O_2$ disequilibrium (e.g., Marinov et al., 2008b). Concerns as those raised by Bernardello et al. (2014), regarding the 'back-tracking' methods of the Ito and Follows framework, are hence not an issue here. $P_{pre}$ allows direct determination of nutrient utilisation efficiency, eliminating any sources of error that could be associated with indirect methods of determination. Finally, $DIC_{pre}$ allows us to estimate the errors associated with the carbon species separation (see e.g. Section 4.1.4).

### 3.2 Theory and experimental setup

The inventory of total carbon, $TC$ [mol], in the equilibrium state can be described by

$$TC = M_a \, pCO_2^{atm} + M_o(\overline{C_{sat}} + \overline{C_{soft}} + \overline{C_{carb}} + \overline{C_{res}}) \tag{3}$$

Equation (3) sums the contributions to $TC$. Changes in e.g. land carbon are not modelled and therefore not included in this equation. The atmospheric carbon content is given by the partial pressure of $CO_2$ times the number of moles of gas in the atmosphere, $M_a$. $M_o$ is the mass of the ocean [kg]. $C_{sat}$ is the concentration [mol kg$^{-1}$] of carbon an individual water parcel would have had if it were in equilibrium with the atmosphere at the ambient temperature, salinity, alkalinity and concentration of $PO_4$. $C_{soft}$ originates from remineralisation of the soft tissue of biogenic material that has entered the water parcel. Carbon from biogenic hard tissue is denoted $C_{carb}$. $C_{res}$ is the residual needed to get the actual carbon concentration in the water parcel. $C_{res}$ contains three components; 1) The disequilibrium component $C_{dis}$, which we will quantify in Section 4.1.5. 2) Carbon in the form of particulate and dissolved organic matter. 3) Errors associated with any imperfect assumptions in the theory used for calculating $C_{sat}$, $C_{soft}$ and $C_{carb}$. The overbars represent global averages. The concentration $C_{pre}$ in the

water parcel consists of $C_{sat} + C_{dis}$, whereas $C_{soft} + C_{carb}$ gives regenerated carbon, $C_{reg}$. We calculate the contributions to Eq. 3 and any changes to this inventory (see Eq. 4) using methods of Kwon et al. (2011) and Lauderdale et al. (2013), when necessary solving the carbon system equations using the solver of Lewis et al. (1998). More details about the calculations of the contributions to $TC$ are given in Appendix B.

Initialising from our pre-industrial equilibrium state $PIES278$, we perform 12 different equilibrium experiments in which
one or two physical tuning parameters have been changed compared to the control (Fig. 1). The parameters and their ranges are further described in Section 3.3 and listed in Table 1. The experiments are run for 10,000 years, which is enough to reach a new equilibrium state. These 12 sensitivity experiment equilibrium states ($SE1–SE12$) are given descriptive notations listed in Table 1. These modifications of physical tuning parameters will cause the ocean circulation to change c.f. $PIES278$; the circulation gets weaker or stronger, overturning cells change their latitudinal extent etc. During the spin-up phase, $pCO_2^{atm}$
is still restored to 278 $\mu$atm (Fig. 1). The ocean reservoir of nutrients (in this case, $PO_4$) is the same as in $PIES278$, but the partitioning between $P_{reg}$ and $P_{pre}$ is changed, and thereby the strength of the biological pump. Likewise, the mean temperature of the ocean changes, and thereby the strength of the solubility pump. Hence, while the atmospheric carbon inventory is the same in all ensemble members as in $PIES278$, the ocean carbon inventory, as well as the $TC$ of these 12 ensemble members is different. We aim at comparing the drawdown potential of models that have the same $pCO_2^{atm}$ in
their initial equilibrium state, but different oceanic carbon distributions and inventories, which is usually the case in model inter-comparison projects. For example, the instructions for the LGM simulations within the framework of the current PMIP3-CMIP5 project specify the LGM $pCO_2^{atm}$ to be set to 185 ppm, whereas there are no specifications for the ocean carbon inventory (see https://pmip3.lsce.ipsl.fr/).

The change in the total carbon inventory, $\Delta TC$ [mol] between $PIES278$ and some equilibrium state $SE(n)$, where $n =$
$(1, ..., 12)$, can be described by

$$\Delta TC = M_o(\Delta \overline{C_{sat}} + \Delta \overline{C_{soft}} + \Delta \overline{C_{carb}} + \Delta \overline{C_{res}}) \tag{4}$$

The contributions by $C_{soft}$, $C_{carb}$ and $C_{sat}$ to this observed change in $TC$ in Eq. (4) are then evaluated as described in the subsections B1 and B2. The change in $C_{res}$ will simply be the residual between the observed change in $TC$ and the sum of the contributions by $C_{soft}$, $C_{carb}$ and $C_{sat}$. $pCO_2^{atm}$ is restored and thus the atmospheric carbon reservoir does not contribute to
$\Delta TC$.

The changes in inventories of $TC$, $C_{sat}$, $C_{soft}$, $C_{carb}$ and $C_{res}$ are translated into the equivalent change in $pCO_2^{atm}$ that would occur if we were not restoring it to pre-industrial levels, but instead kept $TC$ constant in the ensemble (cf. methods of Marinov et al., 2008a, b). This translation is performed as described in detail in Appendix C. This translation allows us to test the validity of the equation describing the effect on global ocean mean temperature effect on $pCO_2^{atm}$ suggested by Goodwin
et al. (2011) (see Appendix C).

Finally, starting from each of $SE1–SE12$ as well as from $PIES278$, we run experiments where the $NUE$ of biology is maximised (100 % efficiency) (Fig. 1) and again allow the model to run for 10,000 years to new equilibrium states ($DE1 - DE12$).

This reveals the differences in drawdown potential between ensemble members with different ocean circulation characteristics. Note that in this step, $pCO_2^{atm}$ is not restored, thus $TC$ is held constant between SE(n) and DE(n) and carbon is only redistributed between reservoirs (see Eq. 3). Eq. 4 is not applicable in this step, because it assumes constant $pCO_2^{atm}$. Maximum $NUE$, i.e. $\overline{P^*} = 1$ (see Eq. (A1)), is achieved by changing the remineralisation length scale in the model (Section 3.1). It is made deep enough (10,000 m) for any carbon that is taken up in biogenic material to be highly efficiently trapped in the deep ocean and not undergo any significant remineralisation. The concentration of dissolved $P$ (at the surface, and on a global annual mean) being reduced by two orders of magnitude in all $DE$s, due to $P$ being bound in organic material, confirms that this effect is achieved. However, due to convection, mixing and local remineralisation of dissolved organic matter, the surface concentration of $P$ does not go to zero. Note that the contribution of biogenic material to the carbon inventory $TC$ is substatial in this step. Thus, quantifying $\Delta C_{soft}$ and $\Delta C_{carb}$ for this step is not useful unless the carbon contained in biogenic material is also considererd to contribute to these reservoirs. However, the very deep remineralisation length scale is a highly hypothetical case and we therefore choose not to separate the contributions by the different carbon reservoirs to the drawdown of $pCO_2^{atm}$.

## 3.3 Sensitivity experiment tuning parameters

The physical tuning parameters that we change are atmospheric heat diffusivity, wind stress and ocean vertical and isopycnal diffusivity. They are selected because they are common tuning parameters (e.g., Müller et al., 2006; Marsh et al., 2013), which influence the ocean circulation.

For this study, our intention is for the ensemble to be representative of a wide range of plausible ocean circulation states. The chosen parameter ranges correspond to a halving and doubling of the values used in the control simulation. Our chosen values are within the parameter space explored for a predecessor to the GENIE-model by Edwards and Marsh (2005), except the low wind stress simulation (see below). Similar parameter ranges are also explored for GENIE by e.g. Marsh et al. (2013). For the most part, our selected values are within the parameter ranges that generate the subset Edwards and Marsh (2005) refer to as low-error simulations. In the Bern3D model, with physics based on Edwards and Marsh (2005) and thus similar to GENIE, Müller et al. (2006) doubled the observed wind stress (W = 2) to get a more realistic gyre circulation. Marinov et al. (2008a, b) used the Geophysical Fluid Dynamics Laboratory Modular Ocean Model version 3, which has the same default value for isopycnal diffusivity (1500 $m^2 s^{-1}$) as our model. Marinov et al. (2008a, b) explore a range of 1000-2000 $m^2 s^{-1}$ (cf. our range of 750-3000 $m^2 s^{-1}$).

When comparing with models that have different available tuning parameters, diagnostic variables such as temperature, salinity and AMOC volume transport can indicate whether our achieved states are within the common tuning range for ocean circulation. The IPCC AR5 WG1 report (Stocker, 2014) shows temperature and salinity ranges in two selected deep ocean grid points, in the North and South Atlantic respectively, of the ensemble of pre-industrial control states ($PIC$) of PMPI2 and CMIP5/PMIP3 (see grid point positions and data in Table 2). We compare those ranges to the corresponding grid cell ranges of our equilibrium states $SE1$–$SE12$. In these selected grid cells, we cover a similar span in salinity and an equally broad range in temperatures as the PMIP-ensemble, though the temperatures in our ensemble are higher (range shifted by 1.5°C). According to Muglia and Schmittner (2015), the PMIP3 $PIC$ AMOC range is 12.6-23.0 Sv (Table 2). If we exclude

the combined simulation with halved wind stress and halved diapycnal (vertical) diffusivity ($WS/2\_DD/2$), which has a very

weak AMOC (2.0 Sv), the AMOC range for our equilibrium states is 8.3-18.0 Sv (Table 2). Thus, there is a difference of 8-9 Sv between highest and lowest value, which is also the case for the PMIP3 $PIC$s, but our ensemble does not cover the two highest PMIP3 AMOC values.

## 3.4    Overturning

In a coarse-resolution model like cGENIE, the overturning circulation, which transports carbon to the deep ocean and back

up to the surface again (Eriksson and Welander, 1956), is one of the most sensitive cirulation components. We diagnose an overturning circulation strength (henceforth denoted $OVT$) by taking the difference between the maximum and minimum of the zonal average overturning streamfunction, $\psi$, below 556 m depth (excluding the uppermost five gridboxes), as shown in Eq. 5. The subscript $gr$ in Eq. 5 denotes the geographical region. $OVT$ is diagnosed for the Atlantic basin and the Pacific basin separately. A global measure of $OVT$ is calculated by taking the difference between the Northern hemisphere maximum and

the Southern hemisphere minimum of $\psi$ below 556 m depth.

$$OVT_{gr} = \psi_{max_{gr}} - \psi_{min_{gr}} \tag{5}$$

The $OVT$ measures the amount of flushing of the deep water, which is important for the carbon storage (Eggleston and Galbraith, 2017). Another important factor influencing the carbon pools is which overturning cell is dominating in terms of volume (Eggleston and Galbraith, 2017); the northern cell (producing North Atlantic Deepwater, NADW) or the southern cell

(producing Antarctic Bottom Water, AABW, and Circumpolar Deep Water, CPDW). Since these water masses are of different origin, they will differ in properties, such as water temperature and nutrients, and will therefore have different capacities for holding $C_{sat}$, $C_{soft}$ and $C_{dis}$. However, the inter-member differences in this aspect are difficult to discern and we have therefore chosen to focus on the impact of $OVT$ described above, which are more clearly identifiable.

## 4    Results

We use a two-step modelling approach, as explained in Section 3.2; the first step is where we change the ocean circulation by changing physical parameters as listed in Table 1, and the second is where we force the biological pump to become 100 % efficient. The results from this two-step approach are described in Sections 4.1 and 4.2.

## 4.1    Step 1 - The effects of ocean circulation changes

In this first step, we achieve a set of pre-industrial equilibrium states — all with pre-industrial $pCO_2^{atm}$ of 278 ppm — where,

as a result of ocean circulation differences, the ensemble members have different carbon reservoirs.

In $PIES278$ and $SE1$–$SE12$, global average salinity, total alkalinity and PO$_4$ are 34.90, 2363 $\mu mol kg^{-1}$ and 2.15 $\mu mol kg^{-1}$ respectively. These properties are conserved, but redistributed, which gives some very small differences in the

global averages between ensemble members. Global average temperature, $T_{avg} = 3.58$°C in $PIES278$, which is close to the modern day observational estimate 3.49°C (Locarnini et al., 2013)). In the $SEs$, $T_{avg}$ ranges between 2.31–4.91°C. Sea-ice cover ranges from 0.0–10.6% in the $SEs$, with $PIES278$ in the centre of the interval (c.f. observational estimates of sea-ice cover, which range between 3-6% due to seasonal variability (Comiso, 2008)). Diagnostic variables for all the individual $SEs$ are given in (Table 3).

The control $PIES278$ model total carbon inventory, $TC$, is $3 \cdot 10^{18}$ mol ($\sim 36,000$ Pg C). $\Delta TC$, and the contributions to $\Delta TC$ by the biological and the solubility pump and by $C_{res}$ (Eq. (4)) in the $SEs$ — which are effects of the changes in ocean circulation — are of the magnitude of a few percent ($\sim 10^{16}$ mol, Fig. 2a–d). This corresponds to a range in $pH_{avg}$ between 7.80 and 7.97 (Table 3), while $pH_{surf}$ stays close to the observational estimate ($\sim 8.2$, see Raven et al., 2005). For reference, Fig. 2e–h show the approximate differences in $pCO_2^{atm}$ that would have occurred in the $SEs$, if we were not restoring it to 278 ppm (see Section 3.2 and Appendix C). In this ensemble, these differences in $pCO_2^{atm}$ stay between $\pm 50$ ppm (Fig. 2e), because of cancellation between the effects of the different carbon pumps. Note that the methods of Appendix C Fig. 2e-h are based on the assumption that changes in $pCO_2^{atm}$ are small and are therefore less reliable for $SEs$ with large changes, e.g. $WSx2$ and $WS/2\_DD/2$. Because the computed changes in $pCO_2^{atm}$ are approximate, the sum of the contributions from the different pumps do not exactly equal the $\Delta pCO_2^{atm}$ corresponding to $\Delta TC$.

### 4.1.1  Sensitivity of overturning streamfunction

The range of $OVT$ (see Eq. 5) resulting from the differences in zonal average overturning stream function ($\psi$, Table S1), is 10.1–32.6 Sv for the global measure (Table 3). The range of Atlantic $OVT$ is 4.1–19.5 Sv (see Figs. 3a and 4a). Note that Atlantic $OVT$ is not the same as $AMOC$ strength, which ranges between 2.0-18.0 Sv for the $SEs$ (Table 2), though the $AMOC$ is a large part of the $OVT$ in the Atlantic. The zonal average overturning streamfunction, $\psi$, for a typical weak ($DD/2$ with $OVT = 17.4$ Sv) and strong ($DDx2$ with $OVT = 32.6$ Sv) circulation $SE$ in Fig. 5a-b show that the overturning circulation patterns in comparison to the control $PIES278$ ($OVT = 20.4$ Sv) are similar, with no major changes in extent of the overturning cells.

Figure 3 suggests a linear relationship between $OVT$ (Eq. 5) and the total carbon inventory $TC$, in this case represented by $\Delta TC$, in the $SEs$. The relationship is clearer for the Atlantic and the global measure (Fig. 3a and 3c) than for the Pacific (Fig. 3b). The correlation coefficients indicate that as much as 90 % of the variance in $TC$ can be explained by changes in $OVT$ (Table 4). For the different carbon pumps, the correlation with $OVT$ is most clear for $C_{soft}$ (see Fig. 4), where 75–80 % of the variance can be explained by the $OVT$ (Table 4). Note that since no biogeochemical manipulations have been made in this step, the remaining variance (20–25 %) is also due to physical perturbations. For $C_{sat}$ and $C_{res}$, the correlation is weak (40–57 % and 25–30 % respectively, Table 4), but statistically significant. If we add $C_{soft}$ and $C_{res}$ and correlate with the $OVT$, the correlation gets stronger than it is for $C_{soft}$ alone. In experiments with stronger $OVT$ (global and/or basin scale), the relative importance of $\Delta C_{soft}$ (negative, see Table 4), $\Delta C_{sat}$ (positive for total, negative for temperature contribution and positive for alkalinity contribution) and $\Delta C_{res}$ (negative) will determine the sign of $\Delta TC$. In our ensemble, stronger $OVT$ (global and basin scale) leads to a decreased storage of carbon in the ocean compared to $PIES278$ (Figure 3).

#### 4.1.2 Sensitivity of total carbon inventory

Although the total carbon inventory is strongly correlated with $OVT$ (Fig. 3), the relative contributions of the carbon pumps to $\Delta TC$ are very different depending on the modified physical characteristics (Fig. 2). They combine in a way to make $\Delta TC$ be more closely correlated with $OVT$ than any of the individual components (Table 4). Altered wind stress ($WS$) intensity ($SE$s 1, 2 and 9), and diapycnal diffusivity ($DD$, $SE$s 5, 9, 10), has a large impact on $TC$, dominated by changes in $C_{soft}$. In experiments with large $\Delta C_{soft}$ (e.g. $SE$s 1, 2 9, 10), $\Delta C_{carb}$ can be as significant as $\Delta C_{sat}$ and $\Delta C_{res}$, whereas it in

most other experiments plays a minor role. The overall impact on $TC$ from changing isopycnal diffusivity ($IDx2$ and $ID/2$ ($SE7--8$) is small, with larger contributions by $C_{sat}$ and $C_{res}$ than from $C_{soft}$. For $SE$s 3, 4 and 10, in which the modified atmospheric heat diffusivity ($AD$) is partly driving the circulation (and temperature) changes, the contributions by changes in $C_{soft}$ and $C_{sat}$ are of similar magnitude and equally important for the change in $TC$. This is also the case for some $SE$s with changes in ocean diffusivity ($ID/2$ and $DD/2\_IDx2$). Overall, no contributing terms can be considered negligible relative to

the other terms, though $\Delta C_{sat,A_{pre}}$ (changes in $C_{sat}$ caused by changes in preformed alkalinity, Eq. B1) and $\Delta C_{carb}$ partly cancel each other (Section 4.1.4). The importance of a given term for $\Delta TC$ depends on the mechanism and the origin of the circulation change (Section 4.1.1).

In the following subsections, we analyze each of the contributing terms.

#### 4.1.3 Sensitivity of temperature and saturation carbon

$\Delta C_{sat,T}$ is the contribution to $\Delta C_{sat}$ from changes in water temperature. The global average temperature, $T_{avg}$, of the equilibrium states has a range of 2.3–4.9 °C (see Table 3), resulting in an interval of change in $C_{sat,T}$ of -0.8 $\cdot 10^{16}$ – +0.6 $\cdot 10^{16}$ mol (Fig. 2b, Fig. 6), or -96 – +72 GtC. Note that this includes a restriction on the solubility constants which prevents solubility from increasing with temperatures below 2°C, which weakens the close-to-linear relationship between $T_{avg}$ and $C_{sat,T}$ (Fig. 6). This corresponds to a range in $\Delta pCO_2^{atm}$ of about -7 – +11 ppm (Figs. 2f and S1) when we solve the carbon system

equations (Appendix C). The simplified equation (Eq. C21) suggested by Goodwin et al. (2011) yields results for $\Delta pCO_2$ that in general are 10 – 20 % lower compared to using the carbon system equation solver (Fig. S1). Changes in $\overline{C_{sat}}$ caused by changes in preformed alkalinity ($\Delta\overline{C_{sat,A_{pre}}}$) span -1.9 $\cdot 10^{16}$ – 1.8 $\cdot 10^{16}$ (Fig. 2b), which roughly corresponds to a range in $\Delta pCO_2^{atm}$ of -21 – +21 ppm (Fig. 2f).

Fig. 2a–c stresses that, in some $SE$s the contribution by $\Delta C_{sat,T}$ to $\Delta TC$, and hence to changes in $pCO_2^{atm}$, is nearly as

important as (e.g. $ADx2$, $ADx2\_DDx2$, $DD/2\_IDx2$, $ID/2$) or even more important than ($IDx2$) the changes in $C_{soft}$. Further, the temperature restriction on the $CO_2$ dissociation constants (see Appendix B1) is likely to cause an underestimation of the effect of $\Delta C_{sat}$ by on average 56 $\pm$67% in our results.

In the ensemble, simulations with a weaker $OVT$ than $PIES278$ tend to have a lower $T_{avg}$ (thus larger $C_{sat,T}$) and a larger ocean storage of $TC$ ($SE$s 4, 6, 7, 9 and 12 in Table 3 and in Fig. 3). The correlation between $C_{sat,T}$ and $OVT$ (see Table 4) is -0.63 globally. The temperature sections in Fig. 7 show that when increased mixing is achieved by higher diapycnal diffusivity ($DDx2$), this causes the stratification to be less sharp between warm surface water and cold deepwater, because the warmer

waters are mixed deeper, and $T_{avg}$ increases while $C_{sat,T}$ decreases. The colder, and more stratified simulation ($DD/2$) is

also the simulation with the weaker $OVT$ (Fig. 5b, Fig. 7b). Note: the Pacific Ocean deep water is warmer in $DD/2$ than in

$DDx2$, but the shallower thermocline has a compensating effect and the net result is a decreased $T_{avg}$ (Fig. S2).

### 4.1.4   Sensitivity of biogenic carbon

($C_{soft} + C_{carb}$) underestimates $C_{reg}$, c.f. the model tracer ($DIC_{reg} = DIC - DIC_{pre}$), by 1-3%. Because this error is consistent, the contribution to $\Delta C_{res}$ (Section 4.1.5) is negligible. Changes in $C_{soft}$ and $C_{carb}$ always have the same sign, but the

effect of $C_{carb}$ has a smaller magnitude (excepted $IDx2$, Fig. 2 c). The effects of $\Delta C_{soft}$ and $\Delta C_{carb}$ on $pCO_2^{atm}$ are of the

same sign (Fig. 2e and g). However, the change caused by $\Delta C_{carb}$ is associated with a $\Delta C_{sat,A}$ of opposite sign and double

magnitude. Thus, the net effect of the biogenic hard tissue pump (the carbonate counter pump) on $pCO_2^{atm}$ is of opposite sign

compared to that of the soft tissue pump (Fig. 2e and f), as expected due to the effect of the carbonate pump on preformed

alkalinity (Goodwin et al., 2008). Globally, 75–80 % of the variance in $C_{soft}$ is explained by the $OVT$ (Fig. 4, Table 4). This

correlation is strongest over the Atlantic basin.

$\overline{P^*}$ in the $SE$s ranges between 0.32–0.59 (Table 3), while $\overline{P^*} = 0.43$ in $PIES278$. For reference, the observational estimate

for $\overline{P^*}$ of the modern ocean is 0.36 (Ito and Follows, 2005). The tight, linear relationship between the inventory of $C_{soft}$ and

$\overline{P^*}$ (Fig. 8b) is expected from the definition of $\overline{P^*}$ (Eqs. A1 and B3), since it is a direct measure of the efficiency of the

biological pump. The relation between $TC$ and $\overline{P^*}$ (Fig. 8 a) is also close to linear and appears dominated by biogenic carbon

(compare Figs. 8a and 8b). Deviations from a perfect straight line are caused by $\Delta C_{carb}$, $\Delta C_{sat}$ and $\Delta C_{res}$. Their influence

on $\Delta TC$ can be small compared to the influence of biogenic carbon (see Fig. 2, e.g. $SE$s 2, 6, 9) or large but partly cancelling

each other (see Fig. 2, $SE$s 1, 3, 5, 10, 11), thus resulting in biogenic carbon "passing on" its linear relationship with $\overline{P^*}$ to

$\Delta TC$. $\overline{P^*}$ is important for the drawdown potential of a model (Fig. 8c), which is examined in the second step of the modelling

(Section 4.2).

### 4.1.5   Sensitivity of residual and disequilibrium carbon

The global inventory of $C_{res}$ in the $SE$s ranges between $-3.7 \cdot 10^{15} - 2.3 \cdot 10^{16}$ (c.f. global $TC$ inventory of $\sim 3 \cdot 10^{18}$ mol).

The contribution from $\Delta C_{res}$ to $\Delta TC$ ((Fig. 2 d)) is generally of the order $\sim 1 \cdot 10^{15}$, and the inventory of $\Delta C_{res}$ in the $SE$s

changes c.f. $PIES278$ by 0–30 %.

When looking at $\Delta C_{res}$ (Fig. 2 d), we are mainly interested in the contribution by $C_{dis}$. We have previously confirmed

that the contribution to $\Delta C_{res}$ by errors in calculations of biogenic carbon are negligible (Section 4.1.4), and we find that the

$\Delta C_{res}$ given by Eqs. 4, B1 and B5–B6 corresponds closely to $\Delta \overline{DIC_{pre}} - \overline{C_{sat}}$, where $DIC_{pre}$ is preformed carbon from the

model. Thus, $\Delta C_{res}$ is representative of $\Delta C_{dis}$.

The processes influencing $C_{dis}$ will be different in the deepwater formation areas in the Northern and Southern hemispheres

(Toggweiler et al., 2003; Ito and Follows, 2005; Lauderdale et al., 2013). In the deepwater formation areas in the North Atlantic,

$C_{dis}$ is mainly a result of the temperature gradient. If a water parcel cools too fast before it sinks, there is not enough time

to equilibrate with the atmosphere and the result will be a negative $C_{dis}$. This is seen as a negative signal in the NADW in

Figure 9a–c. In a warmer global ocean, the temperature gradient between the equator and the poles is smaller. This makes it easier for a parcel travelling north in the Atlantic to reach equilibrium with the atmosphere before deepwater forms, since the parcel does not have to cool as much as in a colder simulation. In a warmer ocean, there is also less sea ice preventing exchange with the atmosphere. However, a warmer global ocean is often associated with faster circulation (see Section 4.1.1), and if the circulation becomes faster at the same time, the negative effect of the shorter time available for equilibration will compete with the positive effect of a smaller temperature gradient and less sea ice. In some cases, but not all, the warmer high latitude temperatures can compensate for the speed up of the circulation. In the Southern Ocean there will also be less time for the surface water to equilibrate its gas concentrations with the atmosphere. Here, oversaturated deepwater coming back to the surface in this area may not have the time to release its carbon to the atmosphere before the water sinks back into the deep, producing positive $C_{dis}$. This appears as a positive signal in the lower half of the sections in Figure 9c. In a case with faster circulation, the contribution of positive $C_{dis}$ will be even larger. This effect dominates over the temperature gradient effect in this area, because the waters being brought back to the surface are already very cold. However, in this model, the effect of sea ice on $C_{dis}$ can also be very pronounced in this area, if it caps the upwelling area and prevents outgassing. This effect is evident in $AD/2$, which has, by far, both the largest sea ice cover (10.6 %, Table 3) and the largest inventory of $C_{dis}$ ($2.1 \cdot 10^{16}$ mol). The competing effects described above result in significant correlations betwen $C_{res}$ (and similarily $C_{dis}$) and global $OVT$ (Table 4), $T_{avg}$ and sea ice cover (%) of -0.30, -0.65 and 0.50 respectively.

Comparing panels a and c in Figure 9, we see that the differences are difficult to attribute to one single process. The overturning circulation in $WSx2$ is stronger than in $PIES278$. This makes the global ocean warmer, reduces sea ice, but also shortens the time for equilibration with the atmosphere in the North Atlantic branch. In this particular case of stronger circulation, the shorter time for equilibration dominates over the reduced temperature gradient and causes more negative disequilibrium in the North Atlantic deepwater formation area compared to the control case in panel c. In the Southern Ocean, there is less sea ice, which allows more direct contact between the ocean and the atmosphere. However, due to the faster overturning, the deep waters that upwell here will quickly sink again. This is particularly seen in the Atlantic sector, as a band of positive $C_{dis}$ extending from the surface and down, whereas in the control, the positive $C_{dis}$ is more confined to just below the sea ice area.

In $AD/2$ (Fig. 9b), we see a signal of positive $C_{dis}$ originating in the Southern Ocean that is much more pronounced compared to the control $PIES278$ (Fig. 9c). $AD/2$ is by far the coldest state and the global percentage of sea ice cover is doubled compared to $PIES278$ (for reference, in the second coldest state the sea ice cover has increased by less than 40 % compared to $PIES278$, see Table 3). It is likely that the extensive sea ice in $AD/2$, to a larger extent than in the control, prevents the oversaturated deepwater from equilibrating with the atmosphere before sinking again. For $AD/2$, the contribution of $C_{dis}$ to $C_{res}$ is large enough to be critical for the sign of $\Delta TC$ (see $AD/2$ in Figure 2).

## 4.2   Step 2 - Drawdown potential

In this step, we use the set of equilibrium states ($SE$s and the control $PIES278$) from step 1 as initial states for determining the drawdown potential, $DP$ (Fig. 1). This reveals the dependence of the resulting equilibrium state $DE1--DE12$ on differences

in the initial states $SE1–SE12$. The control drawdown equilibrium is denoted $CDE$. $DP$ is computed as the difference in $pCO_2^{atm}$ between 278 ppm and the drawdown equilibrium states.

The $DP$ varies strongly between the ensemble members and is close to linearly related to the biological efficiency, in terms
of $\overline{P^*}$, of the initial equilibrium state (Fig. 8 c). The near linear relationship between $DP$ and $\overline{P^*}$ of the initial $SE$ is expected (see e.g. Marinov et al., 2008a). If the biological efficiency in the $SE$ is small, there is a larger pool of unused nutrients that can be used to capture carbon when biological efficiency is increased to 100 %. In this ensemble, an increase in biological efficiency manifested by an increase in $\overline{P^*}$ of 0.1, corresponds to a drawdown of $CO_2$ from the atmosphere of about 20-30 ppm. This is similar to the theoretical prediction by Ito and Follows (2005) of $\sim 30$ ppm. However, the drawdown of atmospheric $CO_2$
achieved during the drawdown experiments is not purely due to biology. There are also additional effects on $pCO_2^{atm}$ due to changes in ocean temperature (caused by changes in radiative balance), circulation and disequilibrium, and due to the climatic conditions of the initial state. Thus, the model results do not correspond exactly to the theoretical prediction in this case. The most prominent example is $AD/2$, which has a low initial $\overline{P^*}$, but still has a low $DP$. This is the coldest of all initial states, with very high ocean sea ice cover (Table 3) compared to the other $SE$s, and the cold conditions are likely to be affecting the
conditions for biological production and disequilibrium. Note also that the near-linear relationship between $\overline{P^*}$ and $DP$ does not predict $DP$ to be exactly zero for $\overline{P^*} = 1$, as would have to be the case. When all climatic changes caused by the drawdown experiment itself are removed, $DP$ is reduced by $4 – 7$ ppm. This shows that the biological component is highly dominant in the total drawdown. However, the climatic differences between the initial states will still not allow the theoretical prediction to be exact.

Those experiments that have a lower $T_{avg}$ (thus a larger inventory of $C_{sat}$) compared to $PIES278$) tend to have a smaller $DP$ than those with higher temperatures, which is due to circulation changes acting in a predictable way. The circulation change that is causing a colder temperature is also causing the $OVT$ to be weaker (Table 3), and at the same time causing a more efficient biological pump (more $C_{soft}$) (Fig. 4), because there is more time for biology to take up nutrients. Hence, there is less preformed nutrients left at the surface, which means $\overline{P^*}$ is higher and the $DP$ is smaller.

## 5 Discussion

### 5.1 Solubility pump and disequilibrium

The effect on $pCO_2^{atm}$ of a change in the solubility pump is approximately quantifyable from the change in global ocean average temperature, $\Delta T_{avg}$, between two simulations, as described by Eq. (C21) suggested by Goodwin et al. (2011). According to this equation, the set of observed $\Delta T_{avg}$ of our ensemble would correspond to 5.9 $\pm$5.0 ppm °C$^{-1}$ (Fig. S1). Solving the
carbon system for the same set of $\Delta T_{avg}$ yields 7.3 $\pm$6.0 ppm °C$^{-1}$ (Fig. S1). Here, the deviations from the straight line are caused by the temperature restriction on the solubility equation (Appendix B1). The error in using the simplified equation seems to be on the order of 20 %. Depending on which process is causing the change in ocean circulation, the impact of changes in the solubility pump on $pCO_2^{atm}$ can be almost as important as the impact of changes in the biological $CO_2$ efficiency of carbon uptake. For changes in ocean isopycnal diffusivity, the solubility pump effect is even the dominant response. Due to the

temperature restriction on the $CO_2$ solubility constants, the effect of $\Delta C_{sat,T}$ is likely to be underestimated by on average 56 $\pm 67\%$ in our results, further emphasising its importance. In previous studies, this has to some extent been disregarded, when the response of the biological pump has been assumed to be the dominant response to the applied changes in circulation (e.g., Archer et al., 2000a; Kwon et al., 2011).

The relationship between the changes in $T_{avg}$ and $\Delta C_{sat,T}$ is close to linear (Fig. 6). Any deviation from a perfect, straight line is again caused by the temperature restriction on the solubility equation (Appendix B1). The slope of the line is $\overline{\frac{\partial C_{sat}}{\partial T}} = -0.59 \cdot 10^{16}$ mol °C$^{-1}$. If the global ocean is cooled by $\sim 2.6$ °C, as expected in a glacial state (Headly and Severinghaus, 2007), the slope of the line suggests the excursion in $C_{sat}$ would be $\sim 1.5 \cdot 10^{16}$ mol. Note that this excursion is underestimated due to the restriction on the solubility equation. For a 2.6 °C cooling, the carbon system equations (see Appendix C) yield a corresponding decrease of about 30 ppm in $pCO_2^{atm}$. Here, we cannot use the simplified Eq. (C21), because the buffered carbon inventory is unknown in this hypothetical case.

In a set of idealised GCM simulations, Eggleston and Galbraith (2017) show that $CO_2$ disequilibrium can be as important as $C_{soft}$. In our ensemble, the effects of $C_{dis}$ appear to be particularily important in simulations with a lot of sea ice (e.g. $AD/2$, see Fig. 2d,h). This leads us to the conclusion that it may also be of importance in glacial simulations. A caveat to this finding is cGENIE's coarse resolution at high latitudes, and its simplified representation of sea ice as a complete barrier to gas exchange. Assuming that $\Delta C_{res}$ is mainly due to $\Delta C_{dis}$, the circulation changes we impose correspond to a change in $pCO_2^{atm}$ of $\sim -30 - -+10$ ppm due to $\Delta C_{dis}$. This is comparable to the results of Marinov et al. (2008a), while Eggleston and Galbraith (2017) suggest as much as $\sim 40 - -70$ ppm drawdown of $CO_2$ due to increased $C_{dis}$ in glacial like simulations. This emphasises the need for further studies on the role of $C_{dis}$ in glacial as well as other climate scenarios with changes in ocean cirulation.

### 5.2 Implications of changes in $OVT$ in relation to changes in carbon

In experiments with stronger $OVT$ (global and basin scale) than in $PIES278$, e.g. $DDx2$ (Figs. 5a and 7a), a water parcel will, on average, stay near the ocean surface for a shorter time (e.g $DD/2$, see Figs. 5b and 7b). Hence, biology will have less time to use the available nutrients and this will give less $C_{soft}$, thus $\Delta C_{soft}$ has a strong negative correlation with $OVT$ (Fig. 4, Table 4). Changes in $OVT$ explain 75-80% of the variance in $C_{soft}$ in the $SE$ ensemble, while the rest of the variance is likely explained by e.g. redistribution of nutrients, light limitation regions or similar. Meanwhile, with stronger $OVT$ there will be more mixing, leading to a deepening of the thermocline, and $T_{avg}$ will increase (Fig. 7a, Tables 3 and 4). Thus, the correlation with $\Delta C_{sat,T}$ will also be negative, though weaker (43–64 %). The response of $\Delta C_{dis}$ is more difficult to predict from the $OVT$, due to competing changes in the temperature gradient (especially important in the North Atlantic), sea ice and outgassing (dominant in the Southern Ocean) resulting from a change in $OVT$. Interestingly, the total carbon inventory $TC$ shows a closer correlation with the $OVT$ than any of the individual ocean carbon components (Figs. 3 and 4, Table 4). Thus, the inventories of the individual DIC components are affected by the choice of model tuning strategy, whereas the total carbon inventory is mainly a result of the strength of the circulation i.e. the ventilation of the deepwater. Similar results are found for

$C_{soft}$ and $C_{dis}$ by Eggleston and Galbraith (2017), who describe the changes in ocean ventilation using an ideal age tracer. Here, we show that the correlation with ocean ventilation is even stronger for total carbon and is also present for $C_{sat}$.

### 5.3 Implications for model validation

When comparing model studies, it is important to recognize differences in biological efficiency in their control states. The pre-industrial $\overline{P^*}$ of a model will determine its pre-industrial inventory of $C_{soft}$ but also its drawdown potential. If the pre-industrial $\overline{P^*}$ is incorrect, the total carbon inventory in the model will adjust to compensate this error, in order to achieve equilibirum with pre-industrial $pCO_2^{atm}$. Hence, failing to tune the models for pre-industrial $\overline{P^*}$ will mean that they start from a non-representative state of the carbon system. Thus, models with different initial $\overline{P^*}$ will have different $\Delta pCO_2$ in response

to similar circulation changes. This point was mentioned in Marinov et al. (2008a) and emphasised by Duteil et al. (2012), but does not seem to have been recognised in the model inter-comparison community and, still, models are not tuned for $\overline{P^*}$. The range of $\overline{P^*}$ in our pre-industrial ensemble ($SE1$–$SE12$) is 0.32–0.59. This range includes the current estimate for the global ocean, which according to Ito and Follows (2005) is 0.36. Our range in initial state $\overline{P^*}$ corresponds to a range in drawdown potential of 94–139 ppm. While using a different model, but a similar approach, we confirm the conclusion of Marinov et al.

(2008a) and want to stress the importance of a similar initial efficiency of the biological pump in model inter-comparison studies where $CO_2$ drawdown is diagnosed.

Few studies have simultaneously diagnosed the individual contributions by the solubility and biological pumps and the effect of surface $CO_2$ disequilibrium. Studies by Ito and Follows (2013), Lauderdale et al. (2013), Bernardello et al. (2014) and Eggleston and Galbraith (2017) use a similar separation of the carbon storage processes as we do. For increases in wind

stress, the sign of $\Delta TC$ (and thus of $\Delta pCO_2^{atm}$) and the individual contributions by the carbon pumps and $C_{dis}$ agree with those found by Lauderdale et al. (2013). In our ensemble, $\Delta TC$ does not fully reveal the magnitude of the differences in the individual carbon pumps between ensemble members, because the changes to the individual pumps tend to be partially compensating each other. We show that the differences in equilibrium state efficiency of the biological pump between $SE$s manifest themselves as as differences in model sensitivity to the perturbation in biological pump efficiency, as predicted by Ito

and Follows (2013). Our result can be important for future model inter-comparison studies, in explanations of results, but also in planning for common tuning strategies and experimental design. Compared to the scenario-specific results of Bernardello et al. (2014), our results could be used more generally as a way of anticipating the model behaviour, based on in which way the ocean circulation changes in a model study. Depending on in which way we have changed the ocean forcing, and what the resulting effect on ocean circulation is, the origin of the change in ocean carbon storage is different. When wind stress

($WS$) or ocean diapycnal diffusivity ($DD$) is changed, the response of the biological pump gives the most important effect on ocean carbon storage, whereas when atmospheric heat diffusivity ($AD$) or ocean isopycnal diffusivity ($ID$) is changed, the solubility pump and the disequilibrium component are also important and sometimes dominant. Our results give a first approximation of the effect of these ocean circulation changes on the ocean carbon storage, but it is important to keep in mind that the results of changes in individual parameters do not always combine linearly. For example, with doubled atmospheric heat diffusivity ($ADx2$) and doubled ocean diapycnal diffusivity ($DDx2$), the response of the solubility pump is very similar

in both simulations. In the combined simulation ($ADx2\_DDx2$), the response of the solubility pump is larger, but far from doubled, and the response of the soft tissue pump is smaller than in $DDx2$ alone. In this particular case, it is difficult to discern which of these two parameters has the strongest impact on the system.

The four different preformed model tracers ($P_{pre}$, $DIC_{pre}$, $O_{2pre}$, $A_{pre}$) are shown to be useful for accurate determination of the initial state carbon partitioning and nutrient utilisation efficiency, of which we demonstrate the importance for model drawdown potential. They eliminate errors associated with indirect methods used to determine $AOU$ and $A_{pre}$ (as described by e.g., Ito et al., 2004; Bernardello et al., 2014; Eggleston and Galbraith, 2017), and facilitate error estimates for the carbon partitioning methods. Some useful applications of preformed tracers have previously been presented by e.g. Marinov et al.

(2008b), Bernardello et al. (2014) and Eggleston and Galbraith (2017), but such extensive use of preformed tracers is, to our knowledge, unprecedented in studies of the ocean carbon system.

## 5.4   Implications for glacial studies

We have shown that, when comparing model simulations with the same $pCO_2^{atm}$, but with differences in ocean circulation and overturning circulation strength ($OVT$, see Section 3.4), the compared simulations will have different carbon inventories, and

different strengths of the ocean carbon pumps. In the PMIP3 inter-comparison project, where glacial simulations with different models are compared, the models are forced with glacial $pCO_2^{atm}$ to achieve the LGM state (https://pmip3.lsce.ipsl.fr/). The ocean circulation state is however not specified. Otto-Bliesner et al. (2007) showed that model simulations in the PMIP2 project developed very different LGM ocean circulation patterns and specifically large differences in AMOC strength, despite displaying similar ocean circulation patterns in pre-industrial simulations. Most models had been initiated with pre-industrial

circulation and LGM boundary conditions according to the PMIP2 protocol. When run to quasi-equilibrium, some models would develop an LGM-like circulation, with a shallower boundary between NADW and AABW than today, as indicated by paleonutrient tracers (see e.g., Marchitto and Broecker, 2006), and some would keep a more pre-industrial like circulation. Since the ocean circulation patterns differ, the ocean carbon storage and thus the model carbon inventories of the compared PMIP-simulations also differ. This will be important when comparing e.g. deglacial scenarios run with these different models

(e.g., Zhang et al., 2013).

When attempting to simulate the glacial $CO_2$ drawdown, it is crucial to critically evaluate the changes in forcing that need to be applied to achieve a glacial state in the model. Do these changes agree with what we believe actually happened in the climate system during a glacial? When applying PMIP3 boundary conditions for the LGM, the height of the ice sheet in the northern hemisphere will tend to intensify both the wind stress over the North Atlantic basin and as a result the AMOC-

circulation, (Muglia and Schmittner, 2015; Klockmann et al., 2016). In a full glacial state, the associated deepening of the AMOC is however counteracted by the decrease in $pCO_2^{atm}$ (Klockmann et al., 2016). Similar effects on the wind fields due to the Laurentide ice sheet are seen in e.g. Löfverström et al. (2014). Sime et al. (2013) and Sime et al. (2016) suggest that Southern Hemisphere winds will also be stronger when applying LGM boundary conditions, though they emphasise that results from different palaeoproxies and models disagree on this. In our simulation, we intensify the wind stress in both hemispheres

and this leads to decreased capacity of both the biological and the solubility pump, and effectively an increase in $pCO_2^{atm}$.

Lauderdale et al. (2013) showed similar results, but for increased Southern Ocean winds. Hence, the changes in wind fields achieved by the applied LGM boundary conditions in models may be contributing to the difficulties in simulating the glacial decrease in $pCO_2^{atm}$.

In those of our ensemble members where ocean diapycnal (i.e. near vertical) diffusivity is halved, we achieve some glacial-like ocean characteristics; the circulation is weaker, the global ocean temperature is colder and the biological pump is stronger. However, it has been shown by Schmittner et al. (2015) that open ocean mixing is likely to have been intensified during glacials, when lower sea level made shelf areas decrease and tidal mixing was shifted to the deep ocean. In their model, global ocean mean vertical diffusitivy increased by more than a factor of 3, leading to an intensification of ocean overturning. In our experiments, a doubling of diapycnal diffusivity leads to a decrease in ocean carbon storage corresponding to an increase of $pCO_2^{atm}$ of more than 20 ppm (see $DDx2$ in Figure 2e). Hence, in a full glacial scenario, processes causing increased ocean carbon storage would have to offset this effect before causing any net decrease in $pCO_2^{atm}$. Other effects on glacial $pCO_2^{atm}$ linked to lower sea level (reduced ocean volme) during glacials, caused by higher salinity, and higher concentration of DIC, alkalinity and nutrients, have been constrained to +12–16 ppm (Köhler and Fischer, 2006; Brovkin et al., 2007; Kohfeld and Ridgwell, 2009). In this process study we are not aiming to reproduce LGM conditions in the model and such effects of changes in ocean volume are beyond the scope of our investigations. Ocean volume, and global averages of salinity, alkalinity and phosphate have thus been kept constant in our simulations.

Since numerous studies of proxy data indicate that the global ocean was in fact less ventilated during glacials (e.g., Broecker et al., 1990; Sikes et al., 2000; Keigwin and Schlegel, 2002; Skinner et al., 2010, 2015), it seems possible that the effect of increased mixing was indeed offset by some other process. One such factor could be that the global ocean was saltier and more stratified (Ballarotta et al., 2014). In our simulations, weaker overturning circulation is also connected to colder temperatures. These cold simulations show a tendency towards lower drawdown potentials. It is likely that the more sluggish circulation is already allowing a more efficient biological pump, leading to a higher $\overline{P^*}$ and thus a smaller drawdown potential. Another important mechanism for the global glacial deep ocean circulation is surface buoyancy loss around Antarctica driven by the brine rejection associated with sea ice formation (Klockmann et al., 2016; Marzocchi and Jansen, 2017). This effect is not explored in our simulations.

According to Headly and Severinghaus (2007), the global average temperature of the glacial ocean was $2.6 \pm 0.6$ °C colder than the modern day ocean. Since the ensemble member with the coldest ocean is only 1.3 °C colder than $PIES278$, the variations in $C_{sat,T}$ during the past glacial cycles were likely larger than in our set of experiments, allowing for a larger contribution by the solubility pump. Hence, the colder temperatures could play an important role in offsetting the effect of increased mixing. Fig. 7 suggests that $\Delta C_{sat,T}$ for a change in $T_{avg}$ of -2.6°C c.f. pre-industrial would be approximately $1.5 \cdot 10^{16}$ mol (180 GtC), whereas $\Delta C_{sat,T}$ for the coldest of our simulations is only $0.58 \cdot 10^{16}$ mol (70 GtC). If we account for a likely underestimation of $\Delta C_{sat,T}$ of $\sim 50$ % (see Section 4.1.3 and Appendix B1) in Fig. 6, a simulation as cold as the LGM state suggested by Headly and Severinghaus (2007) would have an increase in strength of the solubility pump corresponding to $\sim 300$ GtC.

In our ensemble of simulations, 100 % nutrient utilisation efficiency ($NUE$) causes more drawdown than is necessary to reach glacial values. Future efforts need to deduce how big an increase in $NUE$ we could expect for a glacial, when using proxy data of e.g. iron fertilisation (e.g., Petit et al., 1999) and water mass properties (e.g., Elderfield and Rickaby, 2000) as a constraint. By understanding how ocean circulation changes during the glacials may have contributed to altering the ocean $NUE$, it will be easier to quantify how much it may have increased due to e.g. fertilisation by deposition of iron from dust. However, on-going studies indicate that there may have been more, not less, preformed nutrients in the deep ocean during the last glacial (Homola et al., 2015, and Spivack, A. J. (P.C., 2015)), which implies less efficient nutrient utilisation by biology. One aspect that could explain how more carbon could still be trapped by biology in such a case is if the stoichiometric ratios in a glacial scenario no longer follow the averages described by Redfield (1963). In most climate models, this is currently not taken into account. Implementation of variable stoichiometry in models could bring interesting insights in the future.

## 6   Conclusions

In this paper, we have studied three mechanisms for ocean carbon storage — the biological pump, the solubility pump and the contribution from air-sea $CO_2$ disequilibrium — and quantified the response of these mechanisms to differences in the equilibrium ocean circulation state. For a given set of equilibrium states in the model cGENIE, we have constrained the response of the carbon storage associated with the first two mechanisms reasonably well and diagnosed their influence on $pCO_2^{atm}$. We have also seen some response related to ocean $CO_2$ disequilibrium.

We have obtained different states of equilibrium ocean circulation by varying forcings and model parameters (listed in Table 1) in a model ensemble, while keeping atmospheric $CO_2$ constant. This was not done with the aim to achieve a glacial-like circulation, but to study how the ocean carbon storage responds to changes in a wide range of circulation processes and relate the response of the three mechanisms for ocean carbon storage to differences in ocean circulation strength. The contributions to the change in carbon storage by the solubility pump, the biological pump or $CO_2$ disequilibrium are different depending on the origin of the ocean circulation change, i.e. the model tuning strategy. When wind stress or ocean diapycnal diffusivity is changed, the response of the biological pump has the strongest impact on ocean carbon storage. In contrast, when atmospheric heat diffusivity or ocean isopycnal diffusivity is changed, the solubility pump and the disequilibrium component also give important, and sometimes dominant, contributions to the change in ocean carbon storage. Despite this complexity, we obtain a negative linear relationship between total ocean carbon and the combined strength of the northern and southern overturning cells. This relationship is robust to different reservoirs dominating the response to different forcing mechanisms. We show that the individual carbon components are all to some extent correlated with the strength of the circulation, but they combine in a way that makes the response in total carbon inventory be nearly fully explained by changes in circulation strength.

Finally, to constrain the biological pump, we used the $SE$ ensemble members (Fig. 1) as initial states to see how their $pCO_2^{atm}$ responded when the model was forced into a state with 100 % efficient biology. We applied similar adjustments to circulation parameters as those tested in Marinov et al. (2008b), in order to allow some direct comparison of our results with their study. In agreement with Marinov et al. (2008b), we find that the drawdown potential of an ensemble member is

a direct result of its biological efficiency, as measured by the ratio between global average regenerated ($\overline{P_{reg}}$) and total ($\overline{P}$) phosphorus, denoted $\overline{P^*}$, in its initial equilibrium state. We show that it is possible to quantify, from theory, the effect of biases in the carbon inventory of a model's control state on its sensitivity to changes in the biological pump. We test a wide range of changes to the forcing in order to demonstrate that the result is robust. This result should be of value in understanding the biases of individual models, in model inter-comparison studies, and potentially for choosing tuning criteria. Often, a model with stronger circulation will have a higher global ocean mean temperature, thus a weaker solubility pump, and lower biological efficiency, thus also a weaker biological pump. Such a model has a smaller ocean carbon inventory in the control state, but a larger drawdown potential for $CO_2$, compared to a model with weaker circulation. Hence, when different models are used to simulate a glacial scenario, it is likely that a significant part of the difference in their $CO_2$ drawdown potentials results from differences that are already present, but not directly visible, in their control states.

*Code and data availability.* The source code for cGENIE is publically available at http://www.seao2.info/mycgenie.html. Data is available upon request (e-mail to the corresponding author).

## Appendix A: The ocean carbon system

### A1 The ocean carbon pumps

The abiotic, physical pathway, or *the solubility pump*, begins with air-sea gas exchange, which acts to achieve a chemical equilibrium between the atmosphere and the surface ocean. This equilibrium depends on temperature. The ocean circulation then advects both the temperature and the dissolved carbon into the deep ocean. This carbon is also referred to as *preformed* carbon. Since the solubility of $CO_2$ is larger in colder water, the sinking cold water is enriched in carbon compared to surface waters in warmer regions.

The biological pathway, or *the biological pump*, begins with biological production in the surface ocean. Carbon is incorporated into soft-tissue organic compounds. Some of this material then reaches the deep ocean, either by being advected in currents or by falling out of the surface layer. When the organic material is decomposed, inorganic carbon ($CO_2$) comes back into dissolution in the water. This fraction of DIC is referred to as *regenerated* carbon. Carbon is also incorporated into hard tissue (shells) in the form of $CaCO_3$ which can be dissolved in the deep ocean. This dissolution influences deep ocean alkalinity (Section 2).

Due to the difference in the chemical role of soft-tissue and hard-tissue carbon, the biological pump is more correctly referred to as being two separate pumps; the soft-tissue pump and the carbonate (hard-tissue) pump (e.g., Sarmiento and Gruber, 2006; Kohfeld and Ridgwell, 2009). The soft-tissue pump acts to increase deep ocean DIC. The carbonate pump has a counter effect, because uptake of $CaCO_3$ for shell formation reduces alkalinity and the capacity to dissolve $CO_2$ in the surface ocean. The soft-tissue pump is stronger than the carbonate pump. Therefore, the net effect of the biological pump is to enhance the deep ocean concentration of DIC.

## A2  Nutrient utilisation efficiency and $P$*

Biology in the surface ocean uses a fraction of the available nutrients to produce new organic material and binds $CO_2$ in the process. The remaining, unused (or preformed), nutrients, denoted $P_{pre}$, are brought with the circulation (the physical pathway) into the deep ocean, where no new production is possible. The nutrients transported to the deep ocean via the biological pump are called regenerated nutrients, $P_{reg}$. If the biology becomes more efficient at using the nutrients in the surface ocean, the fraction of $P_{reg}$ in the deep ocean increases and $P_{pre}$ decreases.

As described in the framework introduced by Ito and Follows (2005), the global average of $P_{reg}$ relative to the overall global average concentration of inorganic nutrients (denoted by $P$) is a measure of nutrient utilisation efficiency, $NUE$. This can be described using the parameter $P^*$,

$$\overline{P^*} = \frac{\overline{P_{reg}}}{\overline{P}}. \tag{A1}$$

Here, the overbars mark that we are using the global average of a quantity. If $\overline{P^*}$ is 1, all available nutrients in the deep ocean were brought there by the biological pump. In other words, the deep ocean is ventilated by surface waters that have had all nutrients removed, hence (at steady state,) the ocean interior will have no $P_{pre}$.

To calculate $\overline{P^*}$ (Eq. A1), we need to know $\overline{P_{reg}}$, the concentration of dissolved phosphate in the deep ocean that is of regenerated origin. We get $P_{reg}$ by removing $P_{pre}$ from the concentration of total phosphate, $P$.

## Appendix B:  Separation of carbon species

In this section, the calculations of the total carbon inventory described by Eq. 3 are described in detail.

The atmospheric carbon content, which in this model is limited to its content of $CO_2$, is given by the partial pressure of $CO_2$ times the number of moles of gas in the atmosphere, $M_a$. Assuming an atmospheric thickness of 7,777 m, $M_a$ is given to $1.7692 \cdot 10^{20}$ mol. In this case 1 ppm of $CO_2$ corresponds to 2.123 PgC, which is consistent with the OCMIP recommendation. $M_o$ is the mass of the ocean [kg], which in our ensemble of simulations is kept constant at $1.34 \cdot 10^{21}$ kg. Implications of changes in volume are further discussed in Section 5.4. Calculations of $C_{sat}$, $C_{soft}$ and $C_{carb}$ are described in Sections B1-B2. $C_{res}$ is the residual between the sum of the calculated $C_{sat}$, $C_{soft}$ and $C_{carb}$ and the actual carbon concentration in the water parcel, which includes DIC and organic matter ($POC$ and $DOC$). $C_{res}$ contains three components; 1) The first, and most interesting, contribution to $C_{res}$ is the disequilibrium component $C_{dis}$. This is the part of the carbon concentration which results from the water parcel not being in perfect equilibrium with the atmosphere at the time when it left the surface. Hence, the concentration of carbon of abiotic origin (preformed carbon) in the water parcel consists of $C_{sat} + C_{dis}$. We can therefore use the relation $C_{dis} = C_{pre} - C_{sat}$ to estimate this component of $C_{res}$, though we need to keep in mind that it there are calculation errors associated with $C_{sat}$ that will affect this estimate (see point 3 below). 2) The second contribution to $C_{res}$ is the presence of carbon in the form of particulate and dissolved organic matter. At any one model time step, the concentration of such carbon is very small compared to the other terms in the equation (<1 %, $\sim 1 \cdot 10^{15}$ mol) and this is therefore not considered separately.

It is however included when model $TC$ is quantified. 3) The third contribution to $C_{res}$ consists of the errors associated with any imperfect assumptions in the theory used for calculating $C_{sat}$, $C_{soft}$ and $C_{carb}$. These are further discussed in Sections B1-B2.

## B1 Contribution by the solubility pump

For one individual water parcel, $C_{sat}$ corresponds to the concentration [mol kg$^{-1}$] of carbon the water parcel would have had if it would have been in equilibrium with the atmosphere, taking into account its temperature, salinity, alkalinity and also the minor effect of the concentration of $PO_4$ in the absence of biology. We calculate the global average of grid cell $C_{sat}(T_{SE_n}) - C_{sat}(T_{PIES278})$ solving the carbon system equations using the solver CO2SYS (Lewis et al., 1998). The constants used in the scheme do not exactly match those used in the model, but the differences are minor.

The dissociation constants used in the cGENIE calculations of solubility for $CO_2$ in sea water follow Mehrbach et al. (1973), which are only defined for waters between 2–35 °C. Hence, the expression for $CO_2$ solubility in the model is restricted so that all water below 2 °C has the same $CO_2$ solubility (similarily for all water above 35°C). In the calculations of $C_{sat}$, we use CO2SYS with this temperature restriction, to accurately represent the model behaviour. In order to estimate the error introduced by this restriction, we need to assume that the same dissociation constants can be used outside the given temperature interval.

The validity of this assumption is supported by the results of Goyet and Poisson (1989), who find similar dissociation constants for the interval -1–40°C in a study on artificial seawater. When CO2SYS is run using model ocean temperatures without the temperature restriction, we find that the calculated inventory of $C_{sat}$ in the $SE$s is 0.06-0.5% larger than with the restriction. For $PIES278$ the inventory of $C_{sat}$ is 0.25% larger. In terms of $\Delta C_{sat}$, the unrestricted $C_{sat}$ inventories indicate that the contribution by temperature changes to $\Delta TC$ is on average underestimated by approximately 56 $\pm$67% when the restriction

is active. Since the restriction is used consistently, the error caused by the restriction being present in the model should not constitute a significant problem for our analysis. Nonetheless, the underestimation of the effect of temperature changes should be heeded in the discussion of our results.

Since $pCO_2^{atm}$ is constant and the global ocean mean salinity is similar in all ensemble members $SE1-SE12$, any changes in global average $C_{sat}$ between $PIES278$ and the $SE$s will be due to changes in ocean temperature or alkalinity:

$$\overline{\Delta C_{sat}} = \overline{\Delta T \frac{\partial C_{sat}}{\partial T}\Big|_{pCO_2,S,A_{pre}}} + \overline{\Delta A_{pre} \frac{\partial C_{sat}}{\partial A_{pre}}\Big|_{pCO_2,S,T}} \tag{B1}$$

Note: Salinity is conserved, but re-distributed. Hence, when using Eq. (B1) at the local scale, the term dependent on $\Delta S$ must be included. It only disappears after global integration, assuming that $\frac{\partial C_{sat}}{\partial S} \simeq constant$, which is done here.

We calculate the first term on the right hand side in Eq. (B1) on the local scale, by solving the carbon system equations using CO2SYS and taking the global average of the grid cell difference $C_{sat}(T_{SE(n)}) - C_{sat}(T_{PIES278})$ for each ensemble member,

while keeping salinity and alkalinity constant using the $PIES278$ grid cell salinity and $A_{pre}$. Similarly, the second term on the right hand side is calculated as the global average of grid cell $C_{sat}(A_{pre,SE(n)}) - C_{sat}(A_{pre,PIES278})$ for each ensemble member, using $T$ and $S$ of $PIES278$.

## B2 Contribution by the biological pumps

The calculations in this section largely follow the Appendix in Lauderdale et al. (2013), who studied the correlation between wind-driven changes of the residual circulation in the Southern Ocean and changes in ocean carbon reservoirs and atmospheric $CO_2$. In contrast to Lauderdale et al. (2013), our model computes $A_{pre}$ and $O_{2_{pre}}$, which reduces the sources of error in Eqs. B2 and B4. All stoichiometric ratios for organic material are based on Redfield (1963).

$C_{soft}$ corresponds to the carbon that has been added to the water parcel through the remineralisation of the soft tissue of biogenic material that has entered the water parcel. The global ocean average of $C_{soft}$ is calculated as

$$\overline{C_{soft}} = -R_{C:O_2} \cdot \overline{AOU} \tag{B2}$$

$$AOU = O_{2_{pre}} - O_2 \tag{B3}$$

where $R_{C:O_2}$ is the stoichiometric ratio of carbon to oxygen of $-106/138 \approx -0.768$ and $AOU$ is the apparent oxygen utilisation. As seen in Eq. (B3), we calculate $AOU$ as the difference between the preformed concentration of oxygen, $O_{2_{pre}}$, which is the concentration that the water had at the surface before it was subducted, and the actual concentration of oxygen, $O_2$, that is registered in the water. This difference is due to decomposition of organic soft-tissue material, which consumes oxygen. Therefore, $AOU$ can be used to 'back-track' the amount of nutrients or carbon that was brought into the deep ocean trapped in organic material, which has then been remineralised. When $O_{2_{pre}}$ is not an available quantity, it is commonly assumed that oxygen is at equilibrium at the ocean surface. $AOU$ is then calculated by replacing $O_{2_{pre}}$ with the saturation concentration for oxygen at the ambient temperature and salinity. In the real ocean, oxygen disequilibrium is negative and though it is small, it is not negligible. Hence, using $O_{2_{pre}}$ gives a more reliable result for $AOU$.

The biogenic material also carries hard tissue, and the carbon dissolved from this tissue is denoted $C_{carb}$. We calculate the grid cell concentration of $C_{carb}$ as

$$C_{carb} = \frac{1}{2}(A_{tot} - A_{pre} - R_{N:O_2} \cdot AOU) \tag{B4}$$

where $A_{tot}$ is the grid cell alkalinity and $R_{N:O_2}$ is the stoichiometric ratio of nitrogen to oxygen of 16:-138 ($\approx -0.116$). We can then calculate the volume-weighted global average of $C_{carb}$.

Finally, $\Delta C_{soft}$ and $\Delta C_{carb}$ for each $SE(n)$, where $n = (1, ..., 12)$, are given by the difference between the global average concentration in the $SE$ and in $PIES278$, as

$$\Delta\overline{C_{soft}} = \overline{C_{soft_{SE(n)}}} - \overline{C_{soft_{PIES278}}} \tag{B5}$$

$$\Delta\overline{C_{carb}} = \overline{C_{carb_{SE(n)}}} - \overline{C_{carb_{PIES278}}} \tag{B6}$$

$$\tag{B7}$$

## B3   Contribution by residual and desequilibrium carbon

Residual carbon, $C_{res}$, is $TC$ minus the contributions from the computed $C_{sat}$, $C_{soft}$ and $C_{res}$. The residual will mainly consist of desequilibrium carbon, $C_{dis}$. Since $C_{dis}$ is determined at the surface, before water sinks, by definition $C_{dis} = C_{pre} - C_{sat}$. Assuming the methods described in Section B1 give a good approximation of $C_{sat}$, $DIC_{pre} - C_{sat}$, where $DIC_{pre}$ is the model concentration of preformed carbon, gives a good approximation of $C_{dis}$.

## Appendix C:   Corresponding changes in $pCO_2^{atm}$

To achieve the equilibirium states of our ensemble ($SE1$–$SE12$), we have been restoring $pCO_2^{atm}$ to 278 µatm (µatm ≈ ppm). This means that the changes in carbon cycling caused by the imposed circulation changes are only seen as changes in ocean carbon storage. In the real world, we would also get an effect on the air-sea equilibirum and on $pCO_2^{atm}$.

In this section, we translate the observed $\Delta TC$, $\Delta C_{sat}$, $\Delta C_{soft}$, $\Delta C_{carb}$, and $\Delta C_{res}$ to the effect on $pCO_2^{atm}$ that would have been seen if it had not been restored. The fact that we are indeed changing the total carbon inventory of the system means
that these translations are approximate, since they assume the inventory to be constant. However, the changes in inventory are small compared to the size of the total inventory and therefore these calculations are still reasonably correct.

First, we need to know the Revelle buffer factor, $R_C$, for the control equilibrium, where $R_C$ is

$$R_C = \frac{\Delta[CO_2]}{[CO_2]} \bigg/ \frac{\Delta[DIC]}{[DIC]} = \frac{\Delta pCO_2^{atm}}{pCO_2^{atm}} \bigg/ \frac{\Delta[DIC]}{[DIC]} \approx \frac{\Delta[CO_2]}{[CO_2]} \bigg/ \frac{\Delta[C_{sat}]}{[C_{sat}]} \tag{C1}$$

where $[CO_2]$ and $[DIC]$ are the concentrations of dissolved $CO_2$ and DIC in the surface ocean. Assuming thermodynamic equilibrium between the atmosphere and the sea surface carbonate system, $C_{dis} = 0$ and $[DIC] = C_{sat}$ in the surface. Note that $C_{dis} \neq 0$ in our simulations, but $C_{sat} >> C_{dis}$, thus the approximation $[DIC] \approx C_{sat}$ is reasonable. $R_C$ is calculated by using the carbon system equation solver CO2SYS (Lewis et al., 1998), which also assumes thermodynamic equilibrium. As
input, we use the control equilibirum global averages of temperature, salinity and concentrations of $A_{pre}$ and $PO_4$, which are 3.58 °C, 34.90, 2296 µmol kg$^{-1}$, 2.15 µmol kg$^{-1}$ respectively. $R_C$ is given as standard output from CO2SYS. Given the control $pCO_2^{atm}$ of 278 ppm, $R_C$ is 12.4, which we then use as $R_C$ in the rest of our calculations. From the equation solver, we also get the global average ocean concentration of $DIC$ that corresponds to the given conditions. We call this concentration $DIC_{ref}$ and it is 2100 µmol kg$^{-1}$.
$DIC_{ref}$ is used as input when we next prepare to calculate the alkalinity factor, $R_A$, of the system. This factor is used to calculate the effect of a change in alkalinity on $pCO_2^{atm}$. $R_A$ is given by

$$R_A = \frac{\Delta[CO_2]}{[CO_2]} \bigg/ \frac{\Delta[A_{pre}]}{[A_{pre}]} \tag{C2}$$

Again, we use CO2SYS, with the same control state equilibrium parameters as before, but now fixing $DIC_{ref}$. This time we get a value for $pCO_2^{atm}$ as output. We will denote this output $pCO_{2alk}$. We also let CO2SYS calculate $pCO_2^{atm}$ for a 1

% increase in $\overline{A_{pre}}$. $pCO_{2_{alk}}$ is 278 ppm in the case with all control values and 248 ppm in the case with increased $A_{pre}$. We take the average of these two calculations, 263 ppm, as $\overline{pCO_{2_{alk}}}$ and $\Delta pCO_{2_{alk}} = 278 - 248 = 30$ ppm. With these values, Eq. (C2) gives $R_A = -11.4$.

In the following derivation, keep in mind that $|R_C| \sim |R_A|$ and note that

$$\left.\frac{\partial C_{sat}}{\partial A_{pre}}\right|_{pCO_2, T} \simeq 1 \tag{C3}$$

We make our equilibrium state simulation $PIES278$ and an ensemble of equilibrium simulations with modified circulation. According to Eq. 4, and with the separation of $\Delta C_{sat}$ into $\Delta C_{sat,T}$ and $\Delta C_{sat,A_{pre}}$,

$$\Delta TC = M_a \Delta pCO_2^{atm} + M_o(\Delta \overline{C_{sat}} + \Delta C_d) \tag{C4}$$

where

$$\Delta C_d \equiv \Delta \overline{C_{soft}} + \Delta \overline{C_{carb}} + \Delta \overline{C_{res}} \tag{C5}$$

We first study the case where $pCO_2^{atm}$ is held constant. The observed $\Delta TC$ in an ensemble member is

$$\Delta TC^* = M_o(\Delta \overline{C_{sat}^*} + \Delta C_d^*) \tag{C6}$$

which is based on observed changes (marked by *) in each carbon component .

For the hypothetical case where we keep $TC$ constant and allow $pCO_2^{atm}$ to vary, Eq. C4 becomes

$$M_a \Delta pCO_2^{atm} + M_o(\Delta \overline{C_{sat}} + \Delta C_d) = 0 \tag{C7}$$

Note that we henceforth neglect any changes in circulation that would occur in the ensemble member due to this change in $pCO_2^{atm}$. Thus we assume

$$\Delta C_d = \Delta C_d^* \tag{C8}$$

and $\Delta T = \Delta T^*$ ($T$ is water temperature), and $\Delta A_{pre} = \Delta A_{pre}^*$. We know that $C_{sat} = C_{sat}(T, A_{pre}, pCO_2^{atm})$. Consequently, any differences between $\Delta C_{sat,T}$ and $\Delta C_{sat,T}^*$ or $\Delta C_{sat,A_{pre}}$ and $\Delta C_{sat,A_{pre}}^*$ are due to changes in $\Delta pCO_2$.

We seek to compute $\Delta pCO_2^{atm}$ as a function of the result in Eq. C6. We linearise around the ensemble member equilibrium state in the case with constant $pCO_2^{atm}$:

$$\Delta \overline{C_{sat}} = \Delta \overline{C_{sat}^*} + \left.\frac{\partial C_{sat}}{\partial pCO_2^{atm}}\right|_{T,A_{pre}} \Delta pCO_2^{atm} \tag{C9}$$

Using Eqs. C9 and C8 in Eq. C7 gives

$$M_a \Delta pCO_2^{atm} + M_o \left( \Delta pCO_2 \frac{\partial C_{sat}}{\partial pCO_2^{atm}} + \Delta \overline{C_{sat}^*} - \Delta C_d^* \right) \tag{C10}$$

which together with Eq. C6 yields

$$M_o \frac{\partial C_{sat}}{\partial pCO_2^{atm}} \bigg|_{T,A_{pre}} \Delta pCO_2^{atm} = -\Delta TC^* \tag{C11}$$

if we use that $M_a \ll M_o$ and neglect terms that are $\sim M_a$. Thus

$$\Delta pCO_2^{atm} = -\frac{\partial pCO_2^{atm}}{\partial C_{sat}} \bigg|_{T,A_{pre}} \frac{\Delta TC^*}{M_o} \tag{C12}$$

or, using Eq. C1,

$$\Delta pCO_2^{atm} = -R_C \frac{\Delta TC^*}{M_o \overline{C_{sat}}} pCO_2^{atm} \approx -R_C \frac{\Delta TC^*}{M_o [DIC]} pCO_2^{atm} \tag{C13}$$

This equation gives us the total change in $pCO_2^{atm}$ that would occur when going from the control equilibrium state $PIES278$
to one of the $SE$ ensemble member equilibrium states while allowing $pCO_2^{atm}$ to vary. Here, we use $pCO_2^{atm}$ = 278 ppm.
Note again that we make use of the observed $\Delta TC^*$ from the ensemble member simulation where we kept $pCO_2^{atm}$ constant.
$\Delta TC^*$ can be replaced by $M_o \Delta C_{soft}^*$, $M_o \Delta C_{carb}^*$ and $M_o \Delta C_{res}^*$ in the above equation, to give each of their contributions to
the total $\Delta pCO_2^{atm}$. For example

$$\Delta pCO_2^{atm}(C_{res}) = -R_C \frac{\Delta \overline{C_{res}^*}}{[DIC]} pCO_2^{atm} \tag{C14}$$

Changes in $C_{sat}$ are separated into contributions by $C_{sat,A_{pre}}$ and $C_{sat,T}$. $\Delta C_{sat,A_{pre}}$ is due to changes in $A_{pre}$ caused by
the biological hard- and soft-tissue pumps respectively, thus

$$\Delta A_{pre} = \Delta A_{hard} + \Delta A_{soft} \tag{C15}$$

and from Eq. C2 we have that

$$\Delta pCO_2^{atm}(A_{pre}) = R_A \frac{\Delta A_{pre}}{A_{pre}} pCO_2^{atm} \tag{C16}$$

$C_{carb}$ originates from biogenic hard tissue. Formation of such hard tissue in the surface ocean is associated with an uptake of DIC ($C_{carb}$), which acts to decrease $pCO_2^{atm}$, and with an uptake of alkalinity, which is twice as large and acts to decrease solubility of $CO_2$ (i.e. $C_{sat,A_{pre}}$ changes)(see e.g., Williams and Follows, 2011). Thus

$$\Delta A_{hard} = 2\Delta C_{carb} \qquad\qquad\qquad (C17)$$

     Similarily, remineralisation of organic nitrogen decreases alkalinity. This effect is associated with the soft-tissue pump and
changes $A_{pre}$ by

$$\Delta A_{soft} = R_{N:O_2} \cdot \Delta\overline{AOU} = R_{N:O_2}\frac{\Delta\overline{C_{soft}}}{-R_{C:O_2}} = -R_{N:C}\Delta\overline{C_{soft}} \qquad\qquad\qquad (C18)$$

, and thus we get

$$\Delta A_{pre} = 2\Delta C_{carb} - R_{N:C}\Delta\overline{C_{soft}} \qquad\qquad\qquad (C19)$$

     If we use Eq. C19 in Eq. C20, we see that

$\Delta pCO_2^{atm}(A_{pre}) = R_A\dfrac{2\Delta C_{carb} - R_{N:C}\Delta\overline{C_{soft}}}{A_{pre}}pCO_2^{atm}$           (C20)

The simplifications we make here, assuming that $C_{sat}$ can replace [DIC] in Eq. (C1), and that, to leading order approximation, the sensitivity of $pCO_2^{atm}$ to changes in carbon species is independent of the size of the carbon reservoirs in the atmosphere and ocean respectively, are similar to those made by Kwon et al. (2011). In contrast to Kwon et al. (2011, author), we treat the effect of the biological pump on alkalinity, and hence on $CO_2$ solubility, separately by specifying $C_{sat,A_{pre}}$. Note that Kwon et al. (2011, author) estimate that the excursion in $pCO_2^{atm}$ due to changes in $C_{soft}$ is overestimated by about 10–15 %, due
to the assumption that $pCO_2^{atm}$ is insensitive to the size of the carbon reservoirs.

     The only remaining carbon species is $C_{sat,T}$. We calculate the effects on $pCO_2^{atm}$ corresponding to our observed $\Delta C_{sat,T}$ directly by solving the carbon system equations using CO2SYS. As input data we use the observed salinity, global average concentration of $DIC$ and surface $PO_4$ of the control equilibrium, as well as the restored value of $pCO_2^{atm}$ = 278 ppm. We run CO2SYS with the $PIES278$ global average temperature and the $SE$ ensemble member global average temperature. As
output, we then get the $pCO_2^{atm}$ at these two temperatures. Thus, we can compute the $\Delta pCO_2^{atm}$ we would get as a result of solely changing the temperature and keeping everything else but $pCO_2^{atm}$ constant.

     Instead of solving the carbon system equations to get the change in $pCO_2^{atm}$, Goodwin et al. (2011) suggest using a simplified equation, where the fractional change $\Delta pCO_2^{atm}/pCO_2^{atm}$, is described as proportional to a function of global average ocean temperature;

$$\frac{\Delta pCO_2^{atm}}{pCO_2^{atm}} \sim -\frac{V}{I_B}\frac{\partial C_{sat}}{\partial T}\Delta T_{avg} \tag{C21}$$

$$I_B = I_A + I_O/R_{global} \tag{C22}$$

Here, $\frac{\partial C_{sat}}{\partial T}$ (mol m$^{-3}$ °C$^{-1}$) is the change in saturation concentration of DIC per unit change of seawater temperature, in this case global ocean average temperature. We use that $\frac{\partial C_{sat}}{\partial T}\Delta T_{avg} = \Delta\overline{C_{sat,T}}$ and $\Delta\overline{C_{sat,T}}$ is known from Appendix B1. Thus, $\frac{\partial C_{sat}}{\partial T}$ is determined individually for each ensemble member. Note also that we calculate $\Delta\overline{C_{sat,T}}$ using the temperature

restriction on the gas solubility. $V$ is the ocean volume (m$^3$) and $I_B$ is the buffered amount of carbon in the system, or in other words the CO$_2$ that is "available for redistribution between the atmosphere and ocean" (Goodwin et al., 2007). As described by Eq. (C22), $I_B$ is based on the atmospheric carbon inventory, $I_A$, and the ocean inventory of DIC, $I_O$, scaled by the global value for the Revelle buffer factor. In this case, $R_{global} = R_C$ (see Eq. C1).

*Author contributions.* M. Ödalen, J. Nycander and K. I. C. Oliver designed the model experiments. A. Ridgwell developed the cGENIE

model code with addition of new tracers. M. Ödalen adapted the code for the experimental design, performed the model simulations and produced the figures. L. Brodeau provided technical expertise for the model setup. M. Ödalen prepared the manuscript with contributions from all co-authors.

*Competing interests.* No competing interests are present.

*Acknowledgements.* The simulations were performed on resources provided by the Swedish National Infrastructure for Computing (SNIC)

at the National Supercomputer Centre, Linköping University. Malin Ödalen and Jonas Nycander would like to acknowledge the Bolin Centre for Climate Research for financial support. Kevin Oliver would like to acknowledge the support by UK NERC grant NE/K002546/1 and he is grateful to the International Mereorological Institute for generous support during several visists to Stockholm. Andy Ridgwell was supported by EU grant ERC 2013-CoG-617313. This work benefited from helpful discussions with Johan Nilsson.

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

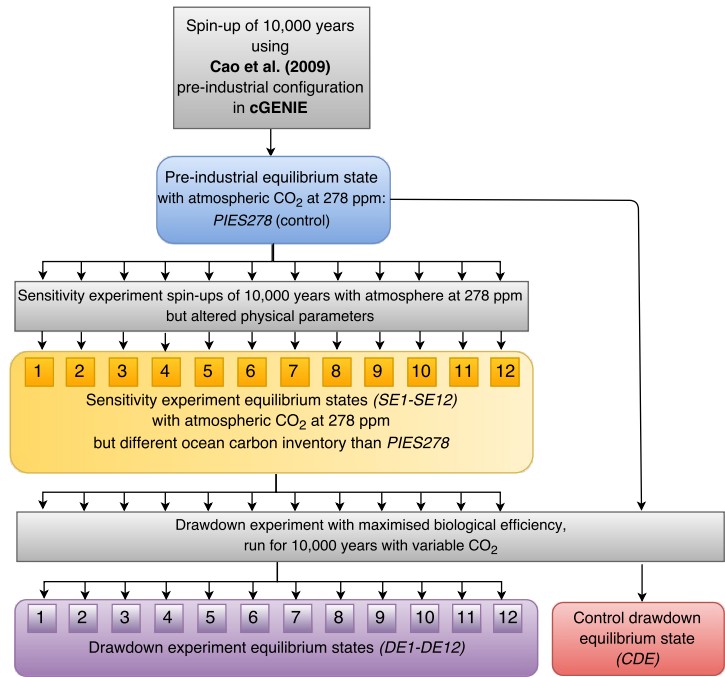

**Figure 1.** Flow chart showing the experimental setup. Grey boxes are spin-ups and transient stages of simulations (not analysed). Coloured boxes are equilibrium states that are analysed in this study. Throughout the study, the pre-industrial equilibrium state $PIES278$ (light blue box) is used as the control state, with which we compare the sensitivity experiment equilibrium states $SE1$–$SE12$ (yellow box). The change in physical characteristics for each $SE$ state compared to the control state $PIES278$ is described in Table 1. The $SE$s are then used as a basis for the $CO_2$ experiments where biological efficiency is maximised. After running the drawdown experiments for 10,000 model years, we achieve a new ensemble of drawdown equilibrium states ($DE1 - DE12$) which are compared to a control drawdown equilibrium state (CDE).

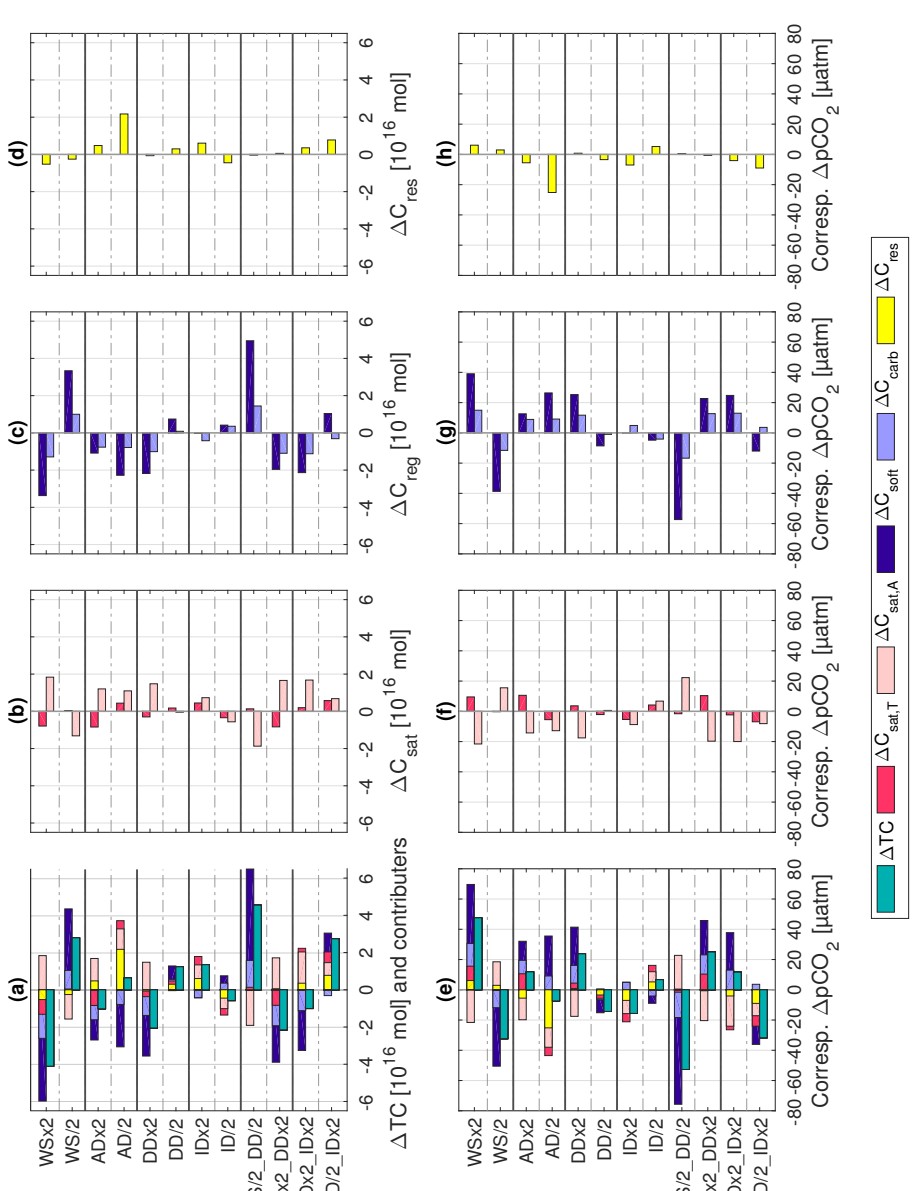

**Figure 2.** In panel a, green bars show observed $\Delta TC$ inventory for $SE1$–$SE12$, as compared to the TC inventory of the control $PIES278$. Other bars show relative contributions to $\Delta TC$ inventory by changes in the solubility (due to temperature = red, due to preformed alkalinity = pink, panels a and b), biological soft tissue (dark blue, panels a and c) and hard tissue (light purple, panels a and c) pumps. The residual of the theoretical contributions by changes in $C_{soft}$, $C_{carb}$ and $C_{sat}$ to $\Delta TC$ (calculations made using Eqs. (B2)–(B1)) and the observed model $\Delta TC$ is denoted $C_{res}$ (yellow bars). The horizontal axis shows magnitude of changes given in $10^{16}$ mol ($\sim 120$ Pg C). The lower panels (e–h) show approximate $pCO_2^{atm}$ equivalent given in $\mu atm$. Hence, this shows how big the difference in $pCO_2^{atm}$ would be between ensemble members if we were not restoring to 278 $\mu atm$. Note that a positive (negative) $\Delta TC$ indicates a higher (lower) storage of $CO_2$ in the ocean, which would cause a lower (higher) $pCO_2^{atm}$.

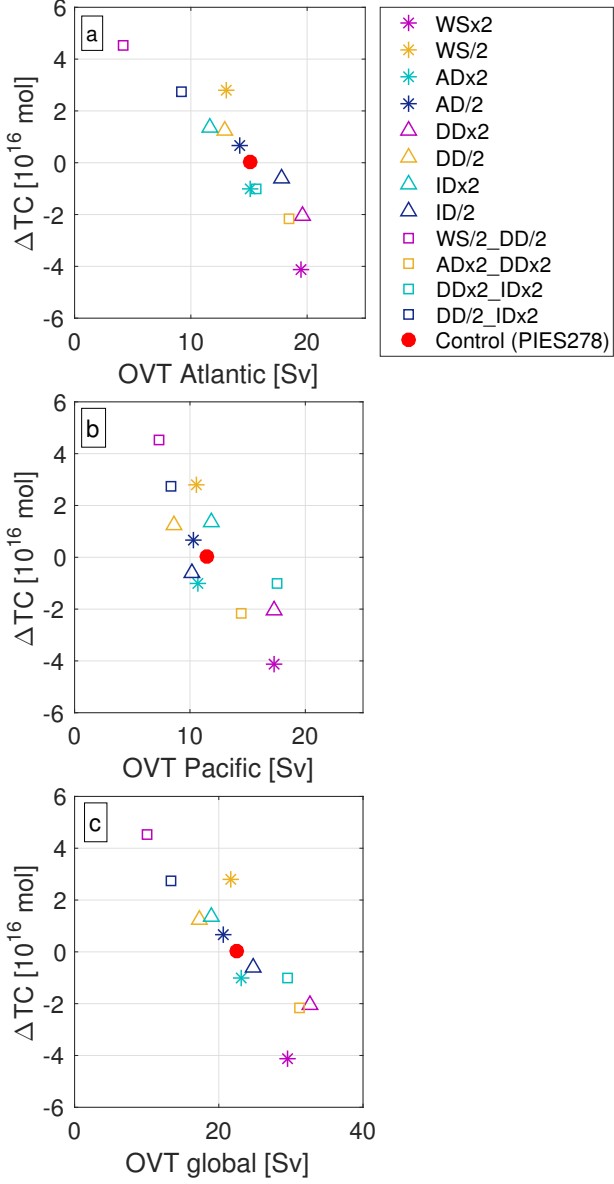

**Figure 3.** Strength of the overturning circulation in terms of $OVT$ (see Section 3.4), and the change in $TC$ inventory for equilibrium states $SE1$–$SE12$ c.f. the control $PIES278$ (red dot). This is shown for the Atlantic basin, Pacific basin and for the global measure based on hemispheric differences, on the horizontal axis of panels a, b and c respectively.

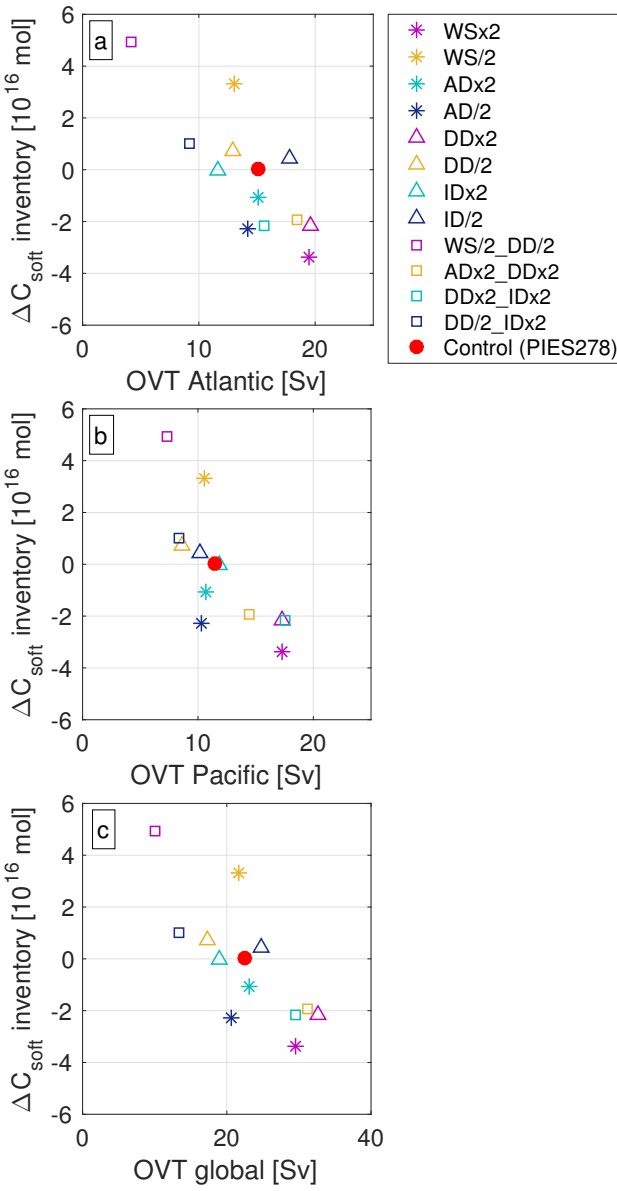

**Figure 4.** Strength of the overturning circulation in terms of $OVT$ (see Section 3.4), and the change in $C_{soft}$ inventory ($M_o \cdot \Delta C_{soft}$) for equilibrium states $SE1$–$SE12$ c.f. the control $PIES278$ (red dot). This is shown for the Atlantic basin, Pacific basin and for the global measure based on hemispheric differences, on the horizontal axis of panels a, b and c respectively.

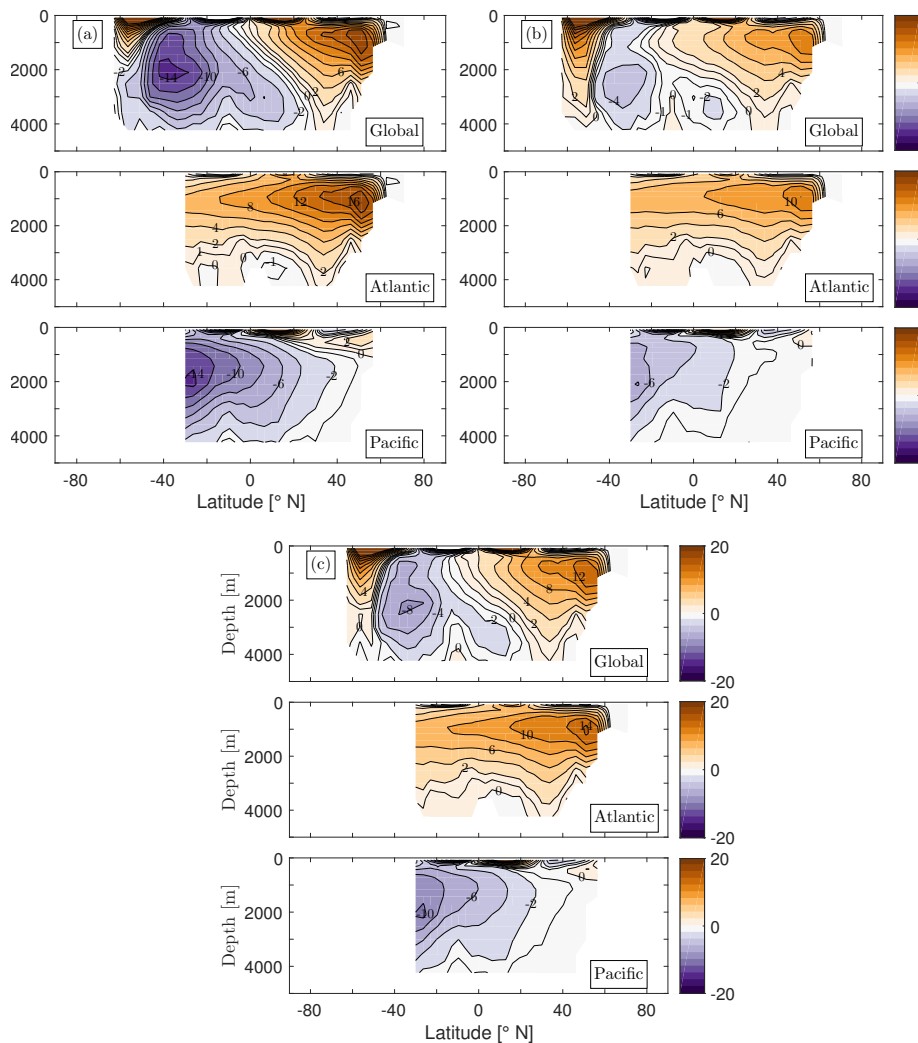

**Figure 5.** Zonal average overturning streamfunction ($\psi$) in Sverdrup Sv for a) $DDx2$, with high diapycnal diffusivity, b) $DD/2$, with low diapycnal diffusivity, and c) the control $PIES278$. The upper panel in each subfigure shows the global $\psi$, the middle panel shows only the Atlantic sector ,and the lower panel shows only the Pacific sector. The southernmost limit for the Atlantic and Pacific sectors is -30° N.

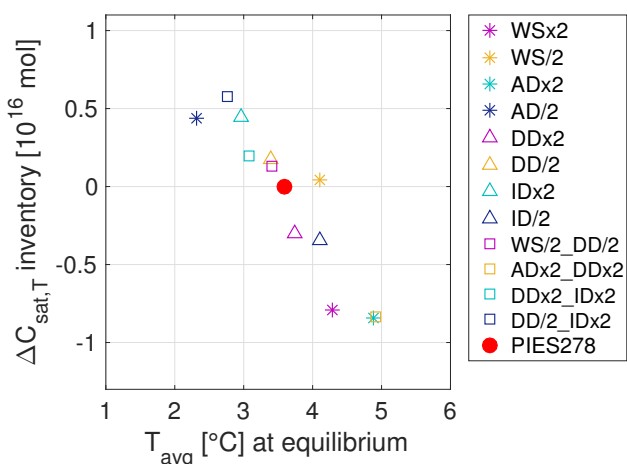

**Figure 6.** Global ocean average temperature ($T_{avg}$) and the changes in $TC$ that are due to solubility changing with temperature ($\Delta C_{sat,T}$) for the ensemble members $SE1$–$SE12$ compared to the control simulation $PIES278$ (red dot).

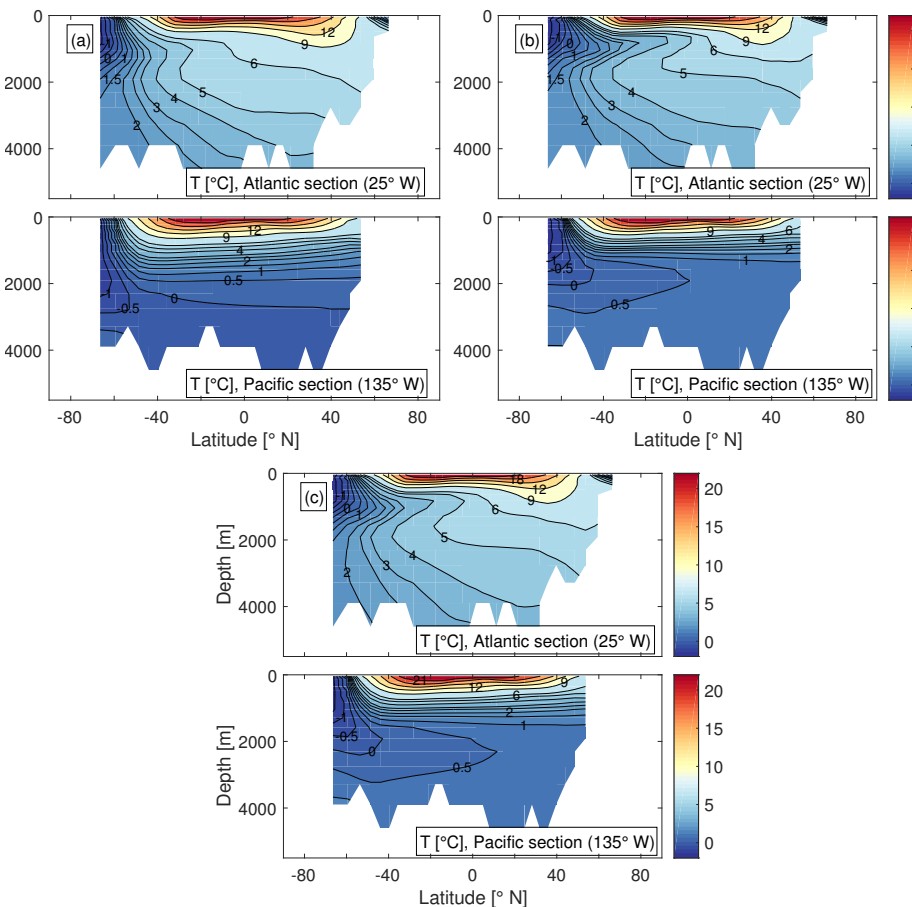

**Figure 7.** Sections of temperature (°C) for a) $DDx2$, with high diapycnal diffusivity, b) $DD/2$, with low diapycnal diffusivity, and c) the control $PIES278$. The upper panel of each subfigure shows a section through the Atlantic, at 25° W and the lower panel shows a section through the Pacific, at 135° W. Both sections also cover latitudes that are in the Southern Ocean (south of -30° N).

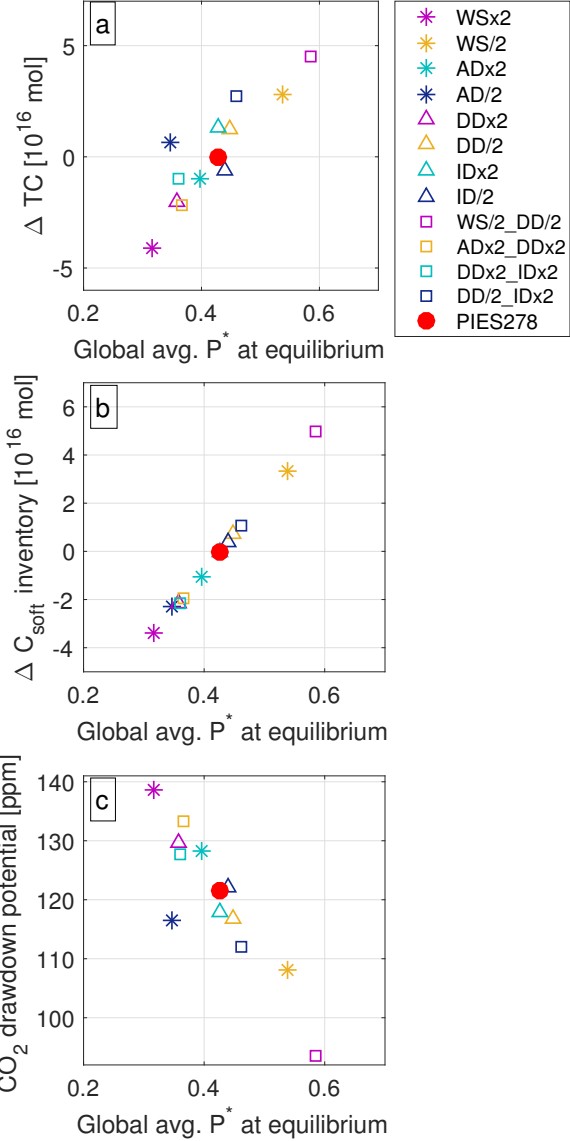

**Figure 8.** Panels showing (whole) global ocean $\overline{P^*}$ for the different ensemble members $SE1-SE12$ plotted versus a) change in total carbon inventory ($\Delta TC$, $10^{16}$ mol), b) the inventory change of $C_{soft}$, hence $M_o \cdot \Delta C_{soft}$ ($10^{16}$ mol), and c) the $CO_2$ drawdown potential (ppm) of each ensemble member, which is the lowering of $pCO_2^{atm}$ achieved by maximising biological efficiency (making $\overline{P^*}$ get equal to 1), see Eq. (1). Ensemble member characteristics are described in Table 1.

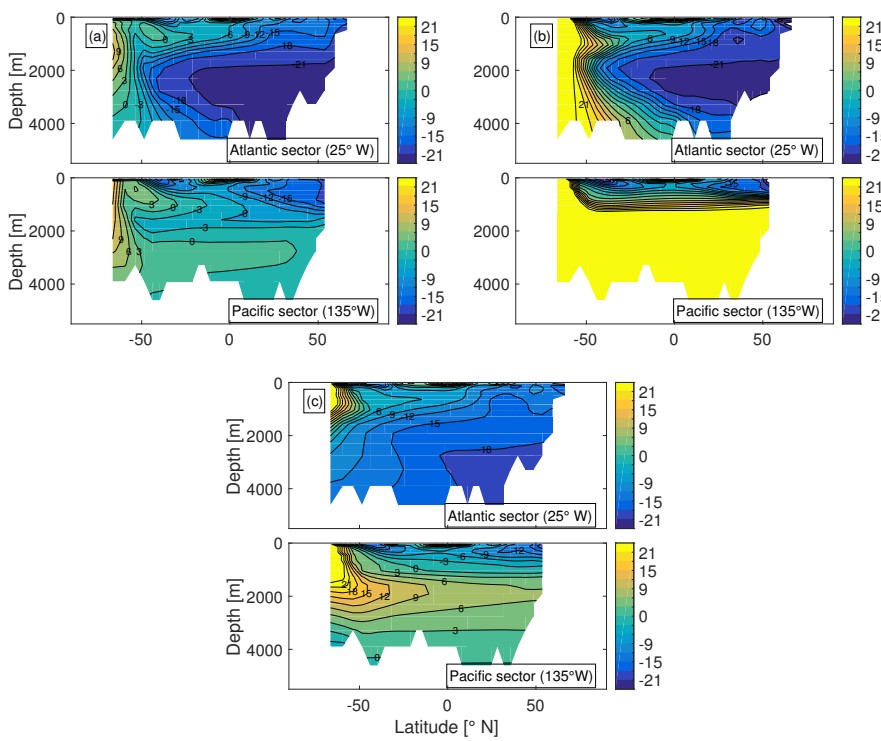

**Figure 9.** Example sections of $C_{dis} = DIC_{pre} - C_{sat}$ (Appendix B3). The panels show $C_{dis}$ ($\mu mol kg^{-1}$) for a) $WSx2$, with doubled wind stress, b) $AD/2$, with halved atmospheric heat diffusivity and c) the control equilibirum $PIES278$. The upper panel of each subfigure shows a section through the Atlantic, at 25° W and the lower panel shows a section through the Pacific, at 135° W. Both sections also cover latitudes that are in the Southern Ocean (south of -30° N).

**Table 1.** List of sensitivity experiment equilibrium states $SE1$–$SE12$, abbreviated ensemble member description, and specification of which one or two physical characteristics have been altered compared to the control $PIES278$. The nature of the change is specified within parenthesis.

| Ensemble member | Abbreviated description | Adjusted parameter (adjustment) |
|---|---|---|
| SE1 | 'WSx2' | Wind stress intensity (doubled) |
| SE2 | 'WS/2' | Wind stress intensity (halved) |
| SE3 | 'ADx2' | Atmospheric heat diffusivity (halved) |
| SE4 | 'AD/2' | Atmospheric heat diffusivity (doubled) |
| SE5 | 'DDx2' | Ocean diapycnal diffusivity (doubled) |
| SE6 | 'DD/2' | Ocean diapycnal diffusivity (halved) |
| SE7 | 'IDx2' | Ocean isopycnal diffusivity (halved) |
| SE8 | 'ID/2' | Ocean isopycnal diffusivity (doubled) |
| SE9 | 'WS/2_DD/2' | Wind stress intensity (halved) and ocean diapycnal diffusivity (halved) |
| SE10 | 'ADx2_DDx2' | Atmospheric heat diffusivity (doubled) and ocean diapycnal diffusivity (doubled) |
| SE11 | 'DDx2_IDx2' | Ocean diapycnal, and isopycnal diffusivity (doubled, doubled) |
| SE12 | 'DDx2_ID/2' | Ocean diapycnal, and isopycnal diffusivity (doubled, halved) |

**Table 2.** Diagnostic variables of the pre-industrial control states ($PIC$s) of PMIP2 and CMIP5/PMIP3 (temperature and salinity as read from Fig. 9.18. of WG1 in IPCC AR5 and AMOC as given in Table 1 in Muglia and Schmittner (2015)) compared to similar diagnostics for our ensemble $SE1$–$SE12$ and control state $PIES278$.

| Variables | $PIC$s of PMIP2 and CMIP5/PMIP3 | Ödalen et al. cGENIE $SE1$–$SE12$ and $PIES278$ |
|---|---|---|
| Potential temperature (°C), N. Atlantic | $2.9 - 6.4^1$ | $4.6 - 8.0^2$ |
| Potential temperature (°C), S. Atlantic | $-1.6 - 2.0^3$ | $-0.3 - 3.7^4$ |
| Salinity, N. Atlantic | $34.8 - 35.5^1$ | $35.3 - 35.6^2$ |
| Salinity, S. Atlantic | $34.6 - 35.0^3$ | $34.9 - 35.1^4$ |
| AMOC $(1Sv = 10^6 m^3 s^{-1})$ | $12.64 - 23.02^5$ | $2.0 - 18.0^6$ |

[1] North Atlantic PMIP grid point: 55.5°N, 14.5°W, 2,184 m depth

[2] North Atlantic cGENIE closest corresponding grid cell: 51-56°N, 10-20°W, 1,738-2,100 m depth

[3] South Atlantic PMIP grid point: 50°S, 5°E, 3,636 m depth

[4] South Atlantic cGENIE closest corresponding grid cell: 46-51°S, 0-10°E, 3,008-3,576 m depth

[5] Muglia and Schmittner (2015), PMIP3 pre-industrial control ensemble AMOC at 25°N, average with interval of one standard deviation [6] cGENIE maximum Atlantic overturning. Ensemble member $WS/2\_DD/2$ (see Table 1) has a collapsed AMOC circulation (2.0 Sv). The average of this variable for all other $SE$s is 13.8 Sv (range 8.3 – 18.0 Sv).

**Table 3.** Diagnostic variables for observations (Obs.), the control $PIES278$ (Ctrl.) and the ensemble members $SE1$–$SE12$. The variables are global ocean averages of temperature ($T_{avg}$, °C) and $pH$ ($pH_{avg}$), surface ocean average $pH$, the sea ice cover (%), the global average of the nutrient utilisation efficiency (expressed in terms of $\overline{P^*}$) and a measure of the strength of the global ocean overturning circulation, $OVT$ (see Section 3.4). Observational estimate for $T_{avg}$ has been calculated using World Ocean Atlas 2013, Locarnini et al. (2013)) and the pre-industrial estimate for $pH$ is given by Raven et al. (2005). Modern day sea ice cover is given as an interval due to seasonal variability Comiso (2008). The observational estimate for $\overline{P^*}$ is given by Ito and Follows (2005)

| Ens. mem. | Experiment name | $T_{avg}$ (°C) | $pH_{avg}$ (SWS) | $pH_{surf}$ (SWS) | Sea ice cover (%) | Global $\overline{P^*}$ | Global $OVT$ (Sv) |
|---|---|---|---|---|---|---|---|
| Obs. | - | 3.49 | - | ~8.2 | 3 to 6 | 0.36 | – |
| Ctrl. | PIES278 | 3.58 | 7.90 | 8.16 | 5.4 | 0.43 | 22.5 |
| SE1 | WSx2 | 4.28 | 7.97 | 8.16 | 2.7 | 0.32 | 29.4 |
| SE2 | WS/2 | 4.10 | 7.84 | 8.16 | 6.7 | 0.54 | 21.6 |
| SE3 | ADx2 | 4.88 | 7.91 | 8.17 | 0.7 | 0.40 | 23.2 |
| SE4 | AD/2 | 2.31 | 7.89 | 8.15 | 10.6 | 0.35 | 20.6 |
| SE5 | DDx2 | 3.73 | 7.94 | 8.16 | 4.1 | 0.36 | 32.6 |
| SE6 | DD/2 | 3.39 | 7.87 | 8.15 | 6.1 | 0.45 | 17.4 |
| SE7 | IDx2 | 2.96 | 7.88 | 8.16 | 5.4 | 0.43 | 19.0 |
| SE8 | ID/2 | 4.10 | 7.90 | 8.16 | 5.3 | 0.44 | 24.8 |
| SE9 | WS/2_DD/2 | 3.41 | 7.80 | 8.15 | 7.4 | 0.59 | 10.1 |
| SE10 | ADx2_DDx2 | 4.91 | 7.93 | 8.17 | 0.0 | 0.37 | 31.1 |
| SE11 | DDx2_IDx2 | 3.07 | 7.93 | 8.16 | 3.7 | 0.36 | 30.0 |
| SE12 | DD/2_IDx2 | 2.76 | 7.85 | 8.15 | 6.0 | 0.46 | 13.4 |

**Table 4.** Correlation coefficients of the changes in $OVT$ (see Section 3.4), sorted by geographical regions, and the anomaly in each carbon species for the $SE$-ensemble (relative to $PIES278$).

| $OVT$ in region | $\Delta TC$ | $\Delta C_{sat,T}$ | $\Delta C_{sat,A}$ | $\Delta C_{soft}$ | $\Delta C_{carb}$ | $\Delta C_{res}$ | $\Delta C_{soft} + \Delta C_{res}$ |
|---|---|---|---|---|---|---|---|
| Atlantic | -0.92 | -0.64 | 0.65 | -0.80 | -0.69 | -0.25 | -0.89 |
| Pacific | -0.81 | -0.43 | 0.72 | -0.75 | -0.74 | -0.27 | -0.84 |
| Global | -0.89 | -0.63 | 0.67 | -0.77 | -0.70 | -0.30 | -0.87 |