# Peer review of "The influence of the ocean circulation state on ocean carbon storage and CO2 drawdown potential in an Earth system model"

_Biogeosciences, 2017_

## Referee Comment (RC1) · Anonymous Referee #1 · 20 Jun 2017

Review of Odalen et al. "The influence of the ocean circulation state on ocean carbon storage and CO2 drawdown potential in an Earth system model"

IMPORTANCE: Earth system models are essential tools for understanding of climate-carbon linkages, both for ocean carbon uptake in the past, and for understanding how future carbon emissions will translate into ocean carbon uptake and global temperature change. Models that conduct these experiments are generally tuned to modern conditions, but this tuning process can result in several initial states (with different initial ocean carbon storage), which may affect the model-ocean's capacity to take up carbon.

SUMMARY: The purpose of this study is to quantify the influence of the initial, equi-

librium state of 12 different model versions (in which vertical and horizontal diffusivity, atm diffusivity, and wind stress were modified) on their initial ocean C storage and CO2 uptake potential. Simulations with higher initial overturning rates tend to have lower total initial ocean C storage, largely attributed to reduced solubility and soft tissue pumps (because of shorter residence times of nutrients at the surface). In contrast, simulations with lower overturning rates (from low wind stress and reduced vertical diffusivity) tend to have higher global nutrient utilization (P*), higher initial carbon storage, and lower C uptake capacity. The initial state of nutrient utilization efficiency (P*) is a strong determiner of CO2 uptake capacity and varies by 50 ppm between initial model states, with versions with low initial efficiencies having higher uptake capacities. Different initial average global ocean temperatures can result in a ∼25-30ppm differences in initial pCO2atm (comparable magnitude to glacial-interglacial effects of temperature-dependent solubility); colder simulations with higher C storage due to saturation responses tend to have lower drawdown potentials.

REVIEW: This is going to be an important paper for highlighting the impact of initial equilibrium state – both for modern and LGM conditions – on modeled ocean capacity to take up carbon, and I look forward to seeing it published. However, at the moment, there is a lot going on in this paper... (1) One major concern is its length and level of detail which dilutes its most important points. On the one hand, I appreciate that the authors are attempting to describe a complex system clearly and completely. The background and methods sections do provide thorough definitions of the different carbon pumps, controls on alkalinity, and nutrient utilization efficiency, as well as a very detailed description of how the different carbon components are estimated in the model. At the same, all these components have been defined previously, so the paper might be shortened by placing large chunks into an appendix. The results section is similarly very wordy; it mixes methods, results, and discussion together; many points that take multiple paragraphs to make could be simplified to 1-2 sentences. I suggest a thorough attempt to go through this paper and streamline the writing. As one example, the entire top of page 13 describes how figure 2 will be put together, with only two

half-sentences (regarding ensemble range of pH and pCO2atm) that describe results. Some additional examples (there are more): LN 12-17 on pg 14 – this section does not describe any results presented. Pg 15 LN 18-31 – the average global temperature of simulations has a range of 2.3-4.9°C, which results in a range in delta pCO2atm from Csat of -16 – +17 ppm or -13 to +12ppm (depending upon how the calculation is made). Glacial-interglacial implications belong in discussion.

(2) Figure 2 summarizes all results, but its current presentation makes it very difficult to distill anything more than the general magnitudes. I suggest (1) providing the labels of the sensitivity experiments and sorting them somehow, perhaps by the anticipated magnitude of total effects, from largest to smallest; (2) separating this figure in to 3-4 panels: biological, residual, solubility, and total (indicating on the total plot the largest contributor to the total change).

(3) Pg 9 Methods: Are the ranges for vertical diffusivity, wind stress, horizontal diffusivity, etc that are used in the sensitivity experiments comparable to the range of values that are normally used to tune models? Would be useful to provide this information here, so that the reader can assess whether your sensitivity experiments represent values that might normally be used.

MINOR POINTS:

Pg. 9 LN 13 – confusing – do you mean that you hold ALK and P constant in your experiments? Pg 9 LN 14-20 – Upon first reading, it was unclear what delta (ïĄĎ) represents. Please define specifically that you are comparing the carbon estimates from PIES278 with equilibrium values from SE(n). Pg 9 and then again on Pg 10 – when you describe the experiments in which you have implemented artificially fast gas exchange to remove Cdis, please identify this experiment with its number listed in Table 1. Pg 14 LN 27 – here and throughout this discussion, the authors indicate that horizontal diffusivity affects Csat and Cres more than Csoft, but the more obvious result is the minimal impact on deltaTC overall. This is worth noting. Pg 15 LN 28 –

"In ensemble members in which horizontal diffusivity in the ocean is changed, _Csat is larger than _Csoft." When horizontal diffusivity is reduced or increased? Specify that it is larger whether horizontal diffusivity is increased or decreased. It is near impossible from Figure 2 to discern this. Pg 15 ln 29-37 – estimating the implications of the relationship between Csat and aveT for the glacial ocean is a point for discussion, not results Pg 20 LN 17 – Technically the authors have not shown the role of "AMOC strength," which refers specifically to the Atlantic overturning limb. The plots calculate the difference between northern (Atlantic) and southern source components and thus combine the roles of the no

---

## Referee Comment (RC2) · Anonymous Referee #2 · 20 Jun 2017

Ödalen et al. explore the partitioning of carbon in the ocean under different circulation and mixing scenarios using a model of intermediate complexity. The authors describe the possible change in atmospheric pCO2 in terms of the "drawdown potential" of each scenario. Changes in saturation and soft tissue pump carbon appear to have the largest effects on the total carbon inventory, though the carbonate and disequilibrium pumps are not entirely negligible.

This is an important topic for the community and addresses a standing question in paleoclimatology. The model and methods used are well established and credulous. However, prior to publication, this paper requires significant editing. In places, the lan-

guage is quite informal and should be revised (e.g. p. 19, lines 24-25: "...but does not seem to have been picked up in the model intercomparison community..."). Additional comments are given below.

General comments:

- I assume that you don't have preformed nutrients (O2, PO4, DIC) written out in the model output? If you do, this eliminates the problem with O2dis, as you can calculate O2sat and then calculate the remineralised O2 and hence Csoft explicitly. Additionally, preformed O2 and PO4 would be much more useful in the parameterisation for Alkpre.

- Is it possible to change the circulation in the model in the Southern Ocean and not in the Atlantic or vice versa? The bipolar seesaw could theoretically induce changes in Cdis in the Southern Hemisphere but not in the Northern Hemisphere.

- If it's not too cumbersome, perhaps consider using descriptive abbreviations for the different SEs

Specific comments:

- p. 3, lines 4-5: Could you specify the order of magnitude of the change in CO2?

- p. 3, line 35: It's also important to note that these studies don't even have Cres due to infinitely fast gas exchange

- p. 4, lines 3-5: I don't find this paragraph necessary

- p. 4, lines 28-29: I don't believe that this is correct, as the pumps can have opposing effects. It should be specified that the net effect of all of the pumps must be to redistribute carbon from the surface to the deep ocean.

- p. 5, line 17: I think what you mean is that alkalinity is not set (or affected) by gas exchange, but referring to an "expected" value of Cpre is a little misleading.

- p. 5, lines 31-32: Please include Martin (1990), Paleoceanography, as this is one of

the central references for increased soft tissue pump efficiency during glacials.

- p. 7, lines 5-6: "and has a level of detail for the carbon system that made it particularly suitable for this study." This is very general; please specify why it is appropriate for this study (and/or why less complex models are not).

- p. 8, line 25: When referring to Cdis, it would be useful to cite Ito and Follows (2013), GBC.

- p. 10, lines 14-24: Another important reference is Ito et al. (2004), GRL.

- Section 3.3: Do you use the same parameterisation for preformed alkalinity in all simulations? Please specify the errors in Cres in the surface field.

- p. 12, lines 17-22: Again, please specify the size of the error introduced by making this approximation.

- p. 14, lines 6-7: Isn't this relationship true per definition of Csoft?

- p. 15, lines 29-31: Please describe here the importance of the temperature limit on the calculation of Csat, if this is on a comparable order of magnitude.

- p. 17, lines 13-14: Have you done experiments to specifically examined he role of sea ice in determining Cdis? Quantitative results would be very interesting!

- p. 17, line 15: Only 0.01%? This seems to be at odds with Fig. 2.

- p. 18, line 15: Changes in the solubility due to ocean temperature changes don't seem "indirect"

- p. 19, lines 31-32: Ito and Follows (2013), GBC also uses the same scheme to look specifically at this; please include this.

- p. 20, lines 23-24: Please specify what you mean by "an LGM-like circulation" and add appropriate citations

- p. 22, line 27: Please cite the statement that "there may have been more, not less,

preformed nutrients in the deep ocean during the last glacial"

- p. 24, line 6: please specify if you mean the soft tissue pump and/or the carbonate pump

- Fig. 8: Perhaps difference sections would be more useful?

- Fig. 9: It would be more illustrative to zoom in with the colourbar

- Figures: Please make the font size larger, particularly in Fig. 7-10

- Table 3: Please give units for DC; why not include DCcarb?

Technical corrections:

- p. 2, line 2: "and" should be "an"

- p. 2, line 20: "constraining;" should be "constraining:"

- p. 4, line 11: "patwhay" should be "pathway"

- p. 5, line 4: "HCO3-" should be properly formatted

- p. 5, line 31: "eg." should be "e.g."

- p. 9, line 28: Full stop missing at end of sentence

- p. 10, line 1: "DE:s" should be "SEs"

- p. 10, line 5: "Eqs. 6-7" should be "Eqs. (6)-(7)"

- p. 14, line 24: "SE:s" should be "SEs"

- p. 14, line 25: "SE:s" should be "SEs;" there is an extra parenthesis

- p. 14, line 27: "SE:a" should be "SEs"

- p. 14, line 33: "SE:a" should be "SEs"

- p. 15, line 13: "SE:s" and "SE:a" should be "SEs"

- p. 15, line 21: extra full stop in "Fig. 2 b."

- p. 15, line 32: space missing in "andDCsat"

- p. 16, line 9: "SE:s" should be "SEs"

- p. 18, line 4: "SE:s" should be "SEs"

- p. 18, line 18: "SE:a" should be "SEs"

- p. 19, line 12: "particularily" should be "particularly"

- p. 20, line 11: "SE:s" should be "SEs"

- p. 20, line 24: "simulation" should probably be "circulation"

- p. 22, line 15: Comma after "that" should be removed

- p. 22, line 28: Citation should be in parentheses

- p. 23, line 1: the space in the website should be removed

- p. 24, line 3: "c.f." should be "cf."

- Fig. 1 caption: "the SE:s" should be "the SEs"

- many references are formatted Author (year) where they should be (Author, year), e.g. p. 2, lines 4 and 16

- bibliography: check that CO2 is written with a subscript; remove "n/a" from page numbers

---

## Referee Comment (RC3) · Anonymous Referee #1 · 21 Jun 2017

Final comment was clipped during upload. Here is the complete sentence:

Pg 20 LN 17 – Technically the authors have not shown the role of "AMOC strength," which refers specifically to the Atlantic overturning limb. The plots calculate the difference between northern (Atlantic) and southern source components and thus combine the roles of the northern and southern overturning strengths. Changes in the southern source might mask changes in the AMOC alone. I suggest using a different phrase here.

---

## Referee Comment (RC4) · Anonymous Referee #3 · 2 Jul 2017

The authors used previously published theoretical frameworks to interpret their sensitivity runs from an Earth system model of intermediate complexity. They first explored the oceanic storage of DIC equilibrated with a preindustrial atmospheric CO2 condition by changing ocean circulation patterns. Then the authors used the preindustrial equilibrium states as initial conditions to the experiment where they maximized the nutrient utilization efficiency (i.e., all of PO4 is utilized by biology). Their major conclusion is that the drawdown potential of atmospheric CO2 differs with different initial states, i.e., different circulation patterns. This could explain earlier model intercomparison studies where atmospheric CO2 response to the same perturbation shows a large spread among models.
Such model sensitivity experiments and extensive analyses using an Earth system model are unprecedented and have a potential to improve our understanding of the past changes in the carbon cycle. However, the present manuscript has some parts that are not clear, and also lacks novelty.

The conclusion that the oceanic storage of DIC and the drawdown of atmospheric CO2 in response to nutrient depletion all depend on the ocean circulation patterns (including the overturning strengths of NADW and AABW, and the volume fraction of the ocean last ventilated from the North Atlantic vs. Southern Ocean) is not new. The circulation effects on the ocean carbon pumps have been extensively studied using models and theoretical frameworks: the solubility pump (e.g., DeVries and Primeau, Atmospheric pCO2 sensitivity to the solubility pump: Role of the low-latitude ocean, GBC 2009), the biological pump (references already cited in the manuscript) and the disequilibrium pump (e.g. Ito and Follows, Air-sea disequilibrium of carbon dioxide enhances the biological carbon sequestration in the Southern Ocean, GBC 2013). One of the new points in this study is that the authors discussed the relative role of the three pumps in the net change of the simulated carbon cycle. But, because the forcing used to generate the different circulation patterns is arbitrary, their discussion of the relative roles does not seem very interesting. Overall, I feel that the authors need to highlight what new findings or insights this study can provide.

Fig. 10 showing the CO2 drawdown potential as a function of a change in the mean ocean temperature does not convey any messages. There seems no relationship between the two. Plus, if my reading is correct, water temperature does not control the strength/efficiency of the biological pump in the model. Therefore, there is no reason that the CO2 drawdown potential should be correlated with ocean temperature. Why don't the authors use other metrics such as the initial preformed PO4 as an X-axis instead, as was done in Marinov et al., 2008?

The way the biological pump is simulated in the model is unclear. The authors included the carbonate pump in their models and analyses (expressed as Ccarb), but there is no

description on how the carbonate pump is represented in the model. For example, are the sedimentation processes included in the model? How are the production and dissolution of calcifying organisms represented in the model? How is the strength/efficiency of the carbonate pump affected by the drawdown experiment? In the drawdown experiment (specified in lines #31-33 of page #9), the remineralization length scale of sinking organic particles is made very deep (10,000m), so that "any carbon that is taken up in organic material to be highly efficiently trapped in the deep ocean and not undergo any significant remineralization". Does it mean that most of inorganic nutrient is converted to organic form and stored in the abyss without being remineralized back to inorganic form? If this is the case, then the amount of organic matter would increase substantially in the drawdown experiment, and the carbon fixed in organic material should be an important component in the mass balance equations and can't be ignored in the theoretical derivations presented in the manuscript. Likewise, the equation "Ppre=P-Preg" would be incorrect. This needs to be clarified.

---

## Author Comment (AC1) · 6 Oct 2017

We thank the referee for an overall positive review, and for helpful comments that will allow us to improve the manuscript. In this comment we respond to the main review points listed by the referee. There is also a list of minor points made by the referee. Those will not be addressed here, but we are happy to include the requested corrections and clarifications in the updated manuscript.

Point 1: One major concern is its length and level of detail, which dilutes its most important points. On the one hand, I appreciate that the authors are attempting to describe a complex system clearly and completely. The background and methods

sections do provide thorough definitions of the different carbon pumps, controls on alkalinity, and nutrient utilization efficiency, as well as a very detailed description of how the different carbon components are estimated in the model. At the same, all these components have been defined previously, so the paper might be shortened by placing large chunks into an appendix. The results section is similarly very wordy; it mixes methods, results, and discussion together; many points that take multiple paragraphs to make could be simplified to 1-2 sentences. I suggest a thorough attempt to go through this paper and streamline the writing. As one example, the entire top of page 13 describes how figure 2 will be put together, with only two half-sentences (regarding ensemble range of pH and pCO2atm) that describe results. Some additional examples (there are more): LN 12-17 on pg 14 – this section does not describe any results presented. Pg 15 LN 18-31 – the average global temperature of simulations has a range of 2.3-4.9_C, which results in a range in delta pCO2atm from Csat of -16 – +17 ppm or -13 to +12ppm (depending upon how the calculation is made). Glacial-interglacial implications belong in discussion.

Response: We find the referee's criticism valid and will address the problem according to the suggestions. For example, as much as possible of Section 2, Section 3.1.1 and Sections 3.2-3.4 will be put into an appendix. After re-running the model with pre-formed tracers (see Author's Response to Referees #2 and #3), the parts of the text that describe back-calculations to pre-formed nutrient from apparent oxygen utilization (see e.g. section 3.2, p. 10) and the regression model for preformed alkalinity (see section 3.3) can be removed entirely. This will be replaced by a short subsection (3.1.3), which describes the use of pre-formed tracers in the model. Note: Using pre-formed tracers rather than back-calculations causes only minor changes to the results, which indicates that the back-calculation method is robust in this case. However, for the sake of clarity and for shortening the manuscript, we change to using the pre-formed tracers. We thank the reviewer for the specific examples of how to improve the results section. We will make the suggested changes and work through the manuscript to find sentences and paragraphs that can be improved in a similar way. The results section

will be compacted and clarified. Text that concerns glacial-interglacial implications will be moved to Section 5.3.

Point 2: Figure 2 summarizes all results, but its current presentation makes it very difficult to distill anything more than the general magnitudes. I suggest (1) providing the labels of the sensitivity experiments and sorting them somehow, perhaps by the anticipated magnitude of total effects, from largest to smallest; (2) separating this figure in to 3-4 panels: biological, residual, solubility, and total (indicating on the total plot the largest contributor to the total change).

Response: We have attempted to make the figure clearer by (1) providing the labels of the sensitivity experiment (note: the acronyms in the labels have changed from the submitted manuscript, to acronyms that should be easier to remember); (2) separating the figure in to the suggested panels; and 3) by re-sorting the simulations. We have chosen to keep the results sorted in pairs of high/low adjustments of circulation parameters (denoted x2 for doubled and /2 for halved, of which the doubled are always listed on top), and made this separation into pairs clearer, in order to make it easy to see the expected range of carbon storage differences within the span of the chosen parameter values. We have changed the order so that all the SEs with changes to atmospheric parameters come first, and put wind stress (WS) on top of atmospheric heat diffusivity (AD) because the wind effect is stronger. Then come the changes to the ocean diapycnal (DD, "vertical") and isopycnal (ID, "horizontal") diffusivities. Finally come the combined simulations, also re-ordered to have the simulations with larger DeltaTC come first. Sorting them by anticipated magnitude of total effects appears to be less useful, since the values of the explored parameters do not cover the full range of extreme values that could potentially be used in climate simulations (see response to point 3). The new version of the figure is attached to this response.

Point 3: Pg 9 Methods: Are the ranges for vertical diffusivity, wind stress, horizontal diffusivity, etc that are used in the sensitivity experiments comparable to the range of values that are normally used to tune models? Would be useful to provide this

information here, so that the reader can assess whether your sensitivity experiments represent values that might normally be used.

Response: For this study, our intention is for the ensemble to be representative of a wide range of plausible ocean circulation states. The chosen parameter ranges correspond to a halving and doubling of the values used in the control simulation. Our chosen values are within the parameter space explored for a predecessor to the GENIE-model by Edwards and Marsh (2005), except the low wind stress simulation (see below). Similar parameter ranges are also explored for GENIE by e.g. Marsh et al. (2013). For the most part, our selected values are within the parameter ranges that generate the subset Edwards and Marsh (2005) refer to as low-error simulations. In the Bern3D model, with physics based on Edwards and Marsh (2005) and thus similar to GENIE, Müller et al. (2006) doubled the observed wind stress (W = 2) to get a more realistic gyre circulation. Marinov et al. (2008 a,b) used the Geophysical Fluid Dynamics Laboratory Modular Ocean Model version 3, which has the same default value for isopycnal diffusivity (1500 m2 s-1 ) as our model. Marinov et al. (2008 a,b) explore a range of 1000-2000 m2 s-1 (c.f. our range of 750-3000 m2 s-1). When comparing with models that have different available tuning parameters, diagnostic variables such as temperature, salinity and AMOC volume transport can indicate whether our achieved states are within the common tuning range for ocean circulation. We compare the temperature and salinity ranges in two selected grid points of the ensemble of pre-industrial control states (PIC) of PMPI2 and CMIP5/PMIP3 (IPCC; see Table B1) to the corresponding grid cell ranges of our equilibrium states SE1-SE12. In these selected grid cells, we cover a similar span in salinity and an equally broad range in temperatures as the PMIP-ensemble, though the temperatures in our ensemble are higher (range shifted by ∼1.5°C). According to Muglia and Schmittner (2015), the PMIP3 PIC AMOC range is 12.6-23.0 Sv (Table B1). If we exclude the combined simulation with halved wind stress and halved diapycnal (vertical) diffusivity (SE12, now denoted WS/2_DD/2), which has a very weak AMOC (2.0 Sv), the AMOC range for our equilibrium states is 8.3-18.0 Sv (Table B1). Thus, there is a difference of ∼8-9

Sv between highest and lowest value, which is also the case for the PMIP3 PIC:s, but our ensemble does not cover the two highest PMIP3 AMOC values. This validation of the chosen parameter changes and a correctly formatted version of Table B1 will be included in the updated manuscript.

Please also note the supplement to this comment:
https://www.biogeosciences-discuss.net/bg-2017-166/bg-2017-166-AC1-supplement.pdf

————————————————————

[Figure]

[Figure]

**Fig. 1.** Updated version of fig. 2

**Supplement:**

**Table B1** Diagnostic variables of the pre-industrial control states of PMIP2 and CMIP5/PMIP3 (temperature and salinity as read from Fig. 9.18. of WG1 in IPCC AR5 and AMOC as given in Table 1 in Muglia and Schmittner (2015)) compared to similar diagnostics for our ensemble *SE1*-SE12 and control state *PIES278*.

| Variables | PMIP2 + CMIP5/PMIP3 pre-industrial control states | Ödalen et al. cGENIE *SE1-SE12* and *PIES278* |
|---|---|---|
| Potential temperature (°C), N.Atl. | 2.9 – 6.4[1] | 4.6 – 8.0[2] |
| Potential temperature (°C), S.Atl. | -1.6 – 2.0[3] | -0.3 – 3.7[4] |
| Salinity, N.Atl. | 34.8 – 35.5[1] | 35.3 – 35.6[2] |
| Salinity, S.Atl. | 34.6 – 35.0[3] | 34.9 – 35.1[4] |
| AMOC (1 Sv = $10^6$ m$^3$ s$^{-1}$) | 12.64 – 23.02[5] | 2.0 – 18.0[6] |

[1] North Atlantic PMIP grid point: 55.5°N, 14.5°W, 2,184 m depth

[2] North Atlantic cGENIE closest corresponding grid cell: 51-56°N, 10-20°W, 1,738-2,100 m depth

[3] South Atlantic PMIP grid point: 50°S, 5°E, 3,636 m depth

[4] South Atlantic cGENIE closest corresponding grid cell: 46-51°S, 0-10°E, 3,008-3,576 m depth

[5] Muglia and Schmittner (2015), PMIP3 pre-industrial control ensemble AMOC at 25°N, average with interval of one standard deviation

[6] cGENIE maximum Atlantic overturning. The *SE* ensemble member with halved wind stress and low vertical (diapycnal) diffusivity has a collapsed AMOC circulation (2.0 Sv). The average of this variable for all other *SE*:s is 13.8 Sv (range 8.3 – 18.0 Sv).

---

## Author Comment (AC2) · 6 Oct 2017

We thank the referee for the supportive comments on choice of topic and methods. We appreciate the detailed and helpful comments, which will improve the manuscript in its revised form. Below we address the general and specific comments of the referee. The technical corrections will be made in the revised paper, if they are still applicable in the new version of the manuscript.

General comment 1) I assume that you don't have preformed nutrients (O2, PO4, DIC) written out in the model output? If you do, this eliminates the problem with O2dis, as you can calculate O2sat and then calculate the remineralised O2 and hence Csoft

explicitly. Additionally, preformed $O_2$ and $PO_4$ would be much more useful in the parameterisation for ALKpre.

Response to General comment 1: We did not have preformed tracers in the output used for the first version of the manuscript. However, for the new version we have re-run the simulations to get the output for pre-formed tracers Cpre, O2pre, PO4pre and ALKpre. This eliminates the problems mentioned by the referee. It also eliminates the need for the regression model for ALKpre.

General comment 2) Is it possible to change the circulation in the model in the Southern Ocean and not in the Atlantic or vice versa? The bipolar seesaw could theoretically induce changes in Cdis in the Southern Hemisphere but not in the Northern Hemisphere.

Response to General comment 2: It is possible to efficiently change Cdis in only one of the hemispheres e.g. by changing the wind stress manually in selected regions rather than re-scaling it globally. We have done preliminary experiments where we reduce the winds only over the Southern Ocean, and even though this mainly causes changes of Cdis in AABW, the magnitude of DeltaCdis (henceforth we replace Delta by D in this response) is similar to the results for changes in global winds. We have therefore chosen not to include these simulations in the study. Studies of the bipolar seesaw go beyond the scope of the present study.

General comment 3) If it's not too cumbersome, perhaps consider using descriptive abbreviations for the different SEs.

Response to General comment 3: We will include such abbreviations in the updated manuscript.

Specific comments (SC): - SC1) p. 3, lines 4-5: Could you specify the order of magnitude of the change in CO2?

Response SC1: The order of magnitude of the CO2 rise in deglacials, as observed in

ice cores (e.g. Petit et al., 1999), is ∼100 ppm. It is likely that circulation differences as large as in PMIP LGM states (see e.g. Otto-Bliesner et al., 2007) could cause differences in deglacial CO2 on the order of tens of ppm, if such experiments were run with equally different circulation states in cGENIE. However, the order of magnitude of the potential discrepancies is not explicitly mentioned by Zhang et al. in the cited paper, and we would not want to speculate about the response of the models they are evaluating in their study.

- SC2) p. 3, line 35: It's also important to note that these studies don't even have Cres due to infinitely fast gas exchange.

Response SC2: Marinov et al. (2008a,b) do not have Cdis in their main ensemble of simulations, because of the fast gas exchange in these simulations. They do, however, study the disequilibrium component separately for a subset of 3 simulations, where they state that they apply 'regular' gas exchange. In this study, we cover a wider range of simulations that include Cdis. In Kwon et al. (2011), they assume the change in Cdis to be negligible in the theoretical derivation and therefore run most of the simulations with very fast gas exchange. They do, however, also analyse some simulations with a 'normal' gas exchange coefficient, in order to validate the theoretical model for atmospheric pCO2, but they do not explicitly study the disequilibrium response. The component of Cres that consists of calculation errors should be present in all of these studies, regardless of whether they have artificially fast or normal gas exchange. E.g. in Kwon et al., the calculation errors from the theoretical calculations of the changes in Csoft and Ccarb will be grouped together with Csat, which is not calculated.

- SC3) p. 4, lines 3-5: I don't find this paragraph necessary.

Response SC3: The paragraph will be removed in the updated manuscript.

- SC4) p. 4, lines 28-29: I don't believe that this is correct, as the pumps can have opposing effects. It should be specified that the net effect of all of the pumps must be to redistribute carbon from the surface to the deep ocean.

Response SC4: We will rephrase the sentence as follows: 'If the net capacity of the carbon pumps to redistribute carbon from the surface to the deep ocean increased, this would act to decrease pCO2atm.

- SC5) p. 5, line 17: I think what you mean is that alkalinity is not set (or affected) by gas exchange, but referring to an "expected" value of Cpre is a little misleading.

Response SC5: Since the regression model for ALKpre is no longer used, the two sentences at the end of this paragraph (starting with 'Unlike CO2…') will be removed.

- SC6) p. 5, lines 31-32: Please include Martin (1990), Paleoceanography, as this is one of the central references for increased soft tissue pump efficiency during glacials.

Response SC6: We will add Martin (1990) in the updated manuscript.

- SC7) p. 7, lines 5-6: "and has a level of detail for the carbon system that made it particularly suitable for this study." This is very general; please specify why it is appropriate for this study (and/or why less complex models are not).

Response SC7: The referee asks particularly about less complex models. A very simple ocean carbon system model such as miniBLING (Galbraith et al., 2015), which only has P, DIC and O2, would not be suitable for this study, because we are using ALK in our calculations. However, we want to emphasise that it is not only a question of complexity being high enough. Too high complexity could also pose a problem, e.g. a highly complex ocean ecosystem model with several functional types would be more expensive to run in terms of computational cost and therefore less suitable for this study. cGENIE balances having enough complexity in the ocean carbon system with a suitable level of complexity in terms of ocean resolution.

The full sentence on p.7, lines 5-6: "cGENIE is higher in complexity than box models, but is still efficient enough to allow running a large ensemble to equilibrium for the carbon system, and has a level of detail for the carbon system that made it particularly suitable for this study." should be rephrased as follows: "cGENIE is higher in complexity than box models, but is still efficient enough to allow running a large ensemble to equilibrium for the carbon system. In terms of ocean carbon system tracers, the minimum required for this type of study is P, DIC, O2, ALK. cGENIE includes many additional tracers, out of which only some are used in this study. For example, particulate (POC) and dissolved organic carbon (DOC) are included in the calculations of model carbon inventory. Of particular importance for this study is the possibility to run with pre-formed tracers (Ppre, Cpre, O2pre, ALKpre)."

- SC8) p. 8, line 25: When referring to Cdis, it would be useful to cite Ito and Follows (2013), GBC.

Response SC8: The citation will be added in the updated manuscript.

- SC9) p. 10, lines 14-24: Another important reference is Ito et al. (2004), GRL.

Response SC9: The citation will be added in the updated manuscript.

- SC10) Section 3.3: Do you use the same parameterisation for preformed alkalinity in all simulations? Please specify the errors in Cres in the surface field.

Response SC10: Yes, for comparability we were using the same parametrisation. Since we are now running with pre-formed tracers (including ALKpre), this question is no longer relevant for the updated manuscript.

- SC11) p. 12, lines 17-22: Again, please specify the size of the error introduced by making this approximation.

Response SC11: First, we want to emphasise that the dissociation constants used to determine CO2 solubility in the model (Mehrbach et al., 1973) are only defined for temperatures between 2–35 °C. Since this restriction is applied in the model, we also apply it in the calculation of Csat throughout the manuscript. To make estimates of the error caused by the restriction, we must assume that the dissociation constants are valid for all temperatures and then make a new calculation of Csat without the restriction (henceforth we call this fullCsat). This assumption inevitably makes the error

estimates approximate. However, the dissociation constants determined by Goyet and Poisson (1989) for artificial seawater between -1–40°C are similar to the constants given by Mehrbach et al. (1973). Thus, the assumption that the dissociation constants are applicable for a wider range of temperatures appears to be partly valid. When we calculate fullCsat, we find that this is larger than Csat in all ensemble members, by between 0.06 and 0.6% (clarification of line 20). When use the inventories of fullCsat to calculate DfullCsat (between the SEs and PIES278), we find that the contribution by temperature changes to DTC is on average underestimated by approximately 33 +/- 36% when the temperature restriction is applied. This hence strengthens our argument in Section 5.1. that the effects of the changes in the solubility pump should not be disregarded.

The paragraph on p. 12, lines 17-22 will be rephrased as follows: "The dissociation constants used in the cGENIE calculations of solubility for $CO_2$ in sea water follow Mehrbach et al. (1973), which are only defined for waters between 2–35 °C. Hence, the expression for $CO_2$ solubility in the model is restricted so that all water below 2 °C has the same $CO_2$ solubility (similarily for all water above 35°C). In the calculations of Csat, we use CO2SYS with this temperature restriction, to accurately represent the model behaviour. In order to estimate the error introduced by this restriction, we need to assume that the same dissociation constants can be used outside the given temperature interval. The validity of this assumption is supported by the results of Goyet and Poisson (1989), who find similar dissociation constants for the interval -1–40°C in a study on artificial seawater. When CO2SYS is run using model ocean temperatures without the temperature restriction, we find that the calculated inventory of Csat in the SEs is 0.06-0.6% larger than with the restriction. For PIES278 the inventory of Csat is 0.25% larger. In terms of DCsat, the unrestricted Csat inventories indicate that the contribution by temperature changes to DTC is on average underestimated by approximately 33 +/- 36% when the restriction is active. Since the restriction is used consistently, the error caused by the restriction being present in the model should not constitute a significant problem for our analysis. Nonetheless, the underestimation of

the effect of temperature changes should be heeded in the discussion of our results."

On p. 19, line 10, after the sentence ending with "…the dominant response", we add: "Due to the temperature restriction on the $CO_2$ solubility constants (see Section 3.4), the effect of DCsat is likely to be underestimated by on average 33 +-/36 % in our results, further emphasising its importance."

- SC12) p. 14, lines 6-7: Isn't this relationship true per definition of Csoft?

Response SC12: The text on line 6-7 on p. 14 does not mention Csoft. We assume the referee is in fact referring to the sentence starting on p. 15, line 6-7, which mentions the linear relationship between Csoft and P*. It is true that this linear relationship results from the definitions of Csoft and P*, which is also mentioned in the next sentence, starting with "P* is a direct measure…" on p. 15, line 7.

- SC13) p. 15, lines 29-31: Please describe here the importance of the temperature limit on the calculation of Csat, if this is on a comparable order of magnitude.

Response SC13: On line 28, after sentence ending "… larger that DCsoft.", we add: "In addition, the temperature restriction on the dissociation constants (see Section 3.4) is likely to cause DCsat to be underestimated by on average 33 +/- 36% in our ensemble." Lines 29-31 are moved to Section 5.3, implications for glacial studies. After the sentence ending "…were likely larger than in our set of experiments.", we add: "Fig. 7 suggests that DCsat for a change in Tavg of -2.6°C c.f. pre-industrial would be approximately 2.6 * 1e16 mol (310 GtC), whereas DCsat for the coldest of our simulations is only 1.3 * 1e16 mol (160 GtC). If we account for a likely underestimation of DCsat of 30% (see Section 3.4) in Fig. 7, a simulation as cold as the LGM state suggested by Headly and Severinghaus (2007) would have an increase in strength of the solubility pump corresponding to ~400 GtC.

- SC14) p. 17, lines 13-14: Have you done experiments to specifically examined he role of sea ice in determining Cdis? Quantitative results would be very interesting!

Response SC14: We have done experiments to examine the differences in Cdis between simulations in a subset of the ensemble (as described in the manuscript). There is some sea ice output available from these simulations, but we have chosen not to include a deeper analysis of this output in this already long manuscript. We have not made separate simulations where we e.g. only vary the extent of the sea ice, but this could potentially be done for a future study.

- SC15) p. 17, line 15: Only 0.01%? This seems to be at odds with Fig. 2.

Response SC15: The number should be 0.1% (the model carbon inventory is approximately 3* 1e18 mol and DeltaCdis is on the order of 1e16 mol). The calculation giving the number 0.01% used the difference between runs with normal and artificially fast gas exchange, which underestimated the signal of Cdis. We thank the referee for finding this error. The estimates of Fig. 2 agree better with the calculation of Cdis = Cpre – Csat, resulting from the new runs with pre-formed tracers (see new Fig. 9).

- SC16) p. 18, line 15: Changes in the solubility due to ocean temperature changes don't seem "indirect"

Response SC16: By indirect we mean secondary, as in a response that is the result of some other change. In this case, the primary change is to pCO2atm as the result of increased biological efficiency. As a response to the lower CO2, climate changes in terms of changed ocean circulation and ocean temperature occur. There is then a secondary, or rather additional, response of the ocean carbon system to these changes. We will rephrase the sentence as follows: "There are also additional effects on pCOatm due to changes in ocean temperature caused by changes in radiative balance, circulation and disequilibrium.

- SC17) p. 19, lines 31-32: Ito and Follows (2013), GBC also uses the same scheme to look specifically at this; please include this.

Response SC17: The citation will be added in the updated manuscript.

- SC18) p. 20, lines 23-24: Please specify what you mean by "an LGM-like circulation" and add appropriate citations

Response SC18: By an LGM-like circulation we mainly mean that the boundary between North Atlantic Deep Water (NADW) and Antarctic Bottom Water (AABW) was substantially shallower during the LGM than today. This circulation pattern is supported by paleonutrient tracers (reviewed by Marchitto and Broecker, 2006). This will be added to the updated manuscript.

- SC19) p. 22, line 27: Please cite the statement that "there may have been more, not less, preformed nutrients in the deep ocean during the last glacial"

Response SC19: The reference is Homola et al. (2015), but the reference is located in the wrong part of the sentence.

- SC20) p. 24, line 6: please specify if you mean the soft tissue pump and/or the carbonate pump

Response SC20: We mean both pumps. There are alkalinity corrections that are associated with the carbonate system as well as with the formation and destruction of organic matter (related to the nitrogen cycle).

- SC21) Fig. 8: Perhaps difference sections would be more useful?

Response SC21: We wish to also show the structure of the water mass in the different states. Hence, we suggest adding difference sections as supplementary material, showing how SE5 and SE6 deviate from PIES278.

- SC22) Fig. 9: It would be more illustrative to zoom in with the colourbar.

Response SC22: This figure will be updated to show Cpre – Csat, which improves the estimate of Cdis and shows that the previous method (calculating the difference between runs with normal and artificially fast gas exchange) did not fully reveal Cdis. See new, updated figure, which is attached to this response. When pre-formed tracers

are available, the simulations with unrealistically fast gas exchange are no longer used for the analysis. Note that the figure in the updated manuscript will be larger than displayed in this comment.

- SC23) Figures: Please make the font size larger, particularly in Fig. 7-10

Response SC23: This will be corrected in the updated manuscript.

- SC24) Table 3: Please give units for DC; why not include DCcarb?

Response SC24: The table shows correlation coefficients between circulation strength and DC, not values of DC in different ocean basins. The order of magnitude of DC could be specified in the table caption. DCcarb will be included in the updated manuscript. The first sentence of the table caption will be rephrased as follows: "Correlations between the changes in carbon species and the changes in strength of the zonal average overturning streamfunction (PSImax - PSImin) below 556 m depth in different geographical regions."

Technical corrections: All technical corrections that are still relevant for the updated manuscript will be made.

References

Galbraith, E. D., Dunne, J. P., Gnanadesikan, A., Slater, R. D., Sarmiento, J. L., Dufour, C. O., de Souza, G. F., Bianchi, D., Claret, M., Rodgers, K. B., Marvasti, S. S.: Complex functionality with minimal computation: Promise and pitfalls of reduced-tracer ocean biogeochemistry models, J. Adv. Model. Earth Syst., 7, 2012–2028, doi:10.1002/2015MS000463, 2015.

Goyet, C., and Poisson, A.: New determination of carbonic acid dissociation constants in seawater as a function of temperature and salinity, Deep Sea Res., 36(11), 1635-1654, 1989.

Marchitto, T. M., and Broecker, W. S.: Deep water mass geometry in the glacial Atlantic Ocean: A review of constraints from the paleonutrient proxy Cd/Ca, Geochem. Geophys. Geosyst., 7, Q12003, doi:10.1029/2006GC001323, 2006.

Mehrbach, C., Culberson, C. H., Hawley, J. E., and Pytkowicx, R. M.: Measurement of the apparent dissociation constants of carbonic acid in seawater at atmospheric pressure, Limnol. Oceanogr., 18(6), 897-907, 1973.

[Figure]

[Figure]

**Fig. 1.** Updated version of Fig. 9

---

## Author Response (AR1)

**Author's response**

Referee comments and submitted author's responses are listed in black. Corresponding manuscript updates are marked in red.

-- Abstract, Introduction and Discussion: The conclusion that the oceanic storage of DIC and the drawdown of atmospheric CO2 in response to nutrient depletion all depend on the ocean circulation patterns (including the overturning strengths of NADW and AABW, and the volume fraction of the ocean last ventilated from the North Atlantic vs. Southern Ocean) is not new. The circulation effects on the ocean carbon pumps have been extensively studied using models and theoretical frameworks: the solubility pump (e.g., DeVries and Primeau, Atmospheric pCO2 sensitivity to the solubility pump: Role of the low-latitude ocean, GBC 2009), the biological pump (references already cited in the manuscript) and the disequilibrium pump (e.g. Ito and Follows, Air-sea disequilibrium of carbon dioxide enhances the biological carbon sequestration in the Southern Ocean, GBC 2013). One of the new points in this study is that the authors discussed the relative role of the three pumps in the net change of the simulated carbon cycle. But, because the forcing used to generate the different circulation patterns is arbitrary, their discussion of the relative roles does not seem very interesting. Overall, I feel that the authors need to highlight what new findings or insights this study can provide.

*Response:* In the study by Ito and Follows (GBC, 2013), mentioned by the referee, it was pointed out that more studies like these, where we analyse model output 'in terms of different carbon pump components' and their behaviour could aid in understanding the behaviour of climate models and carbon cycle models. The reviewer is correct that the circulation effect on the ocean carbon pump has been extensively investigated. There are two main advances here that are of importance to the community, and we will place greater emphasis on these in the revised manuscript:

1) The initial state, not only in terms of ocean circulation, but in terms of ocean carbon inventory and the origin of the already stored carbon (e.g. biological or solubility pump), is crucial for the outcome of a study investigating increased efficiency of the biological pump, and the effect of the initial state is quantifiable;

2) the relative contributions by Csat, Csoft, Ccarb, and Cdis to the ocean carbon inventory in the initial state depends on the model tuning strategy, yet these combine in a manner to give a response in total carbon that is more straightforwardly related to the circulation than any of the four components (Figs. 3-4).

*Regarding 1)* We show that differences in the initial states can explain discrepancies in e.g. a CO2 drawdown scenario where the efficiency of the biological pump increases. As the referee points out, this will help to explain why previous model intercomparison studies have seen a large spread in response among models to the same perturbation. Therefore, it can also be important for future model intercomparison studies, in explanations of results, but also in planning for common tuning strategies and experimental design. This will be important in model studies of both glacial and modern climate scenarios. Here, the findings in described in 2) make it easier to understand the outcome of the drawdown experiment and therefore provide a useful basis for explaining the role of the initial state. The findings presented in e.g. Fig. 6 are new, and have not been covered by the studies mentioned by the referee. Of most value here is that we show that it is possible to quantify, from theory, the effect of any bias in the model's initial state on its sensitivity to changes in the biological pump. We have chosen a variety of changes to the forcing in order to demonstrate that the result is robust. This result should be of value in understanding the biases individual models, in inter-comparison studies, and potentially for choosing tuning criteria.

*Regarding 2)* Firstly, the changes to the forcing that we make are certainly idealised, but they are not arbitrary. We are changing common circulation parameters, mainly within the limits of common tuning ranges, to produce circulation states that are relevant in the context of the amplitude of (palaeo-)climate change (see response to Referee #1, point 3). (The change used to produce a 100%-efficient biological pump is, of course, an intentional exception.) As we explore both an increase and a decrease of the same parameter, the equilibrium states resulting from the increase and decrease, together with the control equilibrium state, provide a range of states that can be expected from tuning of this parameter. By testing several parameters in the same manner, we show that different tuning strategies are expected to affect different reservoirs of the initial state carbon inventory. This leads us to the important conclusion that, depending on the strategy used in model tuning, the relative sizes of the reservoirs of DIC species will differ between models. As the referee points out, this has not been done in previous studies. As a consequence, such differences have not been taken into account in e.g. model intercomparison studies. We hope that our study can bring some attention to this problem.

Secondly, when comparing the different changes in forcing, our main intent is for the reader to compare the relative sizes of change of the different pumps when one parameter is changed (e.g. when the wind stress is changed, the most important contribution to the change in carbon inventory comes from the soft tissue pump, whereas if horizontal (isopycnal) diffusivity is changed, the contribution from the solubility pump and disequilibrium carbon dominate). Hence, the magnitude is not what is most important, but rather which carbon pump(s) is(are) affected by which tuning strategy. By doing this, we emphasize that the tuning strategy is important for determining which processes contribute to a model's ocean carbon inventory, and therefore its behaviour. Even if this conclusion is not entirely new, all the previous studies focus on one or two processes at a time. We therefore believe that the structured approach of the ensemble study, and the fact that all DIC species are investigated in the same manuscript, can provide a comparability that can be useful for the community.

→ We do, however, acknowledge the reviewer's point that these insights need to be better highlighted, and will make appropriate changes to the Abstract, Introduction and Discussion sections.
→ For the corresponding changes in the manuscript, see List of relevant changes and marked up manuscript comparison

-- Methods and Results: *General comment 1)* I assume that you don't have preformed nutrients (O2, PO4, DIC) written out in the model output? If you do, this eliminates the problem with O2dis, as you can calculate O2sat and then calculate the remineralised O2 and hence Csoft explicitly. Additionally, preformed O2 and PO4 would be much more useful in the parameterisation for ALKpre.
*Response:* We did not have preformed tracers in the output used for the first version of the manuscript. However, for the new version we have re-run the simulations to get the output for pre-formed tracers Cpre, O2pre, PO4pre and ALKpre. This eliminates the problems mentioned by the referee. It also eliminates the need for the regression model for ALKpre.
→ For the corresponding changes in the manuscript, see List of relevant changes and marked up manuscript comparison

-- All sections: *General comment 3)* If it's not too cumbersome, perhaps consider using descriptive abbreviations for the different SEs.
*Response:* We will include such abbreviations in the updated manuscript.
→ For the corresponding changes in the manuscript, see List of relevant changes and marked up manuscript comparison

-- p. 3, lines 4-5: Could you specify the order of magnitude of the change in CO2?
*Response:* The order of magnitude of the CO2 rise in deglacials, as observed in ice cores (e.g. Petit et al., 1999), is ~100 ppm. It is likely that circulation differences as large as in PMIP

LGM states (see e.g. Otto-Bliesner et al., 2007) could cause differences in deglacial CO2 on the order of tens of ppm, if such experiments were run with equally different circulation states in cGENIE. However, the order of magnitude of the potential discrepancies is not explicitly mentioned by Zhang et al. in the cited paper, and we would not want to speculate about the response of the models they are evaluating in their study.

→ no change has been made to the manuscript

-- p. 3, line 35: It's also important to note that these studies don't even have Cres due to infinitely fast gas exchange.

*Response:* Marinov et al. (2008a,b) do not have Cdis in their main ensemble of simulations, because of the fast gas exchange in these simulations. They do, however, study the disequilibrium component separately for a subset of 3 simulations, where they state that they apply 'regular' gas exchange. In this study, we cover a wider range of simulations that include Cdis. In Kwon et al. (2011), they assume the change in Cdis to be negligible in the theoretical derivation and therefore run most of the simulations with very fast gas exchange. They do, however, also analyse some simulations with a 'normal' gas exchange coefficient, in order to validate the theoretical model for atmospheric pCO2, but they do not explicitly study the disequilibrium response. The component of Cres that consists of calculation errors should be present in all of these studies, regardless of whether they have artificially fast or normal gas exchange. E.g. in Kwon et al., the calculation errors from the theoretical calculations of the changes in Csoft and Ccarb will be grouped together with Csat, which is not calculated.

→ For the corresponding changes in the manuscript, see marked up manuscript comparison

-- p. 4, lines 3-5: I don't find this paragraph necessary.

*Response:* The paragraph will be removed in the updated manuscript.

→ removed

* as much as possible of Section 2, Section 3.1.1 and Sections 3.2-3.4 will be put into an appendix. After re-running the model with pre-formed tracers (see Author's Response to Referees #2 and #3), the parts of the text that describe back-calculations to pre-formed nutrient from apparent oxygen utilization (see e.g. section 3.2, p. 10) and the regression model for preformed alkalinity (see section 3.3) can be removed entirely. This will be replaced by a short subsection (3.1.3), which describes the use of pre-formed tracers in the model.

→ See Appendices A and B

-- p. 4, lines 28-29: I don't believe that this is correct, as the pumps can have opposing effects. It should be specified that the net effect of all of the pumps must be to redistribute carbon from the surface to the deep ocean.

*Response:* We will rephrase the sentence as follows: 'If the net capacity of the carbon pumps to redistribute carbon from the surface to the deep ocean increased, this would act to decrease pCO2atm.

→ changed

-- p. 5, line 17: I think what you mean is that alkalinity is not set (or affected) by gas exchange, but referring to an "expected" value of Cpre is a little misleading.

*Response:* Since the regression model for ALKpre is no longer used, the two sentences at the end of this paragraph (starting with 'Unlike CO2…') will be removed.

→ removed

-- p. 5, lines 31-32: Please include Martin (1990), Paleoceanography, as this is one of the central references for increased soft tissue pump efficiency during glacials.

*Response:* We will add Martin (1990) in the updated manuscript.
→ added

*** Methods: The way the biological pump is simulated in the model is unclear. The authors included the carbonate pump in their models and analyses (expressed as Ccarb), but there is no description on how the carbonate pump is represented in the model. For example, are the sedimentation processes included in the model? How are the production and dissolution of calcifying organisms represented in the model? How is the strength/efficiency of the carbonate pump affected by the drawdown experiment? In the drawdown experiment (specified in lines #31-33 of page #9), the remineralization length scale of sinking organic particles is made very deep (10,000m), so that "any carbon that is taken up in organic material to be highly efficiently trapped in the deep ocean and not undergo any significant remineralization". Does it mean that most of inorganic nutrient is converted to organic form and stored in the abyss without being remineralized back to inorganic form? If this is the case, then the amount of organic matter would increase substantially in the drawdown experiment, and the carbon fixed in organic material should be an important component in the mass balance equations and can't be ignored in the theoretical derivations presented in the manuscript. Likewise, the equation "Ppre=P-Preg" would be incorrect. This needs to be clarified.

    *Response:* The way the biological pump is simulated in the model is thoroughly described by Ridgwell et al. (2007). Since the description is lengthy and the paper is already long, we decided not to include the full description in the manuscript, but instead describe the most relevant part describing remineralisation (section 3.1.2) and reference the paper for the in-depth model description. We agree with the referee that it should be clarified that the explicit information on the biogeochemical cycling can be found in this paper (see manuscript changes below).

We also agree that we need to make some clarifications regarding the drawdown experiment. The referee asks 'Does it mean that most of inorganic nutrient is converted to organic form and stored in the abyss without being remineralized back to inorganic form?' Yes, that is exactly the case, and the referee correctly states that this means the amount of organic matter in the deep ocean significantly increases. The referee is also correct in saying that the theoretical derivation that neglects the organic matter no longer holds. Therefore, we do not use this derivation for the drawdown experiment (we only use it in the first step, where we compare the initial equilibrium states). For the drawdown experiment, we do not explicitly calculate the change in strength of the soft tissue pump, the carbonate pump, or the changes in disequilibrium. The theoretical derivation could be made valid for this case by assuming that 106 times the amount of particulate P in the deep ocean should be added to Csoft. However, we have chosen not to explore the changes in strength of the different pumps in the drawdown experiment, since it is a highly hypothetical case. We are mainly interested in the differences in CO2 drawdown between different ensemble members in the limit of a highly efficient biological pump. That the biological pump is made highly efficient is clear from the two orders of magnitude decrease in dissolved P in the ocean, which will also be clarified in the manuscript.

→ We will add a clarification in section 3.1, at the end of the second paragraph. This clarification reads "The model description of export flux of organic matter is based on available surface nutrients (see Ridgwell et al., 2007, Eq. 1-4), and instead of having a "standing plankton biomass" in the model, the export of particulate organic matter is derived directly from uptake of P. The carbonate precipitation rate is thermodynamically-based and relates export flux of CaCO3 to the flux of POC (see Ridgwell et al., 2007, Eq. 8). As investigation of carbonate system feedbacks are not the purpose of this study, interactive sediments are not used, and in terms of carbon cycling, the atmosphere-ocean is studied as a closed system."

→ We will also add to section 3.1.2, before the final sentence of the second paragraph, the sentence 'The remineralisation of CaCO3 in the water column is treated in a similar manner to particulate organic carbon."

→ Note that in the updated manuscript, these updates are all in Section 3.1, since Section 3.1.2 has been removed and incorporated into 3.1 (see List of relevant changes)

-- p. 7, lines 5-6: "and has a level of detail for the carbon system that made it particularly suitable for this study." This is very general; please specify why it is appropriate for this study (and/or why less complex models are not).

*Response:* The referee asks particularly about less complex models. A very simple ocean carbon system model such as miniBLING (Galbraith et al., 2015), which only has P, DIC and O2, would not be suitable for this study, because we are using ALK in our calculations. However, we want to emphasise that it is not only a question of complexity being high enough. Too high complexity could also pose a problem, e.g. a highly complex ocean ecosystem model with several functional types would be more expensive to run in terms of computational cost and therefore less suitable for this study. cGENIE balances having enough complexity in the ocean carbon system with a suitable level of complexity in terms of ocean resolution.

The full sentence on p.7, lines 5-6: "cGENIE is higher in complexity than box models, but is still efficient enough to allow running a large ensemble to equilibrium for the carbon system, and has a level of detail for the carbon system that made it particularly suitable for this study." should be rephrased as follows: "cGENIE is higher in complexity than box models, but is still efficient enough to allow running a large ensemble to equilibrium for the carbon system. In terms of ocean carbon system tracers, the minimum required for this type of study is P, DIC, O2, ALK. cGENIE includes many additional tracers, out of which only some are used in this study. For example, particulate (POC) and dissolved organic carbon (DOC) are included in the calculations of model carbon inventory. Of particular importance for this study is the possibility to run with pre-formed tracers (Ppre, Cpre, O2pre, ALKpre)."

-- p. 8, line 25: When referring to Cdis, it would be useful to cite Ito and Follows (2013), GBC.

*Response:* The citation will be added in the updated manuscript.
→ citation added in the Introduction

* Pg 9 Methods: Are the ranges for vertical diffusivity, wind stress, horizontal diffusivity, etc that are used in the sensitivity experiments comparable to the range of values that are normally used to tune models? Would be useful to provide this information here, so that the reader can assess whether your sensitivity experiments represent values that might normally be used.

*Response:* For this study, our intention is for the ensemble to be representative of a wide range of plausible ocean circulation states. The chosen parameter ranges correspond to a halving and doubling of the values used in the control simulation. Our chosen values are within the parameter space explored for a predecessor to the GENIE-model by Edwards and Marsh (2005), except the low wind stress simulation (see below). Similar parameter ranges are also explored for GENIE by e.g. Marsh et al. (2013). For the most part, our selected values are within the parameter ranges that generate the subset Edwards and Marsh (2005) refer to as low-error simulations.

In the Bern3D model, with physics based on Edwards and Marsh (2005) and thus similar to GENIE, Müller et al. (2006) doubled the observed wind stress ($W = 2$) to get a more realistic gyre circulation. Marinov et al. (2008 a,b) used the Geophysical Fluid Dynamics Laboratory Modular Ocean Model version 3, which has the same default value for isopycnal diffusivity (1500 $m^2$ $s^{-1}$ ) as our model. Marinov et al. (2008 a,b) explore a range of 1000-2000 $m^2$ $s^{-1}$ (c.f. our range of 750-3000 $m^2$ $s^{-1}$). When comparing with models that have different available tuning parameters, diagnostic variables such as temperature, salinity and AMOC volume transport can indicate whether our achieved states are within the common tuning range for ocean circulation.

We compare the temperature and salinity ranges in two selected grid points of the ensemble of pre-industrial control states (*PIC*) of PMPI2 and CMIP5/PMIP3 (IPCC; see Table B1) to

the corresponding grid cell ranges of our equilibrium states *SE1-SE12*. In these selected grid cells, we cover a similar span in salinity and an equally broad range in temperatures as the PMIP-ensemble, though the temperatures in our ensemble are higher (range shifted by ~1.5°C). According to Muglia and Schmittner (2015), the PMIP3 *PIC* AMOC range is 12.6-23.0 Sv (Table B1). If we exclude the combined simulation with halved wind stress and halved diapycnal (vertical) diffusivity (*SE12,* now denoted *WS/2_DD/2)*, which has a very weak AMOC (2.0 Sv), the AMOC range for our equilibrium states is 8.3-18.0 Sv (Table B1). Thus, there is a difference of ~8-9 Sv between highest and lowest value, which is also the case for the PMIP3 *PIC:s*, but our ensemble does not cover the two highest PMIP3 AMOC values.

**Table B1** Diagnostic variables of the pre-industrial control states of PMIP2 and CMIP5/PMIP3 (temperature and salinity as read from Fig. 9.18. of WG1 in IPCC AR5 and AMOC as given in Table 1 in Muglia and Schmittner (2015)) compared to similar diagnostics for our ensemble *SE1*-SE12 and control state *PIES278*.

| Variables | PMIP2 + CMIP5/PMIP3 pre-industrial control states | Ödalen et al. cGENIE *SE1-SE12* and *PIES278* |
|---|---|---|
| Potential temperature (°C), N.Atl. | 2.9 – 6.4[1] | 4.6 – 8.0[2] |
| Potential temperature (°C), S.Atl. | -1.6 – 2.0[3] | -0.3 – 3.7[4] |
| Salinity, N.Atl. | 34.8 – 35.5[1] | 35.3 – 35.6[2] |
| Salinity, S.Atl. | 34.6 – 35.0[3] | 34.9 – 35.1[4] |
| AMOC (1 Sv = $10^6$ m$^3$ s$^{-1}$) | 12.64 – 23.02[5] | 2.0 – 18.0[6] |

[1] North Atlantic PMIP grid point: 55.5°N, 14.5°W, 2,184 m depth

[2] North Atlantic cGENIE closest corresponding grid cell: 51-56°N, 10-20°W, 1,738-2,100 m depth

[3] South Atlantic PMIP grid point: 50°S, 5°E, 3,636 m depth

[4] South Atlantic cGENIE closest corresponding grid cell: 46-51°S, 0-10°E, 3,008-3,576 m depth

[5] Muglia and Schmittner (2015), PMIP3 pre-industrial control ensemble AMOC at 25°N, average with interval of one standard deviation

[6] cGENIE maximum Atlantic overturning. The *SE* ensemble member with halved wind stress and low vertical (diapycnal) diffusivity has a collapsed AMOC circulation (2.0 Sv). The average of this variable for all other *SE*:s is 13.8 Sv (range 8.3 – 18.0 Sv).

→This full response has been added as a new subsection "Sensitivity experiment tuning parameters"

\* Pg. 9 LN 13 – confusing – do you mean that you hold ALK and P constant in your experiments?
*Response not given – corrected directly in manuscript*
→ Clarification given: During the spin-up phase, pCO$_2^{atm}$ is still restored to 278 ppm (Fig. 1) and the ocean reservoir of nutrients (in this case, PO$_4$) is the same as *PIES278*, though nutrients are redistributed. $A_T$ has no external sources/sinks and is only affected by the biological pump, which means it is negligibly different between ensemble members.
.

\* Pg 9 LN 14-20 – Upon first reading, it was unclear what delta (ïA¿Dˇ ) represents. Please define specifically that you are comparing the carbon estimates from PIES278 with equilibrium values from SE(n).
*Response not given – correct directly in manuscript*
→ Clarified by changing the sentence at Pg. 9 LN19, which introduces the equation, to: "The change in the total carbon inventory, $\Delta{TC}$ [mol] between $PIES278$ and some equilibrium state $SE(n)$, where $n = (1, ..., 12)$, can be described by"

* Pg 9 and then again on Pg 10 – when you describe the experiments in which you have implemented artificially fast gas exchange to remove Cdis, please identify this experiment with its number listed in Table 1.

*Response not given – correct directly in manuscript*

→ The experiments with artificially fast gas exchange are no longer used, so this has been removed from the manuscript.

-- p. 10, lines 14-24: Another important reference is Ito et al. (2004), GRL.

*Response:* The citation will be added in the updated manuscript.

→ Reference added in the manuscript in the section about preformed nutrients, where the issue of oxygen disequilibrium is mentioned.

-- Section 3.3: Do you use the same parameterisation for preformed alkalinity in all simulations? Please specify the errors in Cres in the surface field.

*Response:* Yes, for comparability we were using the same parametrisation. Since we are now running with pre-formed tracers (including ALKpre), this question is no longer relevant for the updated manuscript.

→ Deleted

-- p. 12, lines 17-22: Again, please specify the size of the error introduced by making this approximation.

*Response:* First, we want to emphasise that the dissociation constants used to determine $CO_2$ solubility in the model (Mehrbach et al., 1973) are only defined for temperatures between 2–35 °C. Since this restriction is applied in the model, we also apply it in the calculation of Csat throughout the manuscript. To make estimates of the error caused by the restriction, we must assume that the dissociation constants are valid for all temperatures and then make a new calculation of Csat without the restriction (henceforth we call this fullCsat). This assumption inevitably makes the error estimates approximate. However, the dissociation constants determined by Goyet and Poisson (1989) for artificial seawater between -1–40°C are similar to the constants given by Mehrbach et al. (1973). Thus, the assumption that the dissociation constants are applicable for a wider range of temperatures appears to be partly valid. When we calculate fullCsat, we find that this is larger than Csat in all ensemble members, by between 0.06 and 0.6% (clarification of line 20). When use the inventories of fullCsat to calculate DfullCsat (between the SEs and PIES278), we find that the contribution by temperature changes to DTC is on average underestimated by approximately 33 +/- 36% when the temperature restriction is applied. This hence strengthens our argument in Section 5.1. that the effects of the changes in the solubility pump should not be disregarded.

The paragraph on p. 12, lines 17-22 will be rephrased as follows:

"The dissociation constants used in the cGENIE calculations of solubility for $CO_2$ in sea water follow Mehrbach et al. (1973), which are only defined for waters between 2–35 °C. Hence, the expression for $CO_2$ solubility in the model is restricted so that all water below 2 °C has the same $CO_2$ solubility (similarly for all water above 35°C). In the calculations of Csat, we use CO2SYS with this temperature restriction, to accurately represent the model behaviour. In order to estimate the error introduced by this restriction, we need to assume that the same dissociation constants can be used outside the given temperature interval. The validity of this assumption is supported by the results of Goyet and Poisson (1989), who find similar dissociation constants for the interval -1–40°C in a study on artificial seawater. When CO2SYS is run using model ocean temperatures without the temperature restriction, we find that the calculated inventory of Csat in the SEs is 0.06-0.6% larger than with the restriction. For PIES278 the inventory of Csat is 0.25% larger. In terms of DCsat, the unrestricted Csat inventories indicate that the contribution by temperature changes to DTC is on average underestimated by approximately 33 +/- 36% when the restriction is active. Since the restriction is used consistently, the error caused by the restriction being present in the model should not constitute a significant problem for our analysis. Nonetheless, the underestimation

of the effect of temperature changes should be heeded in the discussion of our results."

On p. 19, line 10, after the sentence ending with "…the dominant response", we add: "Due to the temperature restriction on the CO2 solubility constants (see Section 3.4), the effect of DCsat is likely to be underestimated by on average 33 +-/36 % in our results, further emphasising its importance."

→ added to Appendix B and Discussion

* Results section: The results section is similarly very wordy; it mixes methods, results, and discussion together; many points that take multiple paragraphs to make could be simplified to 1-2 sentences. I suggest a thorough attempt to go through this paper and streamline the writing.

→ We have made thorough changes to the Results section and separated methods, results and discussion (see List of relevant changes and marked up manuscript)

* The entire top of page 13 describes how figure 2 will be put together, with only two half-sentences (regarding ensemble range of pH and pCO2atm) that describe results.

→ replaced by actual results (ranges)

-- p. 14, lines 6-7: Isn't this relationship true per definition of Csoft?

*Response:* The text on line 6-7 on p. 14 does not mention Csoft. We assume the referee is in fact referring to the sentence starting on p. 15, line 6-7, which mentions the linear relationship between Csoft and P*. It is true that this linear relationship results from the definitions of Csoft and P*, which is also mentioned in the next sentence, starting with "P* is a direct measure…" on p. 15, line 7.

→ Sentence partially rephrased to clarify that the linearity is expected.

* p. 14 LN 12-17 – this section does not describe any results presented.

→ moved to Methods

* p. 14 LN 27 – here and throughout this discussion, the authors indicate that horizontal diffusivity affects Csat and Cres more than Csoft, but the more obvious result is the minimal impact on deltaTC overall. This is worth noting.

*Response not given – correct directly in manuscript*

→ note added

* p. 15 LN 28 – "In ensemble members in which horizontal diffusivity in the ocean is changed, _Csat is larger than _Csoft." When horizontal diffusivity is reduced or increased? Specify that it is larger whether horizontal diffusivity is increased or decreased. It is near impossible from Figure 2 to discern this.

*Response not given – correct directly in manuscript*

→ The paragraph is re-written according to the results of the calculations using preformed tracers.

* p. 15 ln 29-37 – estimating the implications of the relationship between Csat and aveT for the glacial ocean is a point for discussion, not results

*Response not given – Correct directly in manuscript*

→ Moved to subsection 'Solubility pump and disequilibrium' under Discussion

-- p. 15, lines 29-31: Please describe here the importance of the temperature limit on the calculation of Csat, if this is on a comparable order of magnitude.

*Response:* On line 28, after sentence ending "… larger that DCsoft.", we add:

"In addition, the temperature restriction on the dissociation constants (see Section 3.4) is

likely to cause DCsat to be underestimated by on average 33 +/- 36% in our ensemble."
→ added

Lines 29-31 are moved to Section 5.3, implications for glacial studies. After the sentence ending "…were likely larger than in our set of experiments.", we add: "Fig. 7 suggests that DCsat for a change in Tavg of -2.6°C c.f. pre-industrial would be approximately 2.6 * 1e16 mol (310 GtC), whereas DCsat for the coldest of our simulations is only 1.3 * 1e16 mol (160 GtC). If we account for a likely underestimation of DCsat of 30% (see Section 3.4) in Fig. 7, a simulation as cold as the LGM state suggested by Headly and Severinghaus (2007) would have an increase in strength of the solubility pump corresponding to ~400 GtC.
→ moved and added

*p. 15 LN 18-31 – the average global temperature of simulations has a range of 2.3-4.9_C, which results in a range in delta pCO2atm from Csat of -16 – +17 ppm or -13 to +12ppm (depending upon how the calculation is made). → Suggested shortening of the paragraph by the referee.
→ The suggested shortening removes some vital information that we want to include, especially the range in Csat. Also, we want to emphasise the usefulness of the Goodwin et al. equations and would prefer to keep the specific mention of it. Therefore, we suggest the following shortened paragraph:
"Csat,T is the contribution to Csat from water temperature. Global average temperature, Tavg, of the equilibrium states has a range of 2.3–4.9 °C (see Table 3), resulting in an interval of change in Csat;T of $-0.8*10^{16}$ – $+0.6 *10^{16}$ mol (Fig. 2b, Fig. 6), or -96 – +72 GtC. Note that this includes a restriction on the solubility constants which prevents solubility from increasing with temperatures below 2°C, which weakens the close-to-linear relationship between Tavg and Csat;T (Fig. 6). This corresponds to a range in $pCO^2atm$ of about -7 – +11 ppm (Figs. 2f and S1) when we solve the carbon system equations Appendix C). The simplified equation (Eq. C9) suggested by Goodwin et al. (2011) yields results for _ pCO2 that in general are 10 – 20 % lower compared to using the carbon system equation solver (Fig. S1). Changes in Csat caused by changes in preformed alkalinity (Csat;Apre ) spans $-1.9 * 10^{16}$ – $1.8*10^{16}$ (Fig. 2b), which roughly corresponds to a range in $pCO^2atm$ of -21 – +21 ppm (Fig. 2f)."
which spans approximately 8 lines in the updated manuscript (c.f. 13 lines in the discussion paper).

-- p. 17, lines 13-14: Have you done experiments to specifically examined he role of sea ice in determining Cdis? Quantitative results would be very interesting!
*Response:* We have done experiments to examine the differences in Cdis between simulations in a subset of the ensemble (as described in the manuscript). There is some sea ice output available from these simulations, but we have chosen not to include a deeper analysis of this output in this already long manuscript. We have not made separate simulations where we e.g. only vary the extent of the sea ice, but this could potentially be done for a future study.
→ No change

-- p. 17, line 15: Only 0.01%? This seems to be at odds with Fig. 2.
*Response:* The number should be 0.1% (the model carbon inventory is approximately 3* 1e18 mol and DeltaCdis is on the order of 1e16 mol). The calculation giving the number 0.01% used the difference between runs with normal and artificially fast gas exchange, which underestimated the signal of Cdis. We thank the referee for finding this error. The estimates of Fig. 2 agree better with the calculation of Cdis = Cpre – Csat, resulting from the new runs with pre-formed tracers (see new Fig. 9).
→ Changed to 0.1%, figures replaced

-- p. 18, line 15: Changes in the solubility due to ocean temperature changes don't seem "indirect"

*Response:* By indirect we mean secondary, as in a response that is the result of some other change. In this case, the primary change is to pCO2atm as the result of increased biological efficiency. As a response to the lower CO2, climate changes in terms of changed ocean circulation and ocean temperature occur. There is then a secondary, or rather additional, response of the ocean carbon system to these changes.

We will rephrase the sentence as follows: "There are also additional effects on pCOatm due to changes in ocean temperature caused by changes in radiative balance, circulation and disequilibrium.

→ "indirect" replaced by "additional"

<DISCUSSION>

-- p. 19, lines 31-32: Ito and Follows (2013), GBC also uses the same scheme to look specifically at this; please include this.

*Response:* The citation will be added in the updated manuscript.

→ citation added and discussed

\* Pg 20 LN 17 – Technically the authors have not shown the role of "AMOC strength," which refers specifically to the Atlantic overturning limb. The plots calculate the difference between northern (Atlantic) and southern source components and thus combine the roles of the northern and southern overturning strengths. Changes in the southern source might mask changes in the AMOC alone. I suggest using a different phrase here.

→ Changed to overturning circulation strength, $OVT$, which is defined in the new Section 3.4 (see List of relevant changes)

-- p. 20, lines 23-24: Please specify what you mean by "an LGM-like circulation" and add appropriate citations

*Response:* By an LGM-like circulation we mainly mean that the boundary between North Atlantic Deep Water (NADW) and Antarctic Bottom Water (AABW) was substantially shallower during the LGM than today. This circulation pattern is supported by paleonutrient tracers (reviewed by Marchitto and Broecker, 2006). This will be added to the updated manuscript.

→ added description and citation

<CONCLUSIONS>

-- p. 22, line 27: Please cite the statement that "there may have been more, not less, preformed nutrients in the deep ocean during the last glacial"

*Response:* The reference is Homola et al. (2015), but the reference is located in the wrong part of the sentence.

→ This study is not yet published, but has been presented at AGU by Kira Homola and discussed in personal communication with Arthur Spivack. This has been clarified in the manuscript, where the sentence has been rewritten as: "However, on-going studies indicate that there may have been more, not less, preformed nutrients in the deep ocean during the last glacial \citep[][and Spivack, A. J. (P.C., 2015)]{HomolaEtAl2015}, which implies less efficient nutrient utilisation by biology."

<APPENDIX AND FIGURES>

-- p. 24, line 6: please specify if you mean the soft tissue pump and/or the carbonate pump

*Response:* We mean both pumps. There are alkalinity corrections that are associated with the carbonate system as well as with the formation and destruction of organic matter (related to

the nitrogen cycle).
→ clarification added

* Figure 2 summarizes all results, but its current presentation makes it very difficult to distill anything more than the general magnitudes. I suggest (1) providing the labels of the sensitivity experiments and sorting them somehow, perhaps by the anticipated magnitude of total effects, from largest to smallest; (2) separating this figure in to 3-4 panels: biological, residual, solubility, and total (indicating on the total plot the largest contributor to the total change).

*Response:* We have attempted to make the figure clearer by (1) providing the labels of the sensitivity experiment (note: the acronyms in the labels have changed from the submitted manuscript, to acronyms that should be easier to remember); (2) separating the figure in to the suggested panels; and 3) by re-sorting the simulations. We have chosen to keep the results sorted in pairs of high/low adjustments of circulation parameters (denoted x2 for doubled and /2 for halved, of which the doubled are always listed on top), and made this separation into pairs clearer, in order to make it easy to see the expected range of carbon storage differences within the span of the chosen parameter values. We have changed the order so that all the SEs with changes to atmospheric parameters come first, and put wind stress (WS) on top of atmospheric heat diffusivity (AD) because the wind effect is stronger. Then come the changes to the ocean diapycnal (DD, "vertical") and isopycnal (ID, "horizontal") diffusivities. Finally come the combined simulations, also re-ordered to have the simulations with larger DeltaTC come first. Sorting them by anticipated magnitude of total effects appears to be less useful, since the values of the explored parameters do not cover the full range of extreme values that could potentially be used in climate simulations (see response to point 3). The new version of the figure is attached to this response.
→ Figure updated accordingly

-- Fig. 8: Perhaps difference sections would be more useful?
*Response:* We wish to also show the structure of the water mass in the different states. Hence, we suggest adding difference sections as supplementary material, showing how SE5 and SE6 deviate from PIES278.
→ Difference sections added to supplementary material (Fig. S2)

-- Fig. 9: It would be more illustrative to zoom in with the colourbar.
*Response:* This figure will be updated to show Cpre – Csat, which improves the estimate of Cdis and shows that the previous method (calculating the difference between runs with normal and artificially fast gas exchange) did not fully reveal Cdis. See new, updated figure, which is attached to this response. When pre-formed tracers are available, the simulations with unrealistically fast gas exchange are no longer used for the analysis.
→ Figure updated

*** Fig. 10 showing the CO2 drawdown potential as a function of a change in the mean ocean temperature does not convey any messages. There seems no relationship between the two. Plus, if my reading is correct, water temperature does not control the strength/efficiency of the biological pump in the model. Therefore, there is no reason that the CO2 drawdown potential should be correlated with ocean temperature. Why don't the authors use other metrics such as the initial preformed PO4 as an X-axis instead, as was done in Marinov et al., 2008?

*Response:* The referee is correct in saying there is no direct relationship between mean ocean temperature and the efficiency of the biological pump. Any relationship between the two is indirect and due to the fact that the ocean circulation affects both variables. This is what we were trying to show with this figure. However, we agree with the referee that the figure is not crucially important for the story and it will therefore be removed. The figure described by the referee, which presents $CO_2$ drawdown potential and initial preformed $PO_4$, is present in the

manuscript, but in a slightly different format. Panel c in Figure 6 shows $CO_2$ drawdown potential on the Y-axis and global average P* on the X-axis. P* = $P_{reg}$ /P and $P_{pre}$ =   P – $P_{reg}$ (see Eqs. 2 and 8), so the quantities of P* and $P_{pre}$ are closely related. We therefore think an additional figure showing $CO_2$ drawdown potential vs. $P_{pre}$ would be redundant.

→ Fig. 10 removed together with the paragraph where it was discussed.

-- Figures: Please make the font size larger, particularly in Fig. 7-10
*Response:* This will be corrected in the updated manuscript.
→ Figures updated

-- Table 3: Please give units for DC; why not include DCcarb?
*Response:* The table shows correlation coefficients between circulation strength and DC, not values of DC in different ocean basins. The order of magnitude of DC could be specified in the table caption. DCcarb will be included in the updated manuscript. The first sentence of the table caption will be rephrased as follows: "Correlations between the changes in carbon species and the changes in strength of the zonal average overturning streamfunction (PSImax - PSImin) below 556 m depth in different geographical regions."
→ After further revisions, the caption is rephrased to: "Correlation coefficients of the changes in OV T (see Section 3.4), sorted by geographical regions, and the anomaly in each carbon species for the SE-ensemble (relative to PIES278)."

**List of relevant changes**

- Abstract
  - The abstract has been clarified to better describe the manuscript's relevance.

- Introduction
  - The introduction has been restructured and clarified to better describe the manuscript's relevance.

- Framework and general concepts:
  - The section has been significantly shortened and now lacks subsections. The concepts are briefly introduced, while most of the explanations of Sections 2.1 ('The ocean carbon pumps') and 2.3 ('Nutrient utilisation efficiency') have been moved to Appendix A, sections A1 and A2 respectively.

- Methods
  - The general model description (Section 3.1) has been clarified according to specific requests of the referees and the text that was in Section 3.1.2 ('Remineralisation scheme') has been shortened incorporated into 3.1.
  - A new Section 3.1.2 on the use of preformed tracers in the model has been added to the updated manuscript.
  - Section 3.2 has been shortened and some explanations of concepts have been moved to Appendices A2, B and B1. Concept explanations have been updated to match the methods that make use of preformed model tracers.
  - Section 3.3-3.4 on methods concerning the separation of carbon species have been moved to Appendices B3 and B2 respectively. Methods have been updated for the use of preformed model tracers, e.g. the regression model for preformed alkalinity has been removed and we no longer use runs with artificially fast gas exchange in order to capture the effects of $C_{dis}$.
  - A new section 3.3 'Sensitivity experiment tuning parameters' has been added, in accordance with referee requests.
  - A new section 3.4 'Overturning' has been added to introduce the metric OVT, which was not properly described in the initial manuscript.

- Results
  - All results of the carbon separation have been updated according to the new methods, which make use of preformed model tracers.
  - Ensemble members have been given names (descriptive abbreviations).
  - In the initial manuscript, the Results section mixed methods, results and discussion. We have attempted to make a clearer separation between these and move each part to where they belong. We have also made an effort to streamline the writing. The updated Results section is therefore shorter.
  - We have shifted the order of the subsections 4.1.3 and 4.1.4 to match the subsection order with the order of the carbon species as listed in Eqs. 4 and 5.
  - Fig. 10 has been removed along with the associated discussion in the final paragraph of Section 4.2.

- Discussion
  - The discussion has been re-written according to the results obtained with the updated methods using preformed model tracers.
  - A new section 5.2 'Implications of changes in OVT in relation to changes in carbon' has been added. Some of the contents were previously in the Results section, but have been moved here to get a better structure for the manuscript.

- Conclusions
  - The conclusions have been updated to better describe the manuscript's relevance
  - In order to shorten the Conclusions section, the final paragraph has been moved to Section 5.4

- Appendices
  - The old Appendix A is now Appendix C
  - Some concepts and methods described in Sections 2 and 3 have been moved to the new appendices A and B.

- References
  - Several new references have been added, some requested by referees and some that are relevant to other updates in the manuscript (Brovkin et al., 2012; DeVries and Primeau, 2009; Eggelston and Galbraith, 2017; Goodwin et al., 2015; Goyet and Poisson, 1989; Heinze et al., 1991; Ito and Follows, 2013; Klockmann et al., 2016; Marchitto and Broecker, 2006; Martin, 1990; Marzocchi and Jansen, 2017; Mehrbach et al., 1973; Munhoven, 2002; Ridgwell, 2001; Sijp et al., 2014; Sime et al., 2016; Stocker, 2014; von der Heydt and Dijkstra, 2006)
  - References that are no longer relevant for the text in the updated version of the manuscript have been removed (Aumont et al., 2016; Broecker, 1974; Zeebe and Wolf-Gladrow, 2001)

- Figures
  - Relevant figures that show carbon components have been updated with results given by the new methods using preformed model tracers
  - Font size has been increased in most figures
  - Fig. 2 has been updated (requested by referee #1), and now shows subpanels for each of the carbon components. The ensemble members have been re-ordered to give a clearer structure to the results. All other figures that list the SEs in order have been updated according to the same structure.
  - Figs. 6-8 have been re-ordered to match the new structure of the text
  - Fig. 7 (Fig. 6 in the updated manuscript) has been updated with results from non-linear calculations of $C_{sat,T}$
  - Fig. 9 has changed significantly, because the new results for $C_{dis}$ are given by the methods using preformed model tracers instead of subtraction of runs with artificially fast gas exchange.

- Tables
  - Tables have been updated with results given by the new methods using preformed model tracers
  - Tables have been re-ordered according to the new structure of SEs given in Fig. 2
  - A new Table 2 has been added, which compares diagnostic variables of the SE ensemble to PMIP/CMIP ensembles. The table is part of the new Section 3.3 on sensitivity experiments
  - Tables 2 and 3 have been re-numbered 3 and 4 respectively.

[revised manuscript text omitted]